# Identification and characterization of constrained non-exonic bases lacking predictive epigenomic and transcription factor binding annotations

Olivera Grujic [1,2], Tanya N. Phung[3], Soo Bin Kwon [2,3], Adriana Arneson[2,3], Yuju Lee[1],
Kirk E. Lohmueller [3,4,5] & Jason Ernst [1,2,3,6,7,8✉]

Annotations of evolutionary sequence constraint based on multi-species genome alignments and genome-wide maps of epigenomic marks and transcription factor binding provide important complementary information for understanding the human genome and genetic variation. Here we developed the Constrained Non-Exonic Predictor (CNEP) to quantify the evidence of each base in the genome being in an evolutionarily constrained non-exonic element from an input of over 60,000 epigenomic and transcription factor binding features. We find that the CNEP score outperforms baseline and related existing scores at predicting evolutionarily constrained non-exonic bases from such data. However, a subset of them are still not well predicted by CNEP. We developed a complementary Conservation Signature Score by CNEP (CSS-CNEP) that is predictive of those bases. We further characterize the nature of constrained non-exonic bases with low CNEP scores using additional types of information. CNEP and CSS-CNEP are resources for analyzing constrained non-exonic bases in the genome.

[1] Computer Science Department, University of California, Los Angeles, Los Angeles, CA 90095, USA. [2] Department of Biological Chemistry, University of California, Los Angeles, Los Angeles, CA 90095, USA. [3] Interdepartmental Program in Bioinformatics, University of California, Los Angeles, Los Angeles, CA 90095, USA. [4] Department of Ecology and Evolutionary Biology, University of California, Los Angeles, Los Angeles, CA 90095, USA. [5] Department of Human Genetics, University of California, Los Angeles, Los Angeles, CA 90095, USA. [6] Eli and Edythe Broad Center of Regenerative Medicine and Stem Cell Research at University of California, Los Angeles, Los Angeles, CA 90095, USA. [7] Jonsson Comprehensive Cancer Center, University of California, Los Angeles, Los Angeles, CA 90095, USA. [8] Molecular Biology Institute, University of California, Los Angeles, Los Angeles, CA 90095, USA. ✉email: jason.ernst@ucla.edu

A large majority of genetic variation associated with common disease falls into non-exonic regions of the human genome[1] motivating in part the need to better annotate and understand such regions. Annotations of evolutionary constraint based on multiple-sequence alignments[2–5] and maps of epigenomic marks and transcription factor (TF) binding represent two complementary types of information to annotate the non-exonic genome[6–9]. Supporting the importance of evolutionary constraint annotations, heritability analyses have suggested they are heavily enriched for disease-associated variants[10] and they have been important features to integrative methods for prioritizing potentially deleterious non-exonic mutations[11,12].

While useful, evolutionary constraint annotations do not directly provide information on the type of genomic element or the cell or tissue types of activity. Genome-wide maps of histone modifications and variants, chromatin accessibility, chromatin state annotations, and TF binding can give such insights[6,7,13–15]. However, such data is specific to the condition and cell or tissue type of the experiments. Previous analyses have shown that while there is an enrichment for evolutionarily constrained bases in regulatory-associated epigenomic or TF-binding annotations, some evolutionarily constrained bases lack informative annotations[4,14,16–21].

Thus, when investigating the role of constrained non-exonic (CNE) elements, or variants within such elements, with a compendium of epigenomic and TF-binding data, an initial question is whether it is possible to even explain the constraint with data in the compendium. However, with tens of thousands epigenomic and TF-binding datasets available, answering this question is not straightforward. Integrative scores such as CADD[12] that combine epigenomic and TF features with conservation features cannot be directly applied to answer such a question, since a base could receive a high score based on conservation features even without informative epigenomic or TF-binding features.

A few scores have been proposed that quantify information or activity in epigenomic or TF-binding annotations where evolutionary information is used to learn a mapping from epigenomic or TF-binding features to a score, but not as features for predictions. FitCons and FitCons2[22,23] quantified information in epigenomic annotations using probabilistic evolutionary models to provide cell type-specific estimates of fitness. The 'conservation-associated activity score' used evolutionary constraint information to map Segway genome annotations to cell type-specific scores[24]. Additionally, for these approaches a single summary score was derived from the cell type-specific scores. While informative, an approach that summarizes information in a compendium of epigenomic and TF data in a single score without going through cell type-specific scores would have greater flexibility on what datasets are used and how they are combined. Methods also exist to computationally predict epigenomic marks, TF-binding data, and constrained elements from sequence[25–28], but do not directly provide information on whether experimental data in a compendium can explain constraint.

Here we develop the constrained non-exonic predictor (CNEP) to produce a single probabilistic score per base, without respect to cell type, that reflects the evidence within a large-scale compendium of epigenomic and TF-binding data that the base is in a non-exonic evolutionarily constrained element, as defined by prior methods for calling them[2–5]. The CNEP score thus quantifies for a researcher interested in specific CNE elements, or variants within them, the extent to which existing data in the compendium can explain the constraint. Furthermore, the CNEP score enables investigating more general scientific questions about the extent to which information in current epigenomic and TF-binding annotations can explain non-exonic constraint, and the nature of constrained bases that cannot be explained by available

data. We focus specifically on non-exonic bases since they comprise the vast majority of the genome, are less well annotated compared to exons, and constraint in such bases is expected to be largely associated with distinct patterns of epigenomic marks and TF binding relative to that found in exons.

We apply CNEP with a compendium of over 60,000 human epigenomic and TF-binding features. We show that while the CNEP score is able to effectively predict many bases in CNE elements, outperforming baseline and related existing scores, a substantial portion are still not well predicted by CNEP. Using human genetic variation data, we show that a portion of those bases truly appear to be under constraint, and using regulatory sequence motif annotations, experimental data from mouse, and retrospectively considered additional human experimental data we provide insights into their potential role. We also analyze bases that receive a high CNEP score, but are not in a constrained element, a portion of which may correspond to adaptive evolution or changing selective effects over time. In addition, we develop a complementary conservation signature score by CNEP (CSS-CNEP), which we use to identify CNE bases with low CNEP scores that are more likely due to false constraint calls. CNEP and CSS-CNEP are resources for analyzing the genome and variants in terms of large-scale epigenomic and TF-binding data, evolutionary constraint annotations, and their relationship.

## Results

**CNEP method**. We developed the CNEP to make a probabilistic prediction using features defined from large-scale epigenomics and TF-binding data as to whether a base in the human genome is in a CNE element previously called from comparative genomics sequence analysis[2–5] (Fig. 1). We applied CNEP jointly with 63,741 features derived from overlap of peak calls in experiments mapping histone modifications, chromatin accessibility, and TF binding, including general factor binding that is not necessarily sequence specific, as well as TF-binding footprint calls from Digital Genomic Footprinting, and ChromHMM chromatin state annotations (Supplementary Data 1; see "Methods" section). The features are based on data from the Roadmap Epigenomics and ENCODE consortia, or public data from the Gene Expression Omnibus, ArrayExpress, or Sequence Read Archive data curated and reprocessed in the ReMap or ChIP-Atlas databases[6–9,29,30].

The median number of features that overlap a base in the genome is 532, of which at most 263 are chromatin state features. In total 75% of genomic bases were in a peak in at least one DNase I hypersensitivity experiment, which measures chromatin accessibility, and 51% were when restricting to the 350 such experiments from Roadmap Epigenomics (Supplementary Fig. 1). This highlights the limited specificity of such criteria for predicting constraint.

CNEP trains an ensemble of logistic regression classifiers to discriminate between bases overlapping evolutionarily constrained elements outside of annotated exons and those bases in the rest of the genome (see "Methods" section). We trained CNEP with multiple different constrained element sets by applying CNEP separately with each, and then averaging the resulting predictions producing what we termed the CNEP score (Fig. 2a, Supplementary Fig. 2). We applied CNEP with constrained element sets previously produced by four methods: PhastCons[5,31], GERP++[2], SiPhy-pi, and SiPhy-omega[3,4]. These constrained element sets highly enrich for non-exonic bases prioritized by a number of scores used to predict the relative phenotypic impact of genetic variants, including scores that integrate diverse annotations (Supplementary Fig. 3). While CNEP uses constrained element annotations as labels for training, we emphasize that CNEP does not use comparative genomics

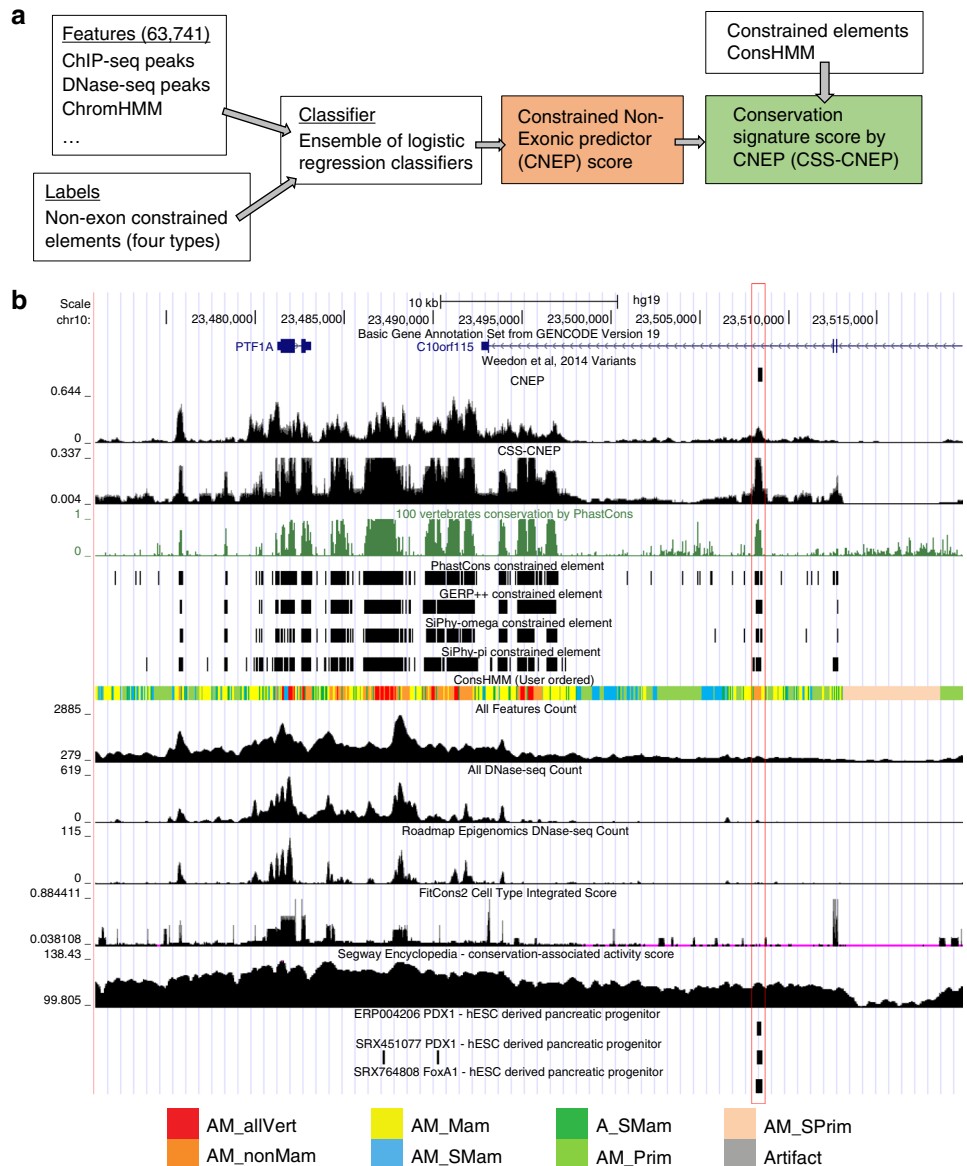

**Fig. 1 Overview and example predictions of CNEP and CSS-CNEP. a** A flow chart giving an overview of the approach to learn the CNEP score and the CSS-CNEP, which are shown in shaded boxes. The input to learn the CNEP score were a set of 63,741 epigenomic and TF-binding features such as ChIP-seq and DNase-seq peaks and ChromHMM annotations. These features were integrated using an ensemble of logistic regression classifiers that were trained based on annotations of constrained non-exonic elements from four methods. The CSS-CNEP was derived based on the CNEP score along with the four constrained element sets and ConsHMM annotations[32]. **b** An example genomic locus containing the *PTF1A* gene illustrating the CNEP score and the CSS-CNEP. The top line is the GENCODE gene annotation track. The next line shows the location of five variants in close proximity that were previously identified to be associated with isolated pancreatic agenesis and fell into constrained non-exonic elements, but had limited prior informative epigenomic and TF-binding annotations[21]. A vertical box goes through these variants. The following tracks show the CNEP score, CSS-CNEP, and the PhastCons score. This is followed by tracks for the PhastCons, GERP++, SiPhy-omega, and SiPhy-pi[2,3,5] constrained element sets and the ConsHMM annotations[32] in dense view. Along the bottom is a color legend for the groups of ConsHMM conservation states as defined in ref. [32]. For segments in red, species through fish have high aligning and matching to the human reference genome, while those in orange have that through some non-mammalian vertebrates. The next sets of tracks show baseline scores of counts of total features, DNase-seq features, and DNase-seq features from Roadmap Epigenomics overlapping a base followed by the FitCons2 cell type integrated score[22] and the Segway Encyclopedia conservation-associated activity score[24]. The final set of tracks show peak calls for PDX1 and FoxA1 in hESC-derived pancreatic progenitors[21,49] that overlapped the SNP. These datasets were ranked using the CNEP software to be the top three datasets in terms of their overlap with constrained non-exonic elements as defined by the expected CNEP score statistic.

data as features for predictions. By using an ensemble of logistic regression classifiers, CNEP provides a robust probabilistic output. For each chromosome, CNEP trains a separate set of classifiers based on subsamples of positions from all chromosomes except the target chromosome. CNEP then makes a probabilistic prediction for each base on the target chromosome as to whether it is in a CNE element.

CNEP's predictions from training using any two different individual constrained element sets were all highly correlated ([0.88,0.93]), and greater than the correlations directly between different constrained element sets (Supplementary Fig. 4). As the CNEP score is based on the average of predictions based on training separately with four different constrained element sets, this is expected to reduce biases associated with the choice of one

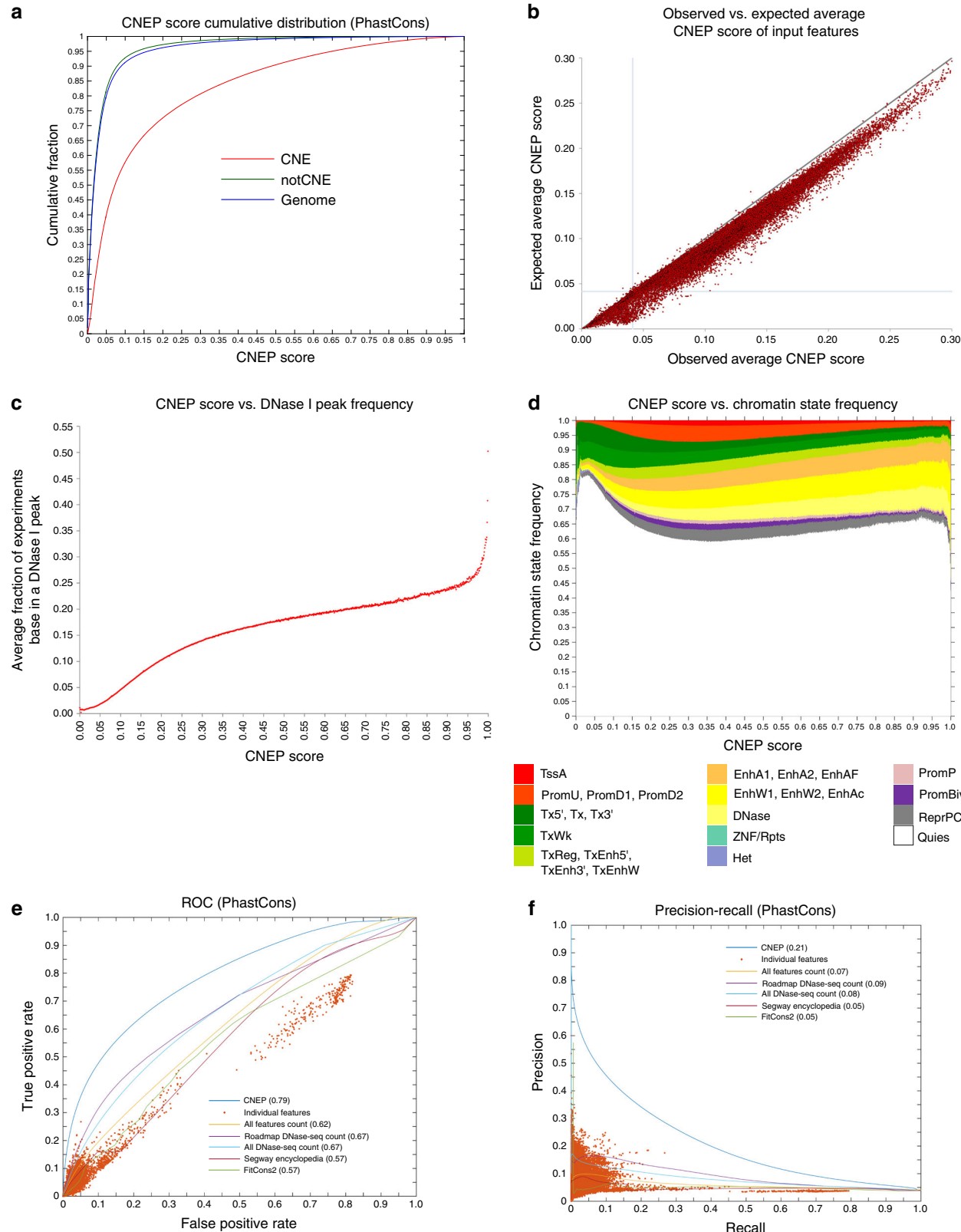

specific constrained element set. Consistent with this, the CNEP score correlation with CNEP's predictions based on training on a single constrained element set were even higher ([0.96,0.98]).

As an example of the CNEP score, we consider the *PTF1A* gene locus, which a previous study identified variants causal for isolated pancreatic agenesis in distal CNE bases[21] (Fig. 1b). The

study noted that the variants lacked informative annotations from the ENCODE or Roadmap Epigenomics projects[21], but obtained them by mapping epigenome marks and TF binding in human embryonic-derived pancreatic progenitor cells. The CNEP score is relatively high for those variant positions, conveying that there now exists epigenomic or TF datasets that explains the constraint,

**Fig. 2 Properties of the CNEP score. a** The graph shows the cumulative distribution of the CNEP score genome-wide (Genome), in PhastCons constrained non-exonic (CNE) bases, and bases that are not in PhastCons-constrained elements and also not in exons (notCNE). **b** A scatter plot with each point corresponding to one feature that CNEP uses. The *x*-axis shows the average CNEP score in bases that have the feature present, while the *y*-axis shows the expected CNEP score based on the feature's overlap with constrained non-exonic bases. Only 48,364 features that cover at least 200 kb are shown. The full set of values can be found in Supplementary Data 2. The diagonal line is the *y* = *x* line. The vertical line corresponds to the genome-wide average CNEP score. The horizontal line corresponds to the genome-wide expected average CNEP score. **c** A plot showing the average fraction of the 350 Roadmap DNase I experiments in which the base overlaps a called peak for each CNEP score value, rounded to the nearest 0.001, covering at least 1000 bases. In total, there was 1000 such values. **d** A plot showing the average fraction of bases annotated across the 127 epigenomes to each of 14-groups defined based on 25 ChromHMM chromatin states previously assigned the same color[40] for each CNEP score value, rounded to the nearest 0.001. A color legend with the state mnemonics from ref. [40] is displayed at the bottom of the panel. **e** A plot of the ROC curve for the CNEP score predicting PhastCons non-exonic bases. Also shown is the performance of individual features and several baseline or existing scores (see "Methods" section). Area under the curve values are shown in parentheses. **f** A similar plot as **e** except for precision-recall as opposed to ROC curves. ROC and precision-recall curves for other constrained element sets can be found in Supplementary Fig. 5. Source data are provided as a Source Data file.

though this would not be apparent from several baseline or related existing scores. For example the initial discovered associated variant (chr10:23508437) was in the top 5.4% by CNEP, while at the top 22.8% for the Segway Encyclopedia 'conservation-associated activity score'[24] and at the top 46.8–49.4% for the FitCons2 'cell-type integrated score'[22] and three baseline scores (see "Methods" section).

**CNEP score associates with signatures of regulatory activity**. We next investigated the relationship between the CNEP score and input features to CNEP. We compared statistics of the observed genome-wide average CNEP score of bases overlapped by each feature and the expected average CNEP score defined as the proportion of the feature's bases overlapping with CNE elements on average for the four element sets (see the "Methods" section, Supplementary Data 2, Fig. 2b). These statistics were highly correlated (0.99 pearson correlation for features covering at least 200 kb). We also confirmed that the average genome-wide observed value of the CNEP score, 0.0419, was close to its expected value, 0.0415, defined based on the proportion of the genome in a CNE element on average for the four-element sets.

We investigated whether bases that received a higher CNEP score were more likely to show signatures of regulatory activity in more datasets. We computed as a function of the CNEP score the average number of experiment in which a base would be covered by a peak from a set of 350 DNase I hypersensitivity experiments from Roadmap Epigenomics (Fig. 2c), which showed that bases with a higher CNEP score tended to be in a peak for more experiments. For example, bases with a CNEP score of 0.500 were in a peak in 18.0% of the experiments on average compared to the genome-wide average of 1.9%. We saw a similar pattern when considering as a function of the CNEP score the frequency of enhancer and promoter chromatin states from a chromatin state model defined across 127 reference epigenomes (Fig. 2d).

**CNEP performance at predicting CNE bases**. We next analyzed the performance of the CNEP score at predicting bases in CNE elements in the genome separately for each element set (Fig. 2e, f, Supplementary Fig. 5). CNEP's area under receiver operator characteristic (ROC) curves (AUC) were in the range [0.79,0.86] with the AUC values for PhastCons elements lower than the other three constrained element sets. The lower AUC for PhastCons might be partly related to this being the only element set defined using non-mammalian vertebrates, which can make it more vulnerable to possible alignment errors, or due to the higher resolution at which the method calls constrained elements[32].

To place CNEP's predictive performance in perspective, we made a number of comparisons (Fig. 2e, f, Supplementary Fig. 5, Supplementary Table 1). The CNEP score had a better true

positive rate at the same false positive rate than any individual input feature and likewise for precision at the same recall. The AUC values for CNEP ([0.79,0.86]) were greater than for the baselines of the number of input features ([0.62,0.69]) and the number of DNase I hypersensitivity experiment features overall ([0.67,0.74]) and from Roadmap Epigenomics ([0.67,0.74]) overlapping a base. Two related existing scores, Segway Encyclopedia 'conservation-associated activity score'[24] and Fit-Cons2 'cell-type integrated score'[22], had even lower AUC values, in the ranges [0.57,0.66] and [0.57,0.62], respectively. AUC values for sequence-based predictions of 919 chromatin features[25] were also lower (maximums in the range [0.64,0.66]). We saw similar results when excluding any base overlapping an exon from the negative set. Using CNEP's predictions trained using only the constrained element set being evaluated also gave similar results. Applying CNEP with a random forest in place of a logistic regression classifier decreased predictive performance a small amount. We note these evaluations were specifically comparing predictions of CNE bases and not predictions of adaptive and recently evolved bases.

**Subset of CNE bases not predicted by CNEP**. To better understand CNEP predictions that disagreed with the annotation of whether a base was in a CNE element, we first defined six sets of bases for each constrained element set. We defined the set CNE as bases covered by a constrained element and not in an exon and the set notCNE as bases not in CNE and also not in an exon. We partitioned CNE bases into the Low_CNE and High_CNE subsets depending on whether its CNEP score was below or above the genome-wide average, respectively, and similarly partitioned notCNE into the Low_notCNE and High_notCNE subsets (Fig. 2a, Supplementary Fig. 2, Supplementary Table 2). For different constrained element sets, Low_CNE bases covered between 0.6% and 1.3% of the genome corresponding to 20.9–34.6% of CNE bases. A substantial fraction of CNE bases thus had a low CNEP score. High_notCNE bases covered 19.6–21.3% of the genome, and had a higher CNEP score than all Low_CNE bases, despite the former not overlapping a base in a constrained element.

To investigate the extent to which limitations in the resolution of the epigenomic and TF-binding data relative to the resolution at which CNE bases are defined can explain High_notCNE bases, we computed their cumulative prevalence as a function of distance to the nearest CNE base and the enrichment relative to notCNE bases (Supplementary Fig. 6). For example, for PhastCons, High_notCNE bases, had a 2.7-fold enrichment immediately next to a CNE base, decreasing to 1.8-fold within 200 bp. In total, 57% of High_notCNE bases were within 200 bp of a CNE base while 43% were greater than that distance.

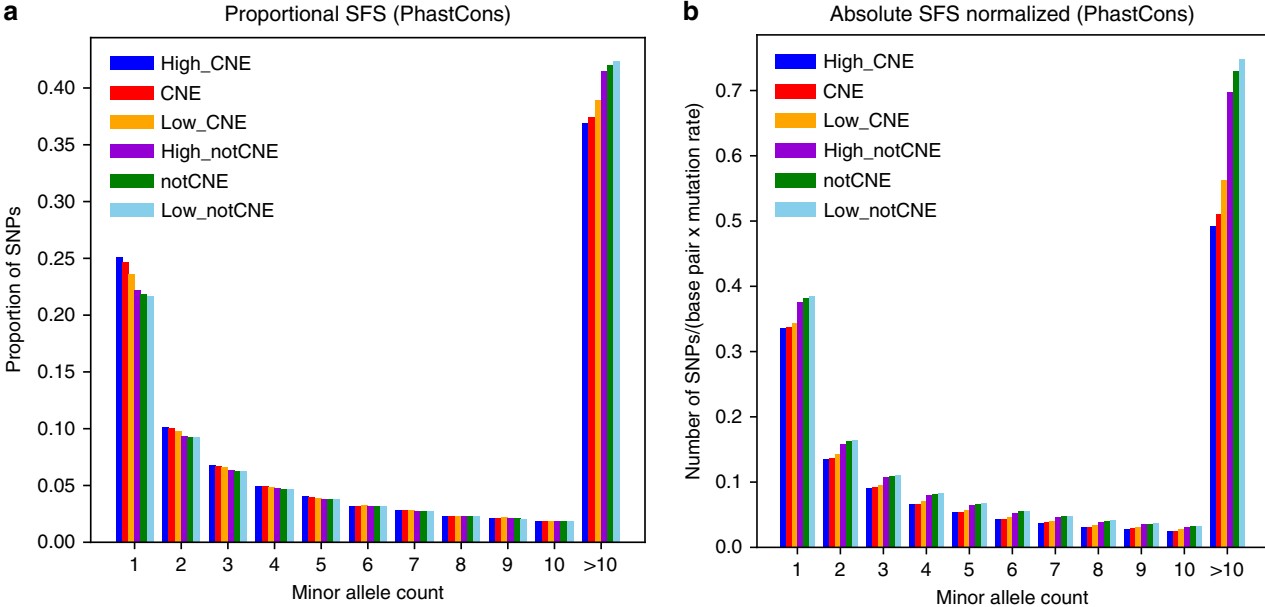

**Fig. 3 CNEP score's relationship to human variation. a** The plot shows for PhastCons High_CNE, CNE, Low_CNE, High_notCNE, notCNE, and Low_notCNE bases, the proportional site frequency spectrum based on a set of 105 unrelated individuals in the YRI population in terms of number of SNPs per base pair eligible for a SNP to be called, normalized for the number of sites with a variant in each set (see "Methods" section). The last column includes all SNPs with minor allele count >10. **b** Similar plot to **a**, except showing the absolute site frequency spectrum per base pair eligible for a SNP to be called normalized by estimated mutation rates. Corresponding plots for additional constrained elements can be found in Supplementary Fig. 12 and plots controlling for difference in background selection can be found in Supplementary Fig. 13. Plots at higher thresholds of the CNEP score for notCNE bases can be found in Supplementary Fig. 14. Source data are provided as a Source Data file.

We also verified proximity to exons provides a limited explanation of Low_CNE bases. Specifically, for Low_CNE bases, we computed the cumulative prevalence as a function of distance to nearest exon and the enrichment relative to CNE bases (Supplementary Fig. 7). For example, for PhastCons, only 10.2% of Low_CNE bases were within 200 bp from an exon at a 1.3-fold enrichment, a decrease from 1.8-fold enrichment immediately next to an exon.

We further analyzed High_notCNE and Low_CNE bases for spatial enrichments relative to genes, transcription start sites (TSS), and exons, and enrichments for chromatin states and repeat elements (Supplementary Figs. 7–11). This highlighted the enrichment of High_notCNE bases relative to notCNE bases proximal to TSS and exons, in enhancers and promoter chromatin states, and in the DNA class of repeats. Among chromatin states, Low_CNE bases relative to CNE bases showed the strongest enrichment for the heterochromatin and zinc finger gene states.

**Evidence of purifying selection in humans for Low_CNE bases**. To test whether Low_CNE bases are still enriched for bases under purifying selection in humans despite the low CNEP score, we turned to human genetic variation data. Specifically, we considered a set of 105 unrelated individuals of the Yoruba in Ibadan (YRI) population from the 1000 Genomes Project[33] and first examined the proportional site frequency spectrum (SFS) (see "Methods" section). Comparing Low_CNE bases to High_notCNE bases, we observed that there is a significant difference in the distribution ($p < 10^{-15}$), with a greater proportion of low-frequency variants for Low_CNE bases, especially singletons and doubletons, and a lower proportion of common variants (Fig. 3a, Supplementary Fig. 12a, c, e, comparing orange and purple bars). The skew towards low-frequency variants and the deficit in high-frequency variants suggest stronger purifying selection in Low_CNE bases relative to High_notCNE bases. As an additional

evaluation of whether purifying selection has been stronger in Low_CNE bases as compared to High_notCNE bases, we examined the absolute SFS normalized by the number of base pairs and an estimated average mutation rate[34]. There were fewer SNPs in Low_CNE bases relative to High_notCNE bases across all bins of allele frequencies (Fig. 3b, Supplementary Fig. 12b, d, f, comparing orange and purple bars). These results further suggest that Low_CNE bases have experienced stronger purifying selection than High_notCNE bases. We obtained similar results when controlling for estimated background selection[35] (Supplementary Fig. 13).

High_CNE relative to Low_CNE bases and High_notCNE relative to Low_notCNE bases had reduced common variation (Fig. 3, Supplementary Fig. 12). However, these differences were generally smaller than the differences of CNE to notCNE bases. Additionally, we compared SFS of Low_CNE bases to subsets of High_notCNE bases that satisfied more stringent CNEP score thresholds (Supplementary Fig. 14), which suggested that Low_-CNE are under stronger purifying selection in humans than notCNE bases with substantially higher CNEP scores.

**Low_CNE bases show enrichments for TF-binding motifs**. Since human population genetics data still supported the importance of Low_CNE bases despite the low CNEP score, we investigated whether regulatory sequence motif analysis, which is cell type and condition invariant, provides evidence of a regulatory role for them. For Low_CNE bases and the other five sets defined above, we computed the distribution of 1646 regulatory motif enrichments relative to control motifs and normalized the distribution relative to that obtained when randomizing motif instances, separately for each constrained element set (Fig. 4a, Supplementary Fig. 15, see "Methods" section). Both Low_CNE and High_notCNE bases had motif enrichments that were above background, though less than High_CNE bases. The Low_CNE enrichments were substantially stronger than the High_notCNE

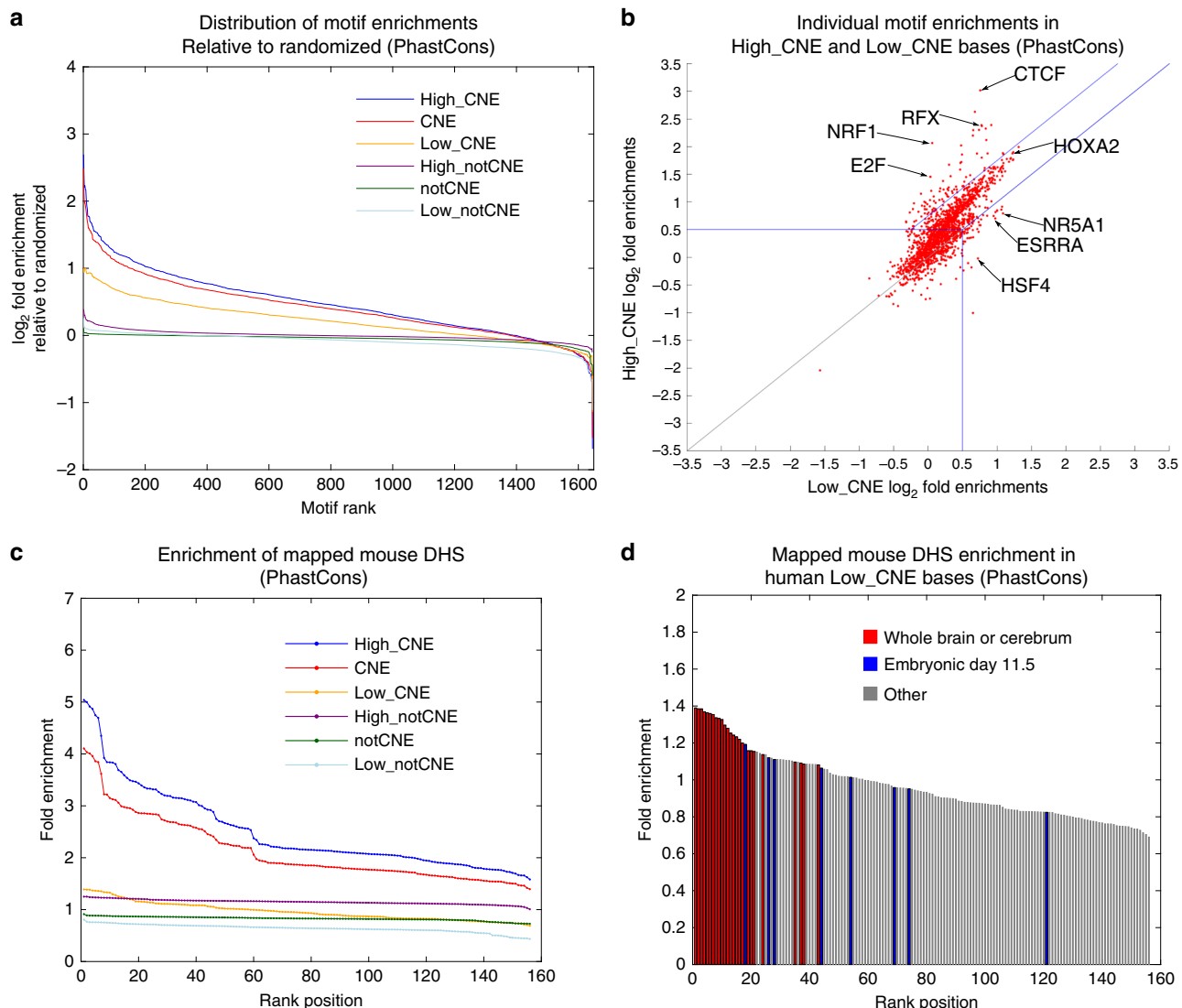

**Fig. 4 CNEP score's relationship to TF sequence motifs and DHS in mouse. a** The plot shows the difference of the distribution of motif enrichments relative to the distribution for a randomized set of the motifs for the PhastCons High_CNE, CNE, Low_CNE, High_notCNE, notCNE, and Low_notCNE bases. The x-axis is the rank position of the motif among the 1646 motifs. The y-axis is the difference between the $\log_2$ fold enrichment based on the actual motif calls and the median $\log_2$ fold enrichment from three randomized versions at the same rank position (see "Methods" section). Similar plots for other constrained elements can be found in Supplementary Fig. 15 and at other thresholds for defining notCNE high bases in Supplementary Fig. 16. **b** Scatter plot of individual motif enrichments. The x and y axes corresponds to the $\log_2$ fold enrichments in PhastCons Low_CNE and High_CNE bases, respectively. The blue lines separate the three regions used for the GO-enrichment analysis, 'High_CNE strongly preferred', 'High_CNE moderately preferred', and 'Low_CNE preferred', where at least one of the Low_CNE or the High_CNE $\log_2$ enrichment is ≥0.5 (Supplementary Data 3 and 4). The gray line is the $y = x$ line where both Low_CNE and High_CNE $\log_2$ enrichments are <0.5. Similar plots based on other thresholds of the CNEP score can be found in Supplementary Fig. 17. Selected motifs are labeled. **c** The distribution of enrichments for DHS from 156 experiments in mouse, where the sites are mapped to human and enrichments are computed relative to enrichments for randomized DHS, for PhastCons High_CNE, CNE, Low_CNE, High_notCNE, notCNE, and Low_notCNE bases (see "Methods" section). Similar plots for other constrained elements can be found in Supplementary Fig. 18. **d** A bar graph corresponding to the enrichments shown in **c** for Low_CNE bases. Bars are colored to indicate if the experiment is of whole brain or cerebrum, embryonic day 11.5, or neither. Similar plots for other constrained elements can be found in Supplementary Fig. 19. A full set of enrichment values can be found in Supplementary Data 5. Source data are provided as a Source Data file.

and notCNE enrichments. Low_CNE enrichments were also greater than enrichments for High_notCNE bases defined at more stringent thresholds of the CNEP score (Supplementary Fig. 16), consistent with the SFS analysis results.

We also analyzed individual motif enrichments for Low_CNE and High_CNE bases (Fig. 4b, Supplementary Fig. 17, Supplementary Data 3). While globally High_CNE bases had stronger motif enrichments than Low_CNE bases, some motifs did have

stronger enrichments for Low_CNE bases. We analyzed Gene Ontology (GO) enrichments for TFs corresponding to three subsets of motifs enriched in High_CNE or Low_CNE bases (Fig. 4b, Supplementary Data 4, see "Methods" section). TFs associated with 'High_CNE strongly preferred' motifs enriched for protein dimerization activity and core-promoter GO terms. TFs associated with 'High_CNE moderately preferred' motifs enriched for development-related GO terms. Finally, TFs

associated with 'Low_CNE preferred' motifs enriched for lipid binding, signaling, and response to stimulus-related GO-terms (corrected $p$-values < 0.05). These results suggest that some Low_CNE bases are associated with motifs of TFs that might only be active in specific developmental stages or under specific stimuli.

**Low_CNE bases enrichments for mapped mouse DNase data**. As mouse experiments relative to human may have coverage of additional tissue types and developmental stages, for each constrained element set, we investigated the enrichments for the six sets defined above for mouse DNase I hypersensitive sites (DHS) from 156 experiments[36,37]. For each experiment, we mapped the mouse DHS to the human genome and did the same for a randomized version (see "Methods" section). We computed enrichments based on the observed dataset overlap relative to the randomized one.

For all constrained element sets, Low_CNE bases enriched at least for some experiments (Fig. 4c, Supplementary Fig. 18). These enrichments were modest, not exceeding two-fold, and lower than for High_CNE bases. However, they were greater than for Low_notCNE and notCNE bases and comparable to High_notCNE bases for at least the most enriched experiments. We observed that the DHS experiments that tended to have the greatest enrichment for Low_CNE bases were for whole brain or cerebrum (Fig. 4d, Supplementary Fig. 19, Supplementary Data 5). For example, for PhastCons, the 21 experiments with the greatest enrichment included 20 of the 25 whole brain and cerebrum experiments. The only other experiment in the top 21 was of mesoderm in day 11.5 embryos. These results suggest that some Low_CNE bases may correspond to DHS, particularly related to the brain.

**Retrospective analysis of information in additional datasets**. We conducted a retrospective analysis to gain insight into the extent to which additional datasets improve the predictive performance of CNEP and the nature of individual datasets that provide additional marginal information predictive of CNE bases even after thousands of datasets are considered. Specifically, we generated another CNEP score using 10,836 features that were available and accessible by 2015 (see "Methods" section). The AUC for this score for predicting CNE bases was in the range [0.75,0.82], a reduction from [0.79,0.86] when using all features (Supplementary Table 1).

This suggests that some individual datasets provide additional marginal information predictive of CNE bases even after conditioning on the information in the 10,836 features. To identify such datasets we defined the CNEP underestimation value of a dataset as the difference between the statistics of the expected average CNEP score based on the dataset overlap with CNE bases and the observed average value of the CNEP score for bases the dataset covers (see "Methods" section). High CNEP underestimation values identify datasets that provide among bases the dataset covers particularly informative annotations of CNE bases beyond the information already in the 10,836 features. We note that datasets with CNEP underestimation values close to zero can still be highly informative of CNE bases when considered in isolation, and those with a negative CNEP underestimation values had observed average CNEP score statistics greater than the expected. We analyzed the CNEP underestimation values as a function of the dataset base coverage.

For most datasets the CNEP underestimation value was relatively small (<~0.01) or the dataset covered few bases, indicating that the additional marginal information for annotating CNE bases in those datasets is limited (Fig. 5, Supplementary

Figs. 20 and 21, Supplementary Data 6). Among the exceptions was a dataset for a DNase I hypersensitivity experiment in spinal cord of a 59-day embryo that covered 71.6 million bases and had an underestimation value of 0.066. For comparison, the underestimation value of the 12.3 million new exon bases added to GENCODE between v19 and v28 was 0.038. Other exceptions included DNase I hypersensitivity of embryonic brain, eye, and retina. Some TF-binding datasets had even greater underestimation values than these DNase I experiments, though covered a smaller fraction of the genome. Datasets for the TFs TEAD4 and ONECUT1 still had both greater genome coverage and underestimation values than new exons. We used the ChIP-Atlas[9] metadata to determine if datasets that covered at least 200 kb and had CNEP underestimation values >0.02 enriched for specific cell type classes revealing significant enrichments for pluripotent stem cell, pancreas, and neural classes (corrected $p$-value < 0.03; Supplementary Table 3).

**Conservation states and CNEP predictions**. Even with additional datasets, CNEP would not be expected to predict CNE bases that are in false-positive constraint calls. We investigated if we could identify CNE bases that are difficult for CNEP to predict using additional signals present within a multiple-sequence alignment not captured by the original unsupervised model used to call CNE bases. Establishing this, would suggest it is possible to identify CNE bases with low CNEP scores that more likely represent false constraint calls. Specifically for this analysis we used ConsHMM conservation states annotations based on the combinatorial and spatial patterns of which species match and align the human genome in a 100-way vertebrate sequence alignment, defined in an analogous way to ChromHMM chromatin states[32,38].

We observed large variation in CNEP scores' ability to predict CNE bases depending on the ConsHMM conservation state they overlap (Fig. 6, Supplementary Figs. 22–24). For example, for PhastCons, which was defined on the same alignment as ConsHMM, while CNEP's overall AUC for predicting CNE bases was 0.79, the AUC reached 0.92 for CNE bases in ConsHMM state 2, which is associated with high frequency of all vertebrates except fish aligning to and matching human. In contrast, the AUC was <0.650 for 56 states, generally having low align frequencies for many mammals, and comprising 17.4% of all CNE bases. After excluding bases within 200 bp of exons, the AUC reached 0.95 for state 1, associated with high frequency for all vertebrates aligning and matching. We saw similar trends in the conservation state enrichments of High_notCNE and depletions of Low_CNE bases relative to notCNE and CNE bases, respectively (Fig. 6a, b, Supplementary Fig. 22).

**CSS by CNEP**. We defined the CSS by CNEP (CSS-CNEP) corresponding to the expected CNEP score of a base given a set of comparative genomic annotations for the base (see "Methods" section, Figs. 1, 7a, Supplementary Fig. 25). CSS-CNEP can score among Low_CNE bases those that more likely represent false constraint calls as opposed to the compendium lacking informative experimental data for the bases. Specifically, the CSS-CNEP for a base on one chromosome was computed as the average CNEP score in non-exonic bases on other chromosomes that had the same conservation state and combination of overlapping constrained elements among the four sets.

The average CSS-CNEP score in CNE bases showed a large variation across conservation states (Fig. 6, Supplementary Fig. 22). For example, for PhastCons, CNE bases in states 1 and 2 had average CSS-CNEP values of 0.32 and 0.30, respectively, while CNE bases in state 100, associated with putative alignment

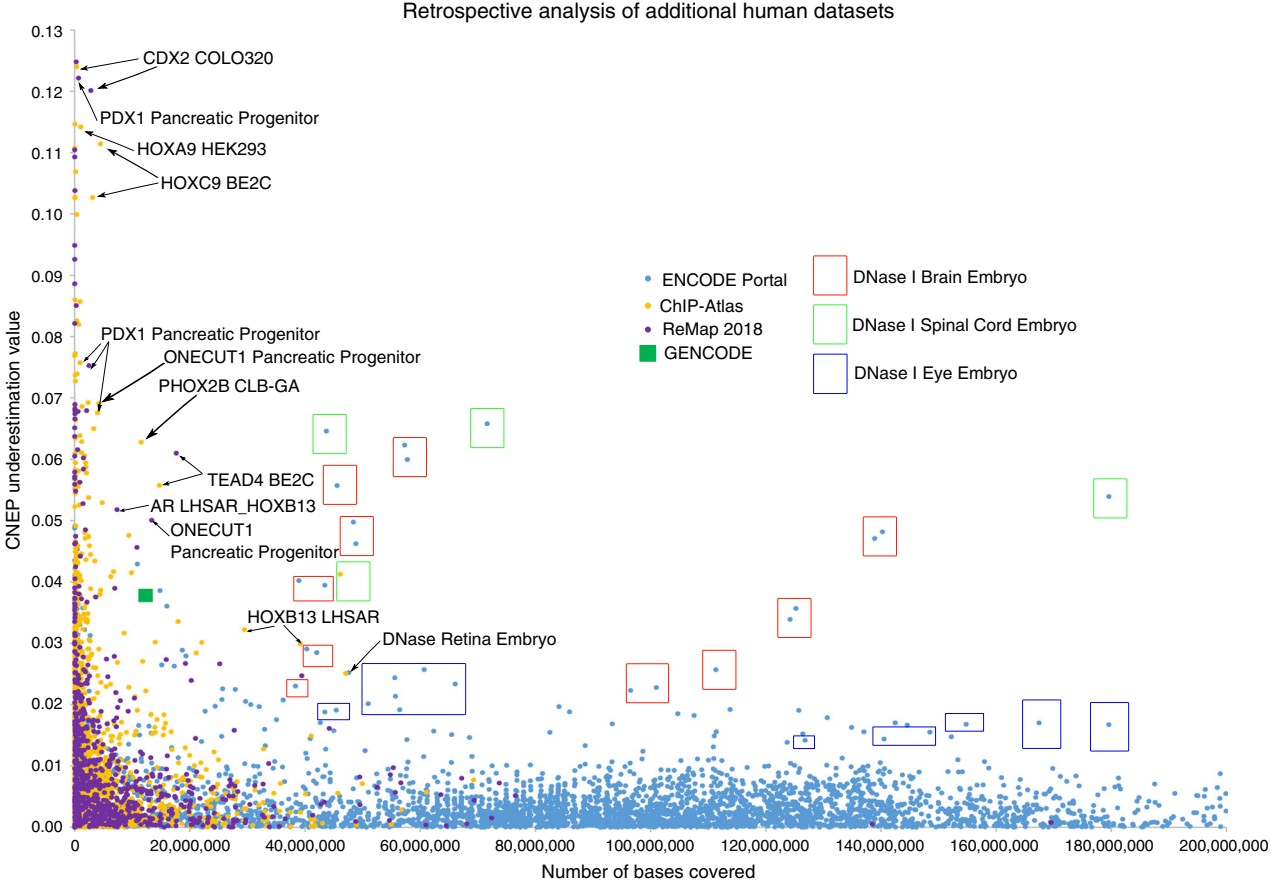

**Fig. 5 Retrospective analysis of information in additional human datasets.** Scatter plot of additional human datasets in a retrospective analysis. The *x*-axis is the number of bases covered by the dataset. The *y*-axis is the CNEP underestimation value for the dataset. Selected datasets with a high combination of base coverage and CNEP underestimation values are labeled or placed in a box if they correspond to a DNase I hypersensitivity experiment of embryonic brain, spinal cord, or eye. The color of the box corresponds to brain, spinal cord, or eye as indicated in the legend. The color and shape of the points are based on whether the point corresponds to a ChIP-Atlas, ENCODE portal, or ReMap 2018 dataset or the set of new GENCODE exons between v19 and v28. Only datasets with a positive underestimation value are shown. Datasets covering more than 200 million base pairs are not shown, but all had an underestimation value of <0.01. Three datasets that had an underestimation value >0.13 are not shown, but all covered <9000 base pairs. Versions of this plot showing negative CNEP underestimation values, for input features used by CNEP to generate the predictions, and for shuffles of additional datasets can be found in Supplementary Fig. 21. Source data are provided as a Source Data file.

artifacts[32], had an average CSS-CNEP value of 0.03. We evaluated regulatory sequence motifs and mouse DHS enrichments of subsets of CNE bases that had CSS-CNEP values less than specific thresholds (Fig. 7c, d, Supplementary Figs. 26 and 27). This showed substantially reduced enrichments for CNE bases that had lower CSS-CNEP values. These results suggest that CSS-CNEP can be used to identify CNE bases that more likely correspond to false-positive constraint calls.

We confirmed that CSS-CNEP was more predictive of Low_CNE bases among CNE bases than any existing constraint score (Fig. 7b, Supplementary Fig. 28). Additionally, in most cases compared to individual constrained element sets, CSS-CNEP had greater precision at their single recall rate. However, some Low_CNE bases were not well predicted by CSS-CNEP. Consistent with this even for CNE bases in ConsHMM states for which CNEP was most predictive, there was still a substantial subset of bases receiving low CNEP scores (Fig. 6, Supplementary Fig. 22). For example, for PhastCons, 19% of Low_CNE bases were in states 2, 4, and 5, which all had greater than four-fold enrichment for Low_CNE bases. Many of these bases may correspond to true constraint calls, but are not active in experiments provided to CNEP.

## Discussion

In this work, we developed and applied the CNEP to provide a score for each base of the human genome that reflects the probability that the base overlaps a CNE from information in large-scale collections of epigenomic and TF-binding data. We used information from an input of more than 60,000 features derived from epigenomic and TF-binding data spanning a wide range of cell and tissue types. We showed that CNEP outperformed baseline and related existing scores at predicting CNE bases using epigenomic and TF-binding data, but that there was still a substantial portion of CNE bases that were not well predicted. For example, for PhastCons, 35% of CNE bases had a CNEP score below the genome-wide average, while 23% of notCNE bases had a score above that average.

We conducted a number of analyses to better understand CNE bases that received a low CNEP score, as well as notCNE bases that received a high CNEP score. Using human population genetic variation data and regulatory sequence motifs, we provided evidence to suggest that Low_CNE bases are under constraint in humans and enrich for having a regulatory role, though to a lesser extent than High_CNE bases. High_notCNE bases had greater enrichments for regulatory motifs and less genetic

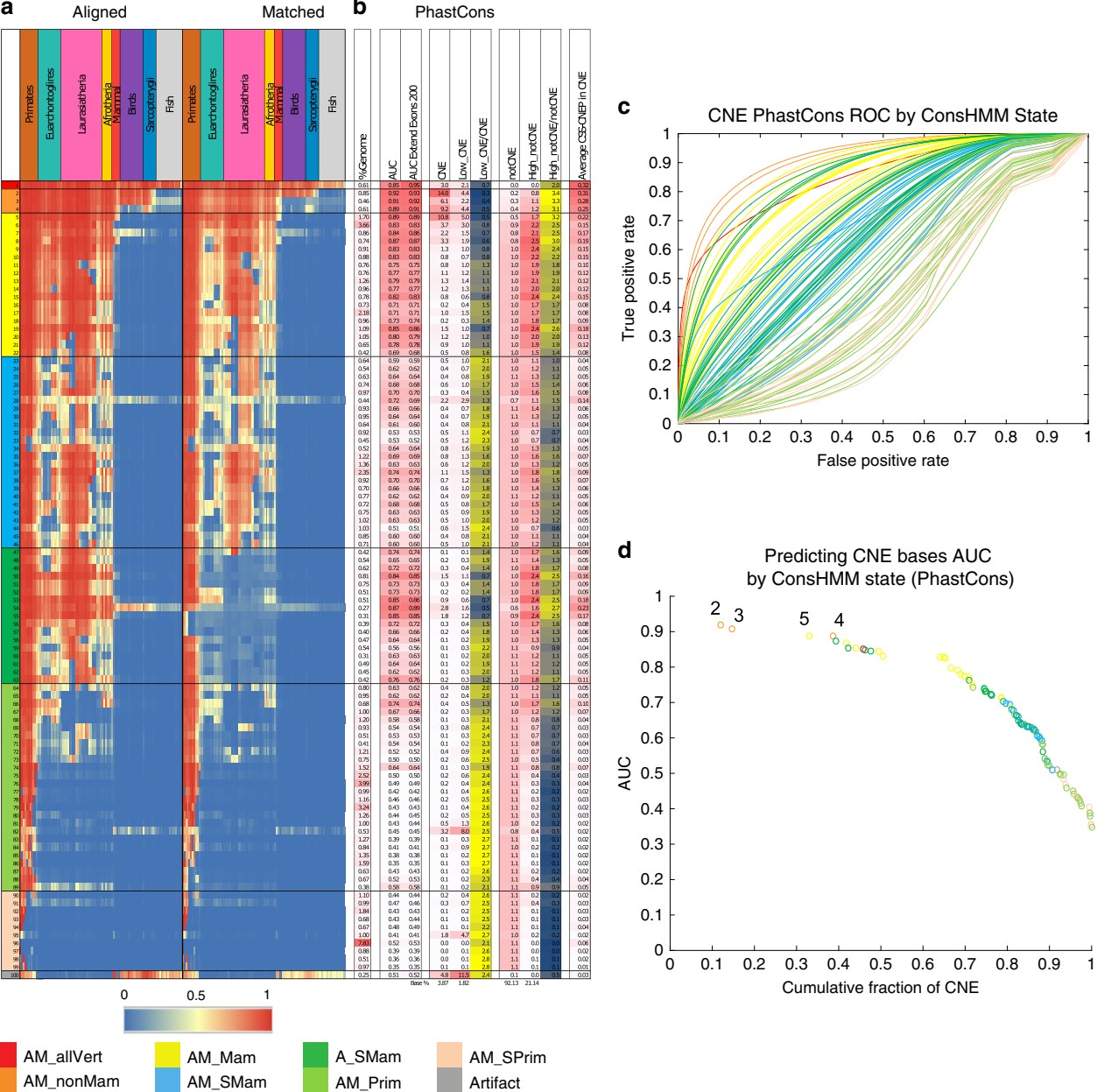

**Fig. 6 CNE prediction depends on conservation state. a** Heatmap for the ConsHMM conservation state model from ref. [32]. Rows correspond to conservation states that were previously clustered into eight groups labeled at bottom and colored accordingly (Fig. 1b). For each state, the left and right halves indicate the probability of the species of the column having a nucleotide aligning to and matching to the human reference genome, respectively. Major groups of species are colored and labeled. **b** The first column reports the genome percent of each state. The second column contains the AUC of the CNEP score for predicting CNE bases in each state (see "Methods" section), where for this and the remaining columns the constrained elements are from PhastCons. The next column reports the AUC when exons are first extended by 200 bp. The next three columns contain the fold enrichment for CNE bases, Low_CNE bases, and the ratio of the Low_CNE to CNE enrichments. The following three columns contain the fold enrichment for notCNE bases, High_notCNE bases, and the ratio of the High_notCNE to notCNE enrichments. The last column shows the average CSS-CNEP score in CNE bases in the state. Adjacent pairs of columns with a red-white color scale are on the same color scale, while other columns are on a column specific color scale. The bottom row gives the genome coverage percent of the column. Results based on all the constrained element sets is in Supplementary Fig. 22. **c** ROC curves for the CNEP score identifying PhastCons CNE bases in specific ConsHMM conservation states colored based on the coloring in **a**. **d** Plot showing the AUC values for each ROC curve shown in **c** with the same coloring. The AUC values are displayed from left to right based on decreasing values and positioned along the x-axis based on the cumulative fraction of PhastCons CNE bases that they cover. States with the highest AUC values are labeled. Similar plots, but for additional constrained element and based on excluding bases within 200 bp of exons from the positives are in Supplementary Figs. 23 and 24. Source data are provided as a Source Data file.

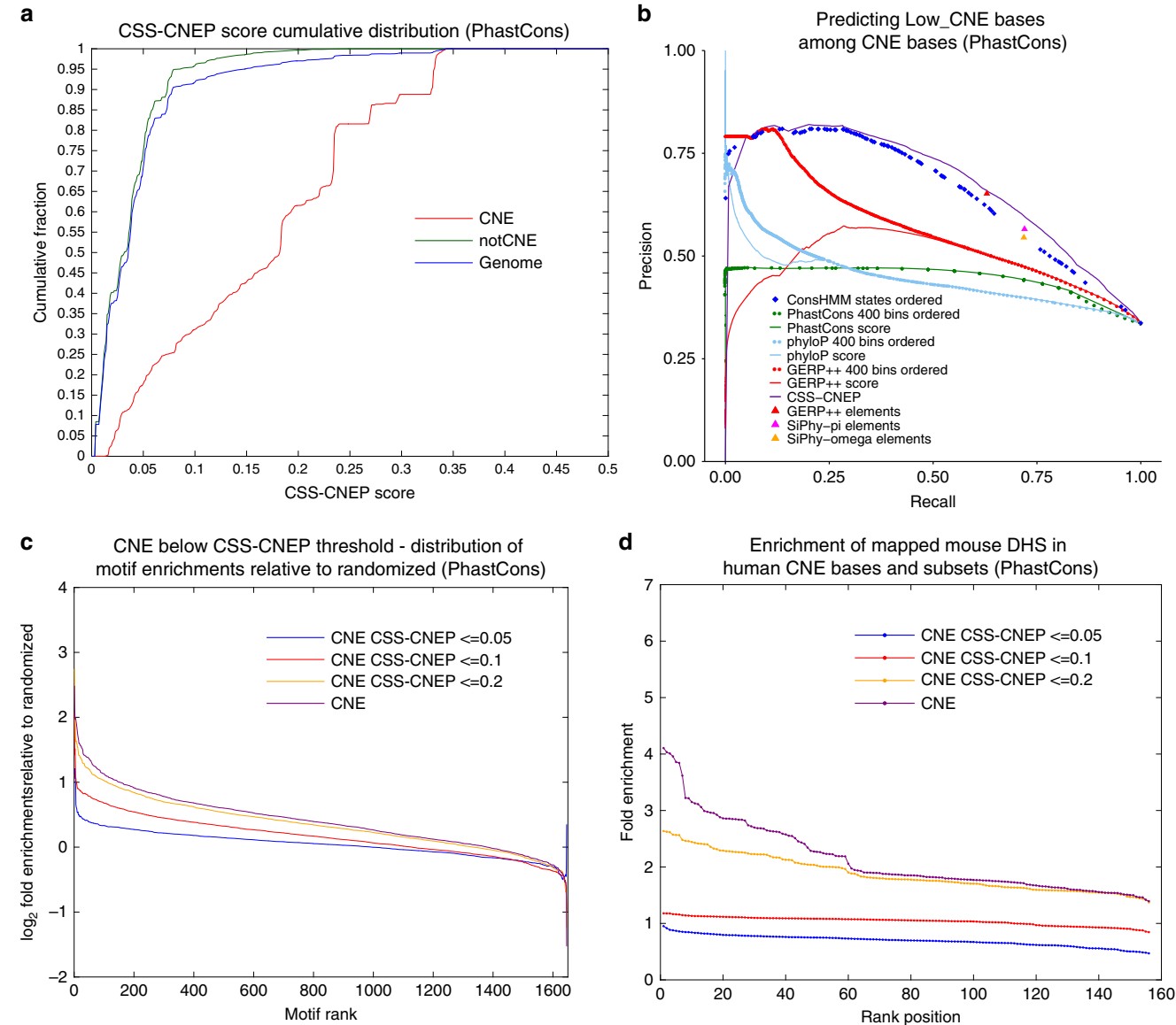

**Fig. 7 Conservation signature score by CNEP (CSS-CNEP). a** Cumulative distribution of the CSS-CNEP score genome-wide and specifically in CNE and notCNE defined by PhastCons. **b** Precision-recall analysis for predicting PhastCons Low_CNE bases among CNE bases using additional comparative genomics information. In this analysis, Low_CNE bases are positive bases and High_CNE bases are negative bases. The predictions based on the CSS-CNEP score as well as the PhastCons, PhyloP, and GERP++ constraint scores are shown based on ranking from lowest to highest value. Also shown for the PhastCons, PhyloP, and GERP++ scores are precision-recall curves, based on dividing a score into 400 bins and ordering the bins in decreasing order of enrichment for Low_CNE bases on a training set containing separate positions than used for the evaluation (see "Methods" section). The plot also shows the cumulative precision-recall of the conservation states when ordered based on decreasing enrichment for Low_CNE bases in the training data. Additionally, a single point is shown for each of the other three constrained element sets corresponding to predictions based on bases not covered by them. **c** Similar plot to Fig. 4a, but showing the difference of the distribution of motif enrichments relative to the distribution for a randomized set of the motifs for PhastCons CNE bases and the subsets that had CSS-CNEP scores ≤ 0.05, 0.10, and 0.20. **d** Similar plot to Fig. 4c, but showing enrichments for DNase I Hypersensitive Sites (DHS) from 156 experiments in mouse, for PhastCons CNE bases and the subsets that had CSS-CNEP scores ≤ .05, 0.10, and 0.20. Similar plots for additional constrained elements can be found in Supplementary Figs. 25–28. Source data are provided as a Source Data file.

variation compared to Low_notCNE bases, though less than the reduction seen for Low_CNE bases. A subset of High_notCNE bases might correspond to bases that are under evolutionary constraint in humans, but not actually in a constrained element call, which was supported by the conservation state enrichments for High_notCNE bases. Another subset of High_notCNE bases may correspond to adaptive and recently evolved bases with a potentially important regulatory role that share epigenomic marks and TF-binding patterns associated with CNE bases,

though we note that some adaptive and recently evolved bases might have distinct epigenomic mark and TF-binding patterns that CNEP is not optimized to detect. A direction for future work could be to investigate training CNEP so it is optimized to detect other types of conservation annotations.

A retrospective analysis showed additional datasets led to modest overall improvements in the CNEP predictions. Datasets with substantial additional information were limited and enriched for pluripotent stem cell, pancreas, or neural cell type classes. It is

possible different types of assays or cell types and conditions distinct from those considered here would further improve prediction of CNE bases.

For a researcher interested in a CNE base, the CNEP score provides information as to whether data in a compendium is sufficient to explain the constraint. To facilitate follow-up on a base with a high CNEP score, the CNEP software can generate a list of features that overlap the base ranked by their expected CNEP score statistic (Fig. 1b). We do not recommend trying to interpret the weights of CNEP's classifiers due to multicollinearity expected when using many features. To determine whether a subset of features are sufficient to explain the constraint, such as from the same cell type, CNEP could be run with the subset of features. If a CNE base receives a low CNEP score, two possible reasons are either a false constraint call or informative experiments were not in the input compendium, which the complementary CSS-CNEP that we developed can be used to help differentiate between. We note that CNEP and CSS-CNEP are not specifically designed to be used directly for the task of prioritizing phenotypic-associated genetic variants, and would not be expected to be competitive with scores designed for this task that use more diverse sets of features. A possible direction of future work is to evaluate incorporating the CNEP and CSS-CNEP scores with other features for the task of prioritizing variants.

While the CNEP score was a relatively effective predictor of CNE bases from epigenomic and TF binding data compared to other approaches, a portion of CNE bases remains difficult to predict from such compendiums. The functional importance of a subset of such bases is supported by orthogonal evidence, thus also highlighting the remaining challenge to a comprehensive understanding of the non-exonic genome.

## Methods

**Genome assembly and gene annotations**. All predictions and analyses were done on human genome assembly hg19 and were restricted to chr1-22 and chrX. For gene annotations we used the GENCODE v19 annotations obtained from ftp://ftp. sanger.ac.uk/pub/gencode/Gencode_human/release_19/gencode.v19.annotation. gtf.gz. Exon annotations include exon bases that are non-coding.

**Constrained element sets**. We used four different constrained element sets based on the PhastCons[5], GERP++[2], SiPhy-omega, and SiPhy-pi[3,4] methods. The PhastCons-constrained elements were based on the human hg19 100-way vertebrate alignment and obtained from the UCSC genome browser[31]. The SiPhy-omega and SiPhy-pi elements were called based on a 29-way mammalian alignment and were the hg19 version obtained from https://www.broadinstitute.org/mammals-models/29-mammals-project-supplementary-info. The GERP++ elements were called based on the mammalian subset of the UCSC genome browser hg19 46-way vertebrate alignment obtained from http://mendel.stanford.edu/SidowLab/downloads/gerp/.

**Epigenomics and TF-binding features**. We used 63,741 binary features defined from epigenomics and TF-binding data. The sources and list of the features are found in Supplementary Data 1. The features were derived from ChIP-seq data of histone modifications, TFs including general factors, DNase I hypersensitivity data, and FAIRE data. The data were from the ENCODE consortium[6,39], Roadmap Epigenomics consortium[7], the ReMap public dataset[8,29], or the ChIP-Atlas[9].

In total 58,484 features were based on peak call datasets, where each dataset corresponds to one feature. For these features, the data was encoded as a '1' if the corresponding base overlapped a peak and '0' otherwise. The peak calls for the Roadmap Epigenomics data was based on the unconsolidated datasets. For peak calls from the ENCODE consortium, we used peak calls available from the second phase of the ENCODE project[6] and peak calls available from the ENCODE portal[39]. For the ENCODE portal data we downloaded all files for the ChIP-seq or DNase-seq assays available in narrowPeak or broadPeak format for hg19 on May 11, 2018 produced by the ENCODE project from https://www.encodeproject.org/. For the ReMap database, which is a reprocessing of ChIP-seq data of TFs from the Gene Expression Omnibus and ArrayExpress, we used the peaks in the hg19 files restricted to the 'Public' data (non-ENCODE) and had features based on both the 2015 and 2018 versions of the database. For the ChIP-Atlas, which is a reprocessing of Sequence Read Archive data, we used all peaks called at the $10^{-5}$ threshold for hg19 available from http://dbarchive.biosciencedbc.jp/kyushu-u/hg19/eachData/bed05/ on May 11, 2018. We note that some of the ChIP-Atlas datasets were generated by the ENCODE or Roadmap Epigenomics project, but processed differently.

We also had 5215 features defined based on chromatin state calls from three different ChromHMM models[38]. The three models were: (1) a 15-state model defined across 9-ENCODE cell types based on eight histone modifications and CTCF; (2) the 15-state 'core' model based on 5-histone modifications defined across 127-reference epigenomes based on consolidated data processed by the Roadmap Epigenomics consortium (111 reference epigenomes were derived from data produced by the Roadmap Epigenomics project and 16 from the ENCODE project); (3) a 25-state model based on imputed data for 12-chromatin marks (10 histone modifications, H2A.Z, and DNase I hypersensitivity) defined across the same 127 reference epigenomes[40]. For each model, we had a separate feature for each chromatin state and cell type or reference epigenome combination. A feature value was encoded as a '1' if a base overlapped the chromatin state in the cell type or reference epigenome and a '0' otherwise. We observed a small increase in predictive performance of CNEP when including the ChromHMM features in addition to the peak features (Supplementary Table 1).

Additionally, we had 42 features defined based on the position of Digital Genomics Footprints[7,41] each corresponding to an experiment in a different cell or tissue types. For these features the data was encoded as a '1' for those bases overlapping a footprint and a '0' otherwise.

**CNEP method**. The CNEP scores are generated by first training an ensemble of logistic regression classifiers. CNEP jointly considers all 63,741 features described above. For a given constrained element set and a set of binary epigenomics and TF-binding features, CNEP trains logistic regression classifiers to discriminate between bases in a constrained element that are outside of all exons as a positive set from all other bases as a negative set. We note that bases in constrained elements within exons, which corresponds to 1.3–1.7% of bases outside of non-exonic constrained elements, were included in the negative set. The specific value in the range depends on the specific constrained element set. We included such bases in the negative set instead of excluding them from the training, since CNEP predictions were still made in exons, and this way high prediction values within exons can be more confidently interpreted as being associated with the exon also overlapping epigenomic and TF binding typical of non-exonic constrained elements (Supplementary Fig. 2).

For generating CNEP scores on one chromosome based on one constrained element set, CNEP-trained 10 logistic regression classifiers using 10 different sets of 1,000,000 randomly sampled positions from the other 22 chromosomes. The overlap of positions of any one set of positions with another set was on average 0.034% of positions. We repeated this for each of the constrained element sets and 23 chromosomes thus training in total 920 logistic regression classifiers in parallel. Across all classifiers, the number of positive examples per classifier was on average 4.1% and varied between 2.9% and 5.5%, and corresponded approximately to the genome-wide percent of CNE base for the constrained element set (Supplementary Fig. 2e). We verified that using 10 classifiers per constrained element set and chromosome for predictions led to improved predictive performance compared to a single classifier (Supplementary Table 1). We also verified that there was similar predictive performance with five classifiers as with 10 classifiers (Supplementary Table 1), thus increasing beyond 10 classifiers would be expected to give diminishing returns, while increasing computational costs. For these analyses, we computed AUC values based on using the same 10 classifiers, but the classifiers were split into either two groups of 5 or 10 groups of one. We then averaged the AUC values based on each group. Additionally, we observed that the average pairwise correlation between predictions based on using a single classifier of the 10 per element set was 0.73, but when the 10 classifiers were split into two halves, the correlation of predictions based on five classifiers per element set increased to 0.93.

CNEP used the Liblinear v.2.1[42] software to train the logistic regression classifiers using $L_1$ regularization (−s 6) with a bias term (−B 1), with the default regularization parameter value of 1 (−c 1). With this regularization on average 59.8% of the features per classifier had a non-zero coefficient, and 80.4% of features had a non-zero coefficient in at least one classifier. The exclusive use of binary features allowed us to make efficient use of the sparse representation of the data in Liblinear.

For generating genome-wide predictions based on one constrained element for each chromosome, CNEP computed and averaged the probabilistic predictions, between 0 and 1, from its 10 corresponding logistic regression classifiers and then outputted the predictions to the nearest 0.001 value. To generate the CNEP score we then averaged the outputted predictions based on each of the four constrained element sets.

**Computing observed and expected CNEP score statistics**. For computing the observed average CNEP score statistic for a feature, we computed the average CNEP score in all bases in the genome where the feature was defined as being present. For computing the expected average CNEP score statistic for a feature, we computed the average over the four constrained element sets of the number of bases for which the feature was present and overlapped a CNE element divided by the total number of bases in which the feature was present. We computed the genome-wide observed and expected CNEP score statistics the same way except all bases in the genome were included.

**CNEP score's relationship to Roadmap Epigenomics DNase data**. For computing the relationship between the CNEP score and the average fraction of Roadmap Epigenomics DNase I hypersensitivity experiments a base is in a peak (Fig. 2c), we used 350 narrowPeak call files with 'ChromatinAccessibility' in the file name available from http://egg2.wustl.edu/roadmap/data/byFileType/peaks/unconsolidated/narrowPeak/. For each value of the CNEP score computed to the nearest 0.001 and covering at least 1000 bases, we took all bases in the genome having that score and determined the average fraction of the 350 experiments in which the bases are overlapped by a peak call.

**Baseline and related score comparisons**. For comparing the predictive performance of the CNEP score, we computed the baseline score based on Roadmap Epigenomics DNase I hypersensitivity data as the number of times a base was overlapped by a peak in the set of 350 experiments described above. For the all DNase I baseline evaluation, we expanded the counts to be based on a set of 4522 DNase I hypersensitivity experiments. We also generated a baseline based on the number of times any of the 63,741 features overlap a base. We also compared to each of those features individually. In addition we compared to two other existing related scores, the Segway Encyclopedia 'Position-wise aggregated conservation-associated activity scores'[24], that we obtained from https://noble.gs.washington.edu/proj/encyclopedia/caas.bed.gz and the FitCons2 cell-type integrated score[22] that we obtained from http://compgen.cshl.edu/fitCons2/hg19/H1/E999-sco.bw. We conducted these comparisons genome-wide. Since CNEP predictions were made for each chromosome by leaving that chromosome out when training, there was no need to create additional training and test sets for these comparisons.

For comparing to the version of the CNEP score that used a random forest in place of a logistic regression classifier, we used the RandomForestClassifier from scikit-learn[43]. We used the same procedure for training and prediction as with CNEP when using logistic regression classifiers, but generated predictions for only a million randomly sampled positions on chr10. We also evaluated CNEP predictions using logistic regression classifiers for those same million randomly sampled positions.

For evaluating DeepSea sequence predictions[25], we obtained predictions for 919 chromatin features available from the DeepSea web-server (http://deepsea.princeton.edu/job/analysis/create/) for 50,000 randomly sampled positions in the genome. The maximum and average AUC over the 919 features was computed.

**Defining sets of bases for analyses**. For each of the four constrained element sets considered, we defined the following six sets of bases: (1) CNE—bases in a constrained element that do not overlap a GENCODE exon; (2) Low_CNE—bases in a constrained element that do not overlap a GENCODE exon and have a CNEP score less than or equal to the CNEP mean, 0.0419; (3) High_CNE—bases in a constrained element that do not overlap a GENCODE exon and have a CNEP score >0.0419; (4) notCNE—bases not in a constrained element and do not overlap a GENCODE exon; (5) Low_notCNE—bases not in a constrained element and do not overlap a GENCODE exon and have a CNEP score ≤0.0419; (6) High_notCNE—bases not in a constrained element and do not overlap a GENCODE exon and have a CNEP score >0.0419 (Supplementary Table 2).

**Analysis of CNEP score's relationship to chromatin states**. For the analysis of the relationship between the CNEP score and chromatin state annotations (Fig. 2d), we used the 25-state ChromHMM chromatin state annotations defined across 127 epigenomes based on imputed data for 12-chromatin marks[40]. For each value of the CNEP score, which were computed to the nearest 0.001, we took all bases in the genome having that value. We then determined for those bases and each of the 25-states on average what fraction of the 127 epigenomes were assigned to that state. We then stacked bar graphs with the fractions starting from the state with the greatest state number (25_Quies) to the lowest state number (1_TssA) with the state numbers and colors from ref. [40]. In the plot, we did not differentiate between different states that were previously given the same color and thus the plot provides information on the 14-state groups that were colored differently. We also computed the fold enrichments of CNE, Low_CNE, High_CNE, notCNE, Low_notCNE, and High_notCNE bases for each of the four constrained element sets using the OverlapEnrichment command of ChromHMM v1.13[38] for each state in each of the 127 epigenomes, and then reported the median enrichments (Supplementary Fig. 10).

**Positional-enrichment analysis**. The enrichment of Low_CNE relative to CNE bases at a specific distance to the nearest exon was defined as the ratio of the fraction of Low_CNE bases whose nearest exon base was at that distance to the fraction of all CNE bases in the genome whose nearest exon base was at that distance. Enrichments for High_CNE bases relative to CNE bases were similarly defined as were the enrichments of High_notCNE and Low_notCNE bases relative to notCNE bases. A similar set of enrichments were computed for nearest TSS and genes. For genes, a base within the boundaries of the transcription start and end site had a distance of zero. Similar enrichments for nearest CNE bases were also computed for High_notCNE and Low_notCNE bases relative to notCNE bases.

**Repeat-enrichment analysis**. We computed the fold enrichment for CNE, Low_CNE, High_CNE, notCNE, Low_notCNE, and High_notCNE bases for overlapping a base in a repeat element called by RepeatMasker[44]. We also computed enrichments restricted to RepeatMasker classes and families that covered at least a million bases of the genome. Fold enrichments were defined as the ratio of the fraction of bases in the target set overlapping the RepeatMasker called bases to the fraction of bases in the genome overlapping the RepeatMasker called bases. The RepeatMasker calls were those we previously obtained from the UCSC genome browser RepeatMasker track[31,32,44].

**Conservation state analysis**. The ConsHMM conservation state annotations were the 100-conservation state annotations for hg19 from ref. [32]. These conservation state annotations were defined on the same 100-way vertebrate alignment for which the PhastCons-constrained elements we used were defined. Fold enrichments for CNE, Low_CNE, notCNE, and High_notCNE bases in the conservation states were computed using the OverlapEnrichment command of ChromHMM v1.13 with the options '-b 1 -lowmem' specified. Conservation state assignments to chrY were excluded from the background in this analysis, as the CNEP scores were not defined on this chromosome. The per state ROC and AUC values for the CNEP score were computed by considering a positive base a CNE in a specific conservation state and a negative base any base in the genome that was not in a CNE. Bases in the CNE set that were in a different conservation state were excluded when generating the ROC and computing the AUC. The ROC and AUC based on extending exons 200 bp in each direction was computed in the same way, except first adjusting the exon start and end positions.

**CSS by CNEP (CSS-CNEP)**. To compute the CSS by CNEP (CSS-CNEP), for each chromosome we first computed the average CNEP score for each of the 1600 combinations of the 100 conservation states and four constrained elements within non-exonic bases on all other chromosomes. We then for each base on the target chromosome, including exons, used the average corresponding to the combination of conservation state and constrained element sets overlapping the base to define the CSS-CNEP value for the base. For over 99.98% of base predictions, the combination average used was derived from more than 1000 bases.

**Evaluation of predictions of Low_CNE bases among CNE bases**. We evaluated the ability of CSS-CNEP and other comparative genomic scores and annotations to predict among CNE bases those that received a low CNEP score. Specifically for this evaluation, we split the CNE bases into two sets, the Low_CNE and High_CNE bases, and treated the Low_CNE bases as positives and High_CNE bases as negatives. We followed a previous evaluation approach where we randomly split 200 kb genome-segments into two halves, one used for training and the other used for testing[32]. We computed the precision-recall for ConsHMM states by first based on the training data ordering the ConsHMM states in decreasing order of enrichment for Low_CNE bases with a background of all CNE bases. We then used that order to evaluate the cumulative precision-recall on the test data of predicting CNE bases from ConsHMM annotations. We formed predictions for the CSS-CNEP, PhyloP[45], GERP++[2], and PhastCons[5] scores by ordering bases from lowest to highest value according to the score. For the PhyloP, GERP++, and PhastCons scores we also followed the procedure of ref. [32] and formed 400 equal-sized bins of the score and then repeated the procedure used for the ConsHMM conservation states treating a bin as if it was a state. PhyloP and PhastCons scores were based on the same alignment as used for calling ConsHMM states and PhastCons constrained elements. GERP++ scores were based on the same alignment as used for calling GERP++ constrained elements. For each constrained element set other than the one used to define the CNE bases in the evaluation, we computed the precision and recall for a prediction based on bases not overlapping the constrained element set.

**Human variation analysis**. The human variation analysis was conducted on a set of 105 unrelated individuals from the YRI population part of the 1000 Genomes Project phase 3[33]. We focused on this population for analyzing the effects of selection, since it is associated with greater genetic diversity and has a simpler demographic history than non-African populations[46]. We selected high-quality sites by applying a mask from the 1000 Genomes Project where a site was defined as high quality if its depth (DP) is within 1.5× the mean DP across all sites[33]. For this analysis, we restricted it to the autosomes and variant calls that were bi-allelic. For each set of coordinates analyzed, we computed a count $c_n$ of how many of the variants occurred in exactly $n$ individuals for each value of $n = 1,...,10$ (low and intermediate frequency variants) and also a count $c_{>10}$ of how many occurred in >10 individuals (common variants). We then computed the proportional SFS as each of these individual counts divided by the sum of all of the counts. We assessed the statistical significance between pairs of coordinate sets by applying a chi-square test to the 11 count values.

The absolute SFS contains the numbers (rather than proportions) of SNPs at particular minor allele counts. Because the count of SNPs is affected by the number of base pairs analyzed (more base pairs would lead to more SNPs) as well the mutation rate (higher mutation rates lead to more SNPs), we normalized for both of these factors. To do this, first we obtained mutation rate estimates from

http://mutation.sph.umich.edu/hg19/[34]. We associated each base with a single mutation rate by averaging its three mutation rates, each corresponding to a mutation from the reference nucleotide to an alternative nucleotide. For a coordinate set, we computed the sum of the mutation rates at all bases that were high-quality sites as defined above and had a mutation rate available. This sum is equivalent to the number of base pairs analyzed in a coordinate set times their average mutation rate. We computed the unnormalized count values as described above for the proportional SFS except excluded positions that did not have mutation rates available. We then divided these counts by the sum of the mutation rates.

To compute SFS controlled for background selection we used the version of $B$-values in hg19 as part of the CADD annotation set[12], which are based on the $B$-values from ref. [35]. For the proportional SFS, we reweighted variant calls in each coordinate set so that the $B$-value distribution was effectively the same as the distribution of $B$-values at all non-exonic bases with a variant call. This analysis was restricted to non-exonic variants that had an estimated $B$-value available. The weighting for a variant with a $B$-value, $x$, was $p_a(x)/p_s(x)$ where $p_a(x)$ and $p_s(x)$ are the proportions of variants with the $B$-value $x$ among all variants considered and the subset in the coordinate set, respectively. For the absolute SFS density normalized by its average mutation rate, we reweighted bases in each coordinate set so the $B$-value distribution was effectively the same as the distribution of $B$-values at all non-exonic bases. This analysis was restricted to non-exonic bases that were in a high-quality site and had both estimated mutation rates and $B$-values available. The weighting for a base with a $B$-value, $x$, was $p_c(x)/p_s(x)$ where $p_c(x)$ and $p_s(x)$ are the proportions of variants with the $B$-value $x$ among all bases considered and the subset in the coordinate set, respectively. The weighting was used in both counting variants and the sum of the mutation rates.

**Motif-enrichment analysis**. For the motif enrichment analysis (Fig. 4a, b, Supplementary Figs. 15–17, Supplementary Data 3), we used motif instances from http://compbio.mit.edu/encode-motifs/matches-with-controls.txt.gz[28]. We used motif instances from a set of 1646 motifs that excluded motifs that were in the compendium based on being discovered from ENCODE ChIP-seq data, so that the set of motifs we analyzed were independent of the features provided to CNEP. The motif instances were called outside of coding, 3′ UTR, and repetitive regions and called independent of conservation. For each motif, there were also a set of corresponding control motif instances called[28], which control for biases from sequence composition or background. To compute the enrichment of a specific motif in a target set of bases, we computed the ratio of the fraction of motif instance bases that also overlapped the target set to the fraction of corresponding control motif instance bases that also overlapped the target set. For each of the four constrained element sets, these enrichments for individual motifs were reported for High_CNE and Low_CNE bases (Fig. 4b, Supplementary Fig. 17, Supplementary Data 3). For the analyses of the distribution of motif enrichments for a target set, we generated three randomized versions of the motif instance calls with controls. To generate a randomized version for each chromosome we performed column-wise random permutations where one column is the motif identifier and the other column contains the motif coordinates. For each target set considered, we computed the distribution of motif enrichments for each of the randomized motif instances, using the same procedure as the actual motif instances. We then ordered the enrichments separately for the actual and three randomized datasets. At each ranked position in the ordering, we took the difference between the $\log_2$ value of the actual enrichment and the $\log_2$ value of the median enrichment from the three randomized datasets. We also followed the above procedures for subsets of CNE bases with CSS-CNEP values at or below specific thresholds (Fig. 7c, Supplementary Fig. 26).

**Motif set GO analysis**. We conducted a GO-enrichment analysis for the TFs corresponding to three subsets of motifs that had at least a $\log_2$ fold enrichment of 0.5 in High_CNE or Low_CNE bases. The three subsets were: (i) 'High_CNE strongly preferred'—those motifs for which the difference in the $\log_2$ fold enrichment for High_CNE and Low_CNE bases was >0.75; (ii) 'High_CNE moderately preferred'—the difference was between 0.75 and 0; (iii) 'Low_CNE preferred'—which had a greater enrichment for Low_CNE bases than High_CNE bases. GO enrichment was conducted using the STEM software v.1.3.11 with default settings (Supplementary Data 4)[47]. STEM uses the hypergeometric distribution to compute $p$-values for a one-sided test. The GO and human gene annotations were downloaded using the STEM software on September 17, 2017. We used as a base set all TFs corresponding to a motif in the compendium. The corresponding TF for a motif was taken to be the portion before the '_' in the motif ID.

**Mouse DNase I hypersensitive site enrichment analysis**. For the mouse DNase I hypersensitivity site (DHS) analysis (Fig. 4c, d, Supplementary Figs. 18 and 19, Supplementary Data 5), we used the 156 narrowPeak files from the University of Washington Mouse ENCODE group available from http://hgdownload.soe.ucsc.edu/goldenPath/mm9/encodeDCC/wgEncodeUwDnase/ and http://hgdownload.soe.ucsc.edu/goldenPath/mm9/encodeDCC/wgEncodeUwDgf/[36,37]. We also generated a randomized version of each set of DHS by randomly selecting a different position for each DHS in the original file on the same chromosome. We lifted over both the real and randomized versions of the DHS files from mm9 to hg19 using

the liftOver tool from the UCSC genome browser with the options '-bedPlus=3 -minMatch = 0.00000001'. The lower value for the minMatch parameter enables a more permissive mapping of peaks from mouse to human and thus an enrichment estimate that is more reflective of a background that includes all mouse DHS. For both the real and randomized version of each set of DHS, for each of the four constrained element sets considered, we computed enrichments for the six target sets: CNE, High_CNE, Low_CNE, notCNE, High_notCNE, and Low_notCNE. Enrichments for each set of DHS were computed by taking the ratio between the fraction of bases in human covered by a DHS that are in the target set to the fraction of bases in the genome that are in the target set. The reported enrichment for an experiment is the ratio of this enrichment for the real DHS compared to the corresponding enrichment for the randomized DHS. We also followed the above procedures for computing enrichments for subsets of CNE bases with CSS-CNEP values at or below specific thresholds (Fig. 7d, Supplementary Fig. 27).

**Retrospective analysis on information in additional datasets**. For the retrospective analysis of additional information in human datasets, we held out from training all features from ChIP-Atlas[9], ReMap 2018[8], the ENCODE portal[6,39]. The version of the CNEP score used for this analysis was generated with data that was available and accessible by 2015. Specifically we used a 10,836 feature subset, which included data from the ENCODE consortium during its second phase[6], Roadmap Epigenomics consortium[7], and the ReMap 2015 public dataset[29] (Supplementary Data 1). We did not exclude any dataset from the analysis for being based on the same experiment as used to generate the CNEP predictions. We excluded files that did not have any peaks called on the chromosomes we considered. For the analysis with updated GENCODE exon annotations we used release 28 mapped to hg19/GRCh37 available from ftp://ftp.ebi.ac.uk/pub/databases/gencode/Gencode_human/release_28/GRCh37_mapping/gencode.v28lift37.annotation.gtf We excluded any base in an exon from release 19. For generating the shuffled data we used the shuffleBed command of BEDTools (v.2.17)[48].

The CNEP underestimation value for a dataset was defined as the difference of its expected average CNEP score and its observed average CNEP score statistic, where the observed CNEP score is computed based on the subset of features available in 2015 as described above. The CNEP underestimation value of a dataset reflects its marginal additional information for annotating CNE bases conditioned on the 2015 feature subset within bases the dataset covers. This value was analyzed relative to the number of bases the dataset covers, where coverage of a dataset is the number of bases for which the feature for the dataset is encoded as a '1'. A property of a logistic regression classifier is that the sum of a feature's values for the positive instances equals the weighted sum of the feature's values weighted by the prediction probabilities for the positive class on the training data without regularization. This leads to the expectation that including a dataset as a feature to the classifier would cause its CNEP underestimation to be close to zero. We verified this holds approximately for those features used to generate the CNEP score for its actual predictions (Supplementary Fig. 20g, h).

We note that a dataset with a CNEP underestimation of zero can still be informative on constraint in isolation, but here we are evaluating the marginal additional information conditioned on the 2015 subset. Higher positive values correspond to datasets that have greater overlap with CNE bases than expected based on the CNEP score derived from information in the 2015 subset, while more negative values correspond to fewer overlaps and an overestimate of the CNEP score for the dataset. We primarily focused on positive CNEP underestimation scores since this can highlight datasets that annotate CNE bases that previously had few if any informative annotations. Additionally, we observed more datasets that had relatively large CNEP underestimation values than overestimate values (Supplementary Fig. 20).

The CNEP underestimation values has two advantages compared to directly analyzing the correlations of CNEP predictions with and without each additional feature. One advantage is it allows identifying datasets that might provide a highly informative annotation, but for only a relatively small number of CNE bases. The other advantage is that it is practical to compute for a large number of features since it does not require retraining the CNEP score for each additional feature.

For computing the cell type class enrichments, we used STEM software v.1.3.11 with user provided annotations treating each dataset as if it was a gene and the cell type class as the annotation category[47]. The foreground for the enrichment were those ChIP-Atlas datasets covering at least 200 kb and having an underestimation value >0.02. The background set for the enrichment analysis was all ChIP-Atlas datasets that covered at least 200 kb. We used default settings except changed the minimum number of genes parameter to 1 and multiple hypothesis testing correction to 'Bonferroni'. STEM computed $p$-values for a one-sided test using the hypergeometric distribution.

**Overlap of constrained element sets with prioritized variants**. We obtained a curation of annotations of top 1% prioritized non-coding bases from 14-different scores used for prioritizing genetic variants from ref. [32]. Separately for each combination of score and four constrained element sets, we computed the percentage of bases prioritized by the score that overlap a constrained element when restricting to non-exonic bases in the genome. We also computed the corresponding fold enrichment by dividing that percent overlap by the percentage of the non-exonic genome the constrained element set covers.

**Reporting summary**. Further information on research design is available in the Nature Research Reporting Summary linked to this article.

## Data availability

The CNEP score and the CSS-CNEP are available from https://github.com/ernstlab/CNEP. All input data used to generate the scores are publicly available. Identifiers for the features are provided in the Supplementary Data 1. SiPhy-pi and SiPhy-omega constrained elements were from https://www.broadinstitute.org/mammals-models/29-mammals-project-supplementary-info. GERP++ constrained elements and scores were from http://mendel.stanford.edu/SidowLab/downloads/gerp/. PhastCons constrained elements and scores, PhyloP scores, and RepeatMasker annotations were from the UCSC genome browser (https://genome.ucsc.edu/). The main exon annotations were from ftp://ftp.sanger.ac.uk/pub/gencode/Gencode_human/release_19/gencode.v19.annotation.gtf.gz and exon annotations from the retrospective analyses were from ftp://ftp.ebi.ac.uk/pub/databases/gencode/Gencode_human/release_28/GRCh37_mapping/gencode.v28lift37.annotation.gtf. ChIP-Atlas data were from http://dbarchive.biosciencedbc.jp/kyushu-u/hg19/eachData/bed05. ReMap data were from http://remap.univ-amu.fr/. Roadmap Epigenomics data were from http://compbio.mit.edu. ENCODE consortium data were from http://hgdownload.cse.ucsc.edu/goldenPath/hg19/encodeDCC and http://www.broadinstitute.org/~anshul/projects/encode/rawdata/peaks_histone/mar2012/narrow/combrep_and_ppr/ and for the ENCODE portal portion from https://www.encodeproject.org/. 1000 Genomes Phase 3 data is available at ftp://ftp.1000genomes.ebi.ac.uk/vol1/ftp/release/20130502/. Regulatory motif data is from http://compbio.mit.edu/encode-motifs/matches-with-controls.txt.gz. Mouse ENCODE data were from http://hgdownload.soe.ucsc.edu/goldenPath/mm9/encodeDCC/wgEncodeUwDnase/ and http://hgdownload.soe.ucsc.edu/goldenPath/mm9/encodeDCC/wgEncodeUwDgf/. Mutation rate estimates were from http://mutation.sph.umich.edu/hg19/. B-values were from http://krishna.gs.washington.edu/download/CADD/v1.4/GRCh37/annotationsGRCh37.tar.gz. Conservation state annotations and bases prioritized by various variant prioritization scores were from https://github.com/ernstlab/ConsHMM/. Segway Encyclopedia and FitCons2 scores used were from https://noble.gs.washington.edu/proj/encyclopedia/caas.bed.gz and http://compgen.cshl.edu/fitCons2/hg19/H1/E999-sco.bw, respectively. All other relevant data supporting the key findings of this study are available within the article and its Supplementary Information files or from the corresponding author upon reasonable request. Source data are provided with this paper. A reporting summary for this Article is available as a Supplementary Information file. Source data are provided with this paper.

## Code availability

The CNEP software (v1.0) is available at https://github.com/ernstlab/CNEP. Liblinear v.2.1, which is used by CNEP is available https://www.csie.ntu.edu.tw/~cjlin/liblinear/. The STEM software (v1.3.11) used for GO enrichment analysis is available at http://sb.cs.cmu.edu/stem/. The ChromHMM software (v1.13) used for computing state enrichments is available from https://ernstlab.biolchem.ucla.edu/ChromHMM/. Scikit-learn (v.0.19) used for the Random Forest comparison is available at https://scikit-learn.org/. BEDTools (v.2.17) used for shuffling bed files is available at https://github.com/arq5x/bedtools2. DeepSea was run through its web interface available at http://deepsea.princeton.edu/job/analysis/create/.

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

## Acknowledgements

We thank Xiaorui Fan, Petko Fiziev, and Bernard Kim for useful discussions or assistance. We acknowledge funding from US National Institutes of Health grants DP1DA044371, R01ES024995, U01HG007912 and U01MH105578 (J.E.), T32CA201160 (A.A.), and R35GM119856 (K.E.L.), US National Science Foundation CAREER Award #1254200 (J.E.), a Kure-IT award, an Alfred P. Sloan Fellowship (J.E.). We acknowledge the ENCODE and Roadmap Epigenomics consortia for generating some of the data used and making it available pre-publication.

## Author contributions

J.E. conceived the project. O.G., Y.L., and J.E. contributed to the design, implementation, and analysis of different versions of the methods and predictions. T.N.P., S.B.K, K.E.L. contributed to the human genetic variation analyses. A.A. contributed to the conservation state analyses. All authors contributed to the drafting or to the editing of the manuscript.

## Competing interests

The authors declare no competing interests.
