## [Peer Review File · Nature Communications]

Reviewers' comments:

Reviewer #1 (Remarks to the Author):

Summary: Interpreting the consequences of non-coding variants remains a difficult challenge. Here, Grujic et al present the Constrained Non-Exonic Predictor (CNEP), a computational method that generates a score reflecting the probability that a non-exonic base occurs in an evolutionarily constrained non-exonic element using available information about epigenomic marks and transcription factor binding sites. They compare their tool to other approaches that prioritize non-coding variation. To investigate CNEs with low CNEP scores, the authors introduce CSS-CNEP, a tool that further integrates information about conservation derived from their ConsHMM tool. The SS-CNEP tool suggests that CNEs with low CNEP scores are significantly biased toward particular conservation states. They perform an analysis of high and low CNE/non-CNE sequences and find enrichment and depletion for different categories of polymorphic variation and functional motif suggesting that sequences in these categories are subjected to different selective pressures. Finally, the evaluate feature sets that add the most information to the CNEP score which provides insight as to what types of experiments may be most informative of CNEs.

Overall, this is an interesting paper that uses machine learning to generate new insights into the intersection between constrained regulatory elements, conservation and genetic variation, and reveals places where we experiments may be most effective to increase our understanding. Although the authors suggest at times that their tool could be useful for prioritizing non-coding variants, the work in the manuscript does not explicitly address this. Instead, a more focused and deeper analysis of evolutionary constraints and study biases related to constrained noncoding elements might make a more exciting story. The authors also need to make sure they place their work into context w.r.t. similar machine learning based tools and analyses. There is some mention in the introduction but it is not comprehensive and is needs to be revised for clarity. The description of the CNEP method itself is a bit complicated and while the results suggest that the approach is working, clarification of choices made with respect to classifier design would be helpful.

Major Comments:

The authors use various features to train CNEP, which comprises an ensemble of logistic regression classifiers. Aspects of this process should be clarified:

Features: The number of features is large and it is not clear, for example, how peak calls map to individual features. Is each dataset considered its own feature, even if they ChIPed for the same factor? Or are they considered different features only if they were performed in different cell lines? It is expected that a number of the features are likely to be highly correlated with one another. Did the authors evaluate correlation among features? Such correlation would seem particularly important when using regression models for the underlying classifiers, as highly correlated features could lead to multicollinearities if they are considered together (although it was unclear from the methods whether multivariable regression was used, or whether a different set of classifiers were used for each individual feature). Also, the authors should comment on the relationship between ChIPAtlas/ENCODE/Roadmap features and ChromHMM features since ChromHMM ingests chromatin mark annotations predict higher order chromatin states and is thus a derivative of those features. Are both features types required to obtain the optimal performance? How does performance change when one set of features is omitted?

Labels: Labels are assigned based on different constrained element sets derived from other methods. How correlated are the predictions of the underlying methods? Are the constrained element sets highly redundant or highly complementary? Some statistics are mentioned, but it is not clear what is driving the correlation. Is the agreement more on the basis of presence of constrained elements or absence of constrained elements?

Classifier: The classifier comprises 10 logistic regression models per chromosome per constrained element set, each trained on a different set of random samples. How was the number 10 determined? Among these 10, how correlated are the predictions? In other words, would the authors obtain different results if they trained 1 logistic regression classifier per chromosome and constrained element set? How many features are evaluated by each classifier – all features every time? Many ensemble learners randomly sample the feature space to reduce correlation between the individual classifiers. The choice of logistic regression classifiers assumes a linear relationship between the feature space and the outcome variable. Did the authors evaluate any nonlinear alternatives?

Training set: Classifiers are trained based on 1 million randomly sampled positions from each chromosome. How often do randomly selected training examples overlap – how much overlap between training sets for individual classifiers? What is the balance between positive (is a constrained element) and negative (is not a constrained element) training examples? If there is a large class imbalance, did the authors try using e.g. Firth's penalized regression? This may be unnecessary, however without knowing the degree of imbalance, it is hard to say. Why does the CNEP negative set include exons? This asserts that regulatory elements will never overlap an exon which may or may not be true. Why not omit exons from the training set altogether?

CNEP score: Please clarify whether there is a CNEP score for each feature. The methods text includes the following: "For computing the actual observed average CNEP score for a feature, we computed the average CNEP score in all bases in the genome where the feature was defined as being present." This implies that there is a separate score for each feature.

CNEP-CSS score: The average CNEP score across a set of bases with the same conservation level is taken, with 100 different conservation states considered as annotated by ConSHMM. Does it make sense to consider all 100 states? How are constrained elements distributed across the 100 ConSHMM categories? How many bases are associated with each of the 100 ConSHMM categories across the genome?

An overview figure describing what information is used and how would be helpful to understand the study design.

The analysis of CNE/non-CNE with respect to conservation is interesting and could be expanded. Were there any spatial characteristics distinguishing low and high CNE/non-CNE bases? Did they tend to be more proximal/distant relative to genes or ncRNAs? Was there any relationship to transposable elements? Was there any bias for promoter or enhancer elements to be associated with low versus high CNE?

In the discussion, the authors state "The CNEP score is designed to allow a researcher interested in gaining additional insights into a variant prioritized based on being in a constrained base to determine whether there are annotations within a compendium of epigenomic and TF binding data to explain the constraint." If the authors wish to claim that their tool is useful for variant prioritization, they should consider comparing its performance to other published methods (CADD, DANN, DeepSea, etc) benchmark sets of clinically or biologically relevant non-coding variants (GWAS variants, eQTLs, etc). See "Rentzsch P, Witten D, Cooper GM, Shendure J, Kircher M. CADD: predicting the deleteriousness of variants throughout the human genome. *Nucleic Acids Res.* 2018 Oct 29. doi: 10.1093/nar/gky1016. PubMed PMID: 30371827." for an example. If they simply wish to portray it as annotating variants according to overlap with likely constrained non-coding elements, they should still be sure to compare to the most similar methods (there is some justification for the choice of methods in the introduction, but this needs to be clearer, and it seems to omit some similar methods, for example DeepSea: "Zhou, Jian, and Olga G. Troyanskaya. "Predicting effects of noncoding variants with deep learning-based sequence model." *Nature methods* 12.10 (2015): 931-934."). Also, it was unclear in the manuscript whether CNEP and CSS-CNEP annotate a variant site according to the specific features associated with a variant

site being designated a CNE (i.e. does the variant affect a specific transcription factor binding site, or a ChIP peak for a specific target?). Can the authors show an application of the tool to annotate a set of variants?

Minor Comments:

Figure 1 is crowded, the labels are sometimes masked or cutoff and many of the labels are cryptic, particularly for the ChromHMM track data.

Figure 2e-f: consider including AUC statistics.

There are typos and grammatical issues. Some examples:

Abstract

- "evolutionarily constraint" -> "evolutionary constraint"
- "Using human genetic variation, regulatory sequence motifs, mouse epigenomic data, and retrospectively considered additional human data" - awkward phrase in bold – maybe "and retrospectively considering existing human data"?

Introduction

- "evolutionary constrained bases" -> "evolutionarily constrained bases"
- "quantified information in epigenomic data with polymorphism and divergence information" – awkward phrasing and difficult to follow
- "based on the genome-wide distribution of an evolutionary constraint score in the annotation in the cell type" – also unclear. What is meant by "score in the annotation in the cell type"?
- "greater flexibility in terms of what data sets are used to and how" – awkward phrasing. Better to just say "what data sets are used to generate the score and how their information is combined"

Results:

- "ReMap provides a uniform processing TF binding data"
- "at which CNE are bases are defined"
- "substantially stronger than compared to"
- "are regulatory active"

Methods:

- "Computing actual observed and expected average CNEP scores for features and genome-wide"

Reviewer #2 (Remarks to the Author):

This paper presents a novel machine learning algorithm that predicts which genomic regions are likely to be conserved in multi species alignments based on various chromatin state annotation data. The resulting CNEP scores predict conserved regions with better precision and recall than previously published similar methods, but still appear to have difficulty predicting the conservation status of some genomic regions using available functional genomic data.

Overall, the method seems interesting and useful, but I found the paper extremely hard to follow. There is a lot of jargon but not many precise definitions of the key terms being used. There are vague statements about the utility of the method, but no demonstrations of concrete toy applications in the context of tasks such as variant prioritization. Fundamentally, it isn't clear whether CNEP is intended to reproduce input conservation tracks as closely as possible, to error-correct these conservation tracks in some way, or to augment them with annotations of regions whose function might have led to their accelerated evolution.

The CNEP method seems clearly useful for investigating whether certain patterns of noncoding DNA conservation are driven by the need to conserve patterns of gene regulatory activity in particular tissues; in other words, for learning about the distribution of fitness effects of noncoding mutations and the relationship between fitness effect magnitude and disruption of gene regulation in a particular tissue. However, the introduction instead motivates CNEP as a useful tool for "variant prioritization," and I can't see how it's actually useful for that task. The paper doesn't provide a precise definition for "variant prioritization," but the idea seems to be that it is useful to know which genomic regions are conserved because variants occurring within conserved regions are more likely to explain disease heritability than variants occurring outside conserved regions. However, since we already know where the conserved regions are, how is it helpful to have an additional tool that can predict where these conserved regions are, but does so imperfectly? For the task of variant prioritization, isn't it better to simply look at which variants are present in conserved regions and in which tissues those variants occur in regulatory peaks? How exactly does CNEP help with this task? Maybe it would be helpful to ask how much of the landscape of conservation can be explained by e.g. brain tissue functional annotation alone, but that analysis doesn't appear to be a focus of the paper. Instead, functional data from all tissues is pooled together to explain the entire conservation landscape as much as possible, and the paper does not explain how this reveals variants likely cause disease and which tissues explain the disease's mechanism of action. The output of FitCons and FitCons2 seems better tailored to solve the motivating problem of variant prioritization than CNEP is, and the paper doesn't explain how having a single, non-cell-type-dependent score is more useful than the outputs of these programs.

There is some interesting discussion of the regions that CNEP fails to classify correctly, but I wasn't clear on the takeaway of this section. At first, the paper presents CNEP's superior classification ROC curve as an unequivocal advantage over similar methods, but this can't be taken for granted if some of the discrepancy between CNEP and conservation annotations might be due to false positive and false negative calls of constraint, or a failure of conserved element detectors to identify regions like human accelerated regions that are functionally significant but not completely conserved. If the paper is acknowledging that conserved element calls might have errors and that it is desirable for CNEP to detect other modes of functionality besides purifying selection, then how can we tell whether CNEP is outperforming other underperforming the competing methods that are less good at reproducing the input conservation tracks? Is the ultimate goal for CNEP to be able to identify all nonneutrally evolving regions? What is the paper's final verdict on how much of the discrepancy between CNEP and conservation tracks is a deficiency on CNEP's part versus a limitation of the input conservation tracks?

In the section of the paper that describes CSS-CNEP, it's hard to tell whether the usage of conservation scores to predict conservation cores becomes circular. It sounds like the method is looking at the conservation state at site s , training the model across sites on other chromosomes that have the same conservation state, and then predicting how conserved site s is relative to those other sites with the same conservation state. What is this score meant to be predictive of if not the fact that site s has a certain conservation state? This seems not very far from the tautological exercise of using CNEP to predict which sites in the genome have low CNEP scores. What is the orthogonal information being predicted here?

The definition of the CNEP underestimation value also seems counterintuitive. When you compare the average CNEP score under a set of peaks to the expected CNEP score based on the overlap between those peaks and constrained elements, I believe that this evaluates the performance of CNEP to predict how functionally important these peaks are, but this doesn't necessarily imply that adding these peaks to the CNEP model would bring the predicted and actual scores closer together. What if the predicted CNEP score is higher than the actual score? Then the predicted functionality would be an overestimate, not an underestimate, and presumably addition this additional set of peaks would cause the difference between the two scores to increase, not decrease. Alternatively, the CNEP score at a peak set might be close to the value predicted based on conservation, but the marginal value of that set of peak information might be high if a CNEP

score computed without that track of peak information would have been much farther from the expected score. There could be two functional tracks that have a peak on the same genomic position, and in that case it doesn't appear that this criterion would be able to tell whether the peak is still predicted to be conserved after leaving out only one of the sets. I assumed the goal of computing a CNEP underestimation value would be to estimate how much better the model gets at predicting conserved elements when adding a specific set of peaks to the training set, and if this is the correct goal, wouldn't it be better to report the correlation between the CNEP scores computed without this additional peak set and the CNEP scores computed with this peak set added? If I'm misunderstanding the goal of this score, it would be helpful to spell out that goal more explicitly.

Summary: Interpreting the consequences of non-coding variants remains a difficult challenge. Here, Grujic et al present the Constrained Non-Exonic Predictor (CNEP), a computational method that generates a score reflecting the probability that a non-exonic base occurs in an evolutionarily constrained non-exonic element using available information about epigenomic marks and transcription factor binding sites. They compare their tool to other approaches that prioritize non-coding variation. To investigate CNEs with low CNEP scores, the authors introduce CSS-CNEP, a tool that further integrates information about conservation derived from their ConsHMM tool. The SS-CNEP tool suggests that CNEs with low CNEP scores are significantly biased toward particular conservation states. They perform an analysis of high and low CNE/non-CNE sequences and find enrichment and depletion for different categories of polymorphic variation and functional motif suggesting that sequences in these categories are subjected to different selective pressures. Finally, they evaluate feature sets that add the most information to the CNEP score which provides insight as to what types of experiments may be most informative of CNEs.

Overall, this is an interesting paper that uses machine learning to generate new insights into the intersection between constrained regulatory elements, conservation and genetic variation, and reveals places where we experiments may be most effective to increase our understanding. Although the authors suggest at times that their tool could be useful for prioritizing non-coding variants, the work in the manuscript does not explicitly address this. Instead, a more focused and deeper analysis of evolutionary constraints and study biases related to constrained noncoding elements might make a more exciting story. The authors also need to make sure they place their work into context w.r.t. similar machine learning based tools and analyses. There is some mention in the introduction but it is not comprehensive and it needs to be revised for clarity. The description of the CNEP method itself is a bit complicated and while the results suggest that the approach is working, clarification of choices made with respect to classifier design would be helpful.

We thank the reviewer for the summary and the positive comments. We also thank the reviewer for the constructive feedback that we feel has strengthened the manuscript. We address the points raised above in our response to the specific comments below.

Major Comments:

The authors use various features to train CNEP, which comprises an ensemble of logistic regression classifiers. Aspects of this process should be clarified:

Features: The number of features is large and it is not clear, for example, how peak calls map to individual features. Is each dataset considered its own feature, even if they ChIPed for the same factor? Or are they considered different features only if they were performed in different cell lines?

We thank the reviewer for raising this point and we apologize that this was unclear. Each peak call data set corresponds to an individual feature. We have revised the methods to clarify and now state:

“In total 58,484 features were based on peak call data sets, where each data set corresponds to one feature.”

Additionally we have extended Supplementary Table 1, to list explicitly all the input features.

It is expected that a number of the features are likely to be highly correlated with one another. Did the authors evaluate correlation among features? Such correlation would seem particularly important when using regression models for the underlying classifiers, as highly correlated features could lead to multicollinearities if they are considered together (although it was unclear from the methods whether multivariable regression was used, or whether a different set of classifiers were used for each individual feature).

We thank the reviewer for raising these points.

We first note that CNEP does use a multivariable logistic regression and we apologize that this was not clear. To make this more explicit, in the first subsection of the results section we have added the word 'jointly' to this sentence

“We applied CNEP jointly with 63,741 features.”

We have also added a sentence in the Methods section under 'CNEP Method' which states “CNEP jointly considers all 63,741 features described above.”

In addition, as described in the response to the comment on the 'CNEP score' below we have clarified that for individual features we were computing statistics that are single numbers, not genomewide scores.

We note that while considering multicollinearities in a regression model is important when trying to analyze the coefficient weights of features, in this paper we do not analyze the coefficient weights thus our analysis should not be effected by multicollinearities of the features. In this paper we only use the model for predictions. We have revised the discussion and included a sentence addressing this point:

“We do not recommend trying to directly interpret the weights of the CNEP classifiers due to the multicollinearity expected when using a large number of features.”

Also in terms of fitting the model, we used a L_1 regularized lasso regression, and this regularization will remove a subset of features if they are not informative due to multiple collinearity. We have added a sentence in the methods now reporting the average percentage of features with non-zero coefficients, specifically we state:

“With this regularization on average 59.8% of the features per classifier had a non-zero coefficient and 80.5% of features were used by at least one classifier.”

In previous papers we have analyzed the correlation between many of the features considered here (e.g. Ernst and Kellis, Genome Research 2013; Ernst and Kellis, Nature Biotechnology 2015; Roadmap Epigenomics Consortium et al, Nature 2015), but feel those types of analyses would be outside the scope of this manuscript.

Also, the authors should comment on the relationship between ChIPAtlas/ENCODE/Roadmap features and ChromHMM features since ChromHMM ingests chromatin mark annotations predict higher order chromatin states and is thus a derivative of those features. Are both features

types required to obtain the optimal performance? How does performance change when one set of features is omitted?

We thank you for the suggestion and questions. We have extended Supplementary Fig. 3 to report the performance of CNEP with just ChromHMM features and all features except ChromHMM features. We observed a small increase when including the ChromHMM features, the average AUC over the four constrained element sets increased from 0.832 to 0.839 in predicting constrained non-exonic bases from all other bases. ChromHMM features alone had AUC between 0.70 and 0.77 depending on the constrained element set. We note that ChromHMM features represent only 8% of all features and many data sets used in generating the CNEP score have not been incorporated in ChromHMM annotations.

We have added into the methods the statement in the subsection ‘Epigenomics and TF binding features’ after describing the ChromHMM features:

“We observed a small increase in predictive performance of CNEP when including the ChromHMM features in addition to the peak features (**Supplementary Table 3**).”

We show below the updated **Supplementary Table 3**, with the first row showing the predictive performance with all features, the sixth row all features except ChromHMM features, and the seventh row ChromHMM features only.

Method	Exons included in Negatives					Exons excluded from Negatives and Positives			
	PhastCons	GERP++	SPhy-omega	SPhy-pi		PhastCons	GERP++	SPhy-omega	SPhy-pi
CNEP	0.79	0.86	0.86	0.84		0.79	0.86	0.87	0.84
CNEP - matched	0.79	0.86	0.86	0.84		0.79	0.86	0.86	0.84
CNEP - five classifiers per element set	0.79	0.86	0.86	0.84		0.79	0.86	0.86	0.84
CNEP - single classifier per element set	0.76	0.83	0.83	0.80		0.76	0.83	0.83	0.80
CNEP - 2015 features only	0.75	0.82	0.82	0.81		0.75	0.82	0.83	0.81
CNEP - excluding ChromHMM features	0.78	0.86	0.86	0.83		0.78	0.86	0.86	0.83
CNEP - ChromHMM features only	0.70	0.76	0.77	0.76		0.70	0.76	0.77	0.76
All Features Count	0.62	0.64	0.68	0.69		0.62	0.64	0.68	0.69
All DNase-seq count	0.67	0.71	0.74	0.74		0.67	0.71	0.75	0.75
Roadmap Epigenomics DNase-seq Count	0.67	0.71	0.74	0.73		0.67	0.72	0.74	0.74
Segway Encyclopedia - 'conservation-associated activity score'	0.57	0.60	0.64	0.66		0.58	0.61	0.66	0.67
FitCons2 - 'cell-type integrated score'	0.57	0.58	0.62	0.62		0.58	0.59	0.63	0.63
CNEP - on sample of 50,000 data points genomewide	0.81	0.84	0.87	0.84		0.81	0.84	0.87	0.85
DeepSea max data set - on sample of 50,000 data points genomewide	0.64	0.65	0.66	0.66		0.65	0.66	0.67	0.67
DeepSea average data set - on sample of 50,000 data points genomewide	0.52	0.53	0.53	0.54		0.53	0.53	0.55	0.54
CNEP - on sample of 1 million data points on chr10	0.81	0.84	0.86	0.84		0.81	0.84	0.86	0.84
Random Forest - on sample of 1 million data points on chr10	0.77	0.80	0.82	0.81		0.77	0.80	0.83	0.82

Supplementary Table 3: AUC performance at predicting constrained elements – The first row of the table provides the AUC values for the ROC curves for the CNEP score. The AUC values are shown for each of the four constrained non-exonic element sets considered both when bases overlapping exons are included in the negative set and when they are excluded from both the positive and negative sets for the evaluations. The second row of the table shows the AUC values based on just the score trained on that element set opposed to the average from the four elements, which led to similar performance. The next two

rows show the predictive performance when CNEP uses five and one classifier per constrained element set instead of 10 to predict for one chromosome. The AUC values were computed based on using the same 10 classifiers as the ensemble with classifiers, but split into either two groups of five or ten groups of one and the AUC values based on each group was averaged. The next row shows the performance of CNEP based on a subset of features available by 2015 in the retrospective analysis. The following two rows report the performance of all features except the 5,215 ChromHMM features, and the performance using only the ChromHMM features. Below that is the baseline of counting the number of present features overlapping a base, counting the number of present features from DNase I experiments and just Roadmap Epigenomics DNase I experiments. Also shown is the performance of two existing scores, the Segway Encyclopedia 'conservation-associated activity score'(Libbrecht et al., 2019) and FitCons2 'cell-type integrated score'(Gulko and Siepel, 2019). The next row is gray section break and the following three rows report the performance on a sample of 50,000 randomly sampled genomic positions. The first of those is the CNEP score followed by the maximum and average performance of DeepSea predictions available from its webserver (Zhou and Troyanskaya, 2015). The next row is a gray section break and two rows that report performance on 1 million randomly sampled positions on chr10. The first of these two rows report CNEP's performance using a logistic regression classifier. The second of these two rows reports CNEP's performance using a random forest in place of a logistic regression classifier.

Labels: Labels are assigned based on different constrained element sets derived from other methods. How correlated are the predictions of the underlying methods? Are the constrained element sets highly redundant or highly complementary? Some statistics are mentioned, but it is not clear what is driving the correlation. Is the agreement more on the basis of presence of constrained elements or absence of constrained elements?

We thank you for the questions. In response to these questions, we have computed the correlation between each element set by giving a genomic base that overlaps a constrained non-exonic element a value of 1, and a genomic base that does not overlap a constrained non-exonic element a value of 0. These are shown below and have now been added to the current **Supplementary Fig. 3**. The correlations between pairs of element sets were between 0.51 and 0.72, compared to 0.88 and 0.93 when considering CNEP's predictions individually. We also created a combined frequency track, which is the proportion of the four constrained element sets overlapping a base, and that had correlations in the range 0.79 to 0.87 with the individual element set tracks. In comparison, the combined CNEP prediction had correlations of 0.96 and 0.98 with the individual tracks.

These results suggest the different constrained element sets are somewhere in between being highly redundant and highly complementary. That the correlation of predictions of CNEP based on different constrained elements is more correlated suggests there could be a smoothing or denoising aspects to the predictions. While the presence of a constrained element provides full information about the absence of a constrained element, bases that overlap a constrained element are more unique and less common so from that perspective it could be said correlations would be more driven by the presence of the constrained element. We are reporting in the current **Supplementary Fig. 4e** the % of the genome that overlapped a constrained non-exonic base, which ranged between 3.0 and 5.3% depending on the constrained element set.

We have revised the results section of the main text to now state:

"Predictions based on training on any two different constrained element sets were all highly correlated, with correlations ranging from 0.88 to 0.93, and greater than the correlations directly between different constrained element sets" (**Supplementary Fig. 3**).

We have revised **Supplementary Fig. 3** as shown below:

Supplementary Figure 3: CNEP score correlation with scores based on individual constrained element sets. (a) The panel shows the genome-wide pairwise correlations between the predictions based on applying the CNEP method to each constrained element individually and the CNEP score. The CNEP score was determined based on averaging predictions based on training on each of the four constrained element sets separately. **(b)** The panel shows the genome-wide pairwise correlations computed directly based on the constrained element labels. These were computed by first encoding the value of '1' if a base was a non-exonic base that overlapped a constrained element of the set and a 0 otherwise. Correlations between different element sets were computed directly based on this encoding. Additionally a combined track was computed as the proportion of the four constrained element sets that overlapped a base, and the correlation of each individual set with this combined track was computed.

Classifier: The classifier comprises 10 logistic regression models per chromosome per constrained element set, each trained on a different set of random samples. How was the number 10 determined? Among these 10, how correlated are the predictions? In other words, would the authors obtain different results if they trained 1 logistic regression classifier per chromosome and constrained element set?

We thank you for the questions. In general, ensembling classifiers can improve predictions, but there is diminishing returns expected in the improvement with each additional classifiers. At least in our application there would still be linear increases in computation time and use of disk storage for each additional classifier. We used 10 classifiers since it is a round number and felt it would represent a reasonable balance between gaining improved predictions without excessive computation time and disk usage.

We have conducted additional analyses to demonstrate the benefit of ensembling and also the diminishing returns if the number of classifiers included in the ensemble increased beyond 10. Specifically we now also report the average AUC performance of CNEP when using one classifier per constrained element set. We also report the average AUC performance of CNEP when we split the 10 classifiers into two halves. When using a single classifier per constrained element set the average AUC for predicting constrained non-exonic elements from the rest of the genome across the four constrained element sets is 80.4 compared to 83.9 when using an ensemble of ten classifiers. When using an ensemble of five classifiers the average AUC was 83.5, illustrating the diminishing returns that would be expected increasing beyond 10 classifiers. We show the results for predictions of individual constrained element sets in an updated **Supplementary Table 3**.

Additionally, and in response to the reviewer's questions, we computed the average pairwise correlation between predictions based on a single classifier for each element set. This correlation was 0.73. This result implies we would get different predictions when using only one classifier per constrained element set than using the predictions based on averaging more than one. We note that the correlation between the two sets of predictions when we split the classifiers into two halves with five classifiers per element set increased substantially to 0.93.

We have extended the 'CNEP method' sub-section of the Methods of the text to now include:

"We verified that using 10 classifiers per constrained element set for predictions on one chromosome led to improved predictive performance compared to a single classifier (**Supplementary Table 3**). We also verified that there was similar predictive performance with five classifiers as with ten classifiers (**Supplementary Table 3**), thus increasing beyond 10 classifiers would be expected to give diminishing returns, while increasing computational costs. Additionally, we observed that the average pairwise correlation between predictions based on using a single classifier of the 10 per element set was 0.73, but when the 10 classifiers were split into two halves, the correlation of predictions based on five classifiers per element set increased to 0.93."

The updated **Supplementary Table 3** was shown above. The first row shows the predictive performance with all features, the third row based on five classifiers, and the fourth row a single classifier.

How many features are evaluated by each classifier – all features every time? Many ensemble learners randomly sample the feature space to reduce correlation between the individual classifiers.

We thank you for the question. All features were given to the classifier, but since we were applying L1 regularization, the coefficients of a subsets of features were set to 0. The specific features with non-zero coefficients were different for different classifiers in the ensemble. With the default regularization parameter of 1 that we used, on average 59.8% of the 63,741 features were used and 80.5% were used by at least one classifier. As noted above we have added a sentence in the methods, which states:

“With this regularization on average 59.8% of the features had a non-zero coefficient and 80.5% of features were used by at least one classifier.”

The choice of logistic regression classifiers assumes a linear relationship between the feature space and the outcome variable. Did the authors evaluate any nonlinear alternatives?

We thank the reviewer for raising this point. We have now added a comparison using a RandomForest for CNEP in place of a logistic regression classifier. We used a random forest as implemented in sci-kit learn with default settings. This led in our evaluation to an average AUC of 0.80 across the four constrained element sets when using a random forest compared to 0.83 when using logistic regression classifier. The values for individual constrained element sets are shown in **Supplementary Table 3** also shown above. We have also added the following sentence to the main text:

“Additionally, we also evaluated the predictive performance of CNEP using a random forest in place of a logistic regression classifier and saw a small decrease in predictive performance (**Supplementary Table 3**).”

Training set: Classifiers are trained based on 1 million randomly sampled positions from each chromosome. How often do randomly selected training examples overlap – how much overlap between training sets for individual classifiers?

Randomly selected training examples between two classifiers overlap on average 0.034% of the time. This number is low since we are sampling from the entire approximately 3 billion bases of the genome. We have revised the ‘CNEP’ method section to provide this information in these sentences:

“For generating CNEP scores on one chromosome based on one constrained element set, CNEP trained ten logistic regression classifiers using ten different sets of 1,000,000 randomly sampled positions from the other 22 chromosomes. The overlap of positions of any one set of positions with another set was on average 0.034% of positions.”

What is the balance between positive (is a constrained element) and negative (is not a constrained element) training examples? If there is a large class imbalance, did the authors try using e.g. firth’s penalized regression? This may be unnecessary, however without knowing the degree of imbalance, it is hard to say.

The % of positive training examples across all training sets ranged from 2.9% to 5.5% and was on average 4.1%. We are reporting in what is now **Supplementary Fig. 4e** the genomewide % bases that are in constrained non-exonic bases, for each constrained element sets, and the

positive % corresponds approximately to these values. We have added a sentence to provide more information about this in the CNEP method subsection of the Methods:

“Across all classifiers, the number of positive examples per classifier was on average 4.1% and varied between 2.9% and 5.5%, and corresponded approximately to the genomewide percent of CNE bases for the constrained element set (**Supplementary Fig. 4e**).”

While these percentages are not large since we trained with a million training points per classifier so in absolute terms there are still a large number of examples for the lower frequency class, which would make firth’s penalized regression unnecessary if one is interested in parameter effect sizes. Additionally as explained above our focus is on the predictions of the classifier and not estimating parameter effect sizes.

Why does the CNEP negative set include exons? This asserts that regulatory elements will never overlap an exon which may or may not be true. Why not omit exons from the training set altogether?

We thank the reviewer for the questions. We first note that while the negative set for training included all bases in exons, some bases still had high CNEP scores. We have extended **Supplementary Fig. 4**, also shown below, to show the cumulative distribution of the CNEP score within exons, as well as the constrained and non-constrained subset of exons. This figure also shows for each value of the CNEP score, a greater fraction of bases within exons exceeded that score than genomewide, and the proportion rises further when restricted to constrained bases.

By including exons in the negative set, we can be more confident that high CNEP scores within exons are associated with a regulatory signature. If CNEP tried to make predictions for exons without having seen any training data for them, there is the risk the classifier prediction would be unstable for them, which means we could not interpret a high score to mean a likely regulatory element.

We do appreciate there are arguments that can be made for both approaches, but for the reasoning above we felt it made more sense to include constrained exonic bases in the negative set. We also note that bases in exons that overlap constrained elements only represent 1.3-1.7% of bases sampled for the negative set. We have also excluded exons from our downstream analyses when defining CNE and notCNE bases.

We have added sentences in the methods section under the ‘CNEP method’ subsection which states

“We note that bases in constrained elements within exons, which corresponds to 1.3-1.7% of bases outside of non-exonic constrained elements, were included in the negative set. The specific value in the range depends on the specific constrained element set. We included such bases in the negative set instead of excluding them from the training, since CNEP predictions were still made in exons, and this way high prediction values within exons can be more confidently interpreted as being associated with the exon also overlapping epigenomic marks and TF binding typical of non-exonic constrained elements (**Supplementary Fig. 4**).”

We also updated **Supplementary Fig. 4** with the panels shown below

Supplementary Figure 4: Cumulative distribution of CNEP scores for additional constrained element sets and genome coverage. (a) Similar plot to Fig. 2a, the graph shows the cumulative distribution of the CNEP score genome-wide (green), in constrained non-exonic (CNE) bases (red), and bases that are not in constrained elements and also not in exons (notCNE) (blue) for PhastCons. This plot is extended to also show the cumulative distribution of the CNEP score with exons (black), bases within exons that overlap a constrained element (cyan), and bases within exons that do not overlap a constrained element (magenta). Similar plots to (a), but showing plots for (b) GERP++, (c) SiPhy-omega and (d) SiPhy-pi constrained element sets.

CNEP score: Please clarify whether there is a CNEP score for each feature. The methods text includes the following: “For computing the actual observed average CNEP score for a feature, we computed the average CNEP score in all bases in the genome where the feature was defined as being present.” This implies that there is a separate score for each feature.

We thank you for raising this point and we apologize for the confusion. For each feature, we computed two statistics that are each single numbers. These statistics are the observed and expected CNEP score in bases where the feature is present. There is not a separate genomewide score track generated for each feature.

We have revised that sentence to now read:

“For computing the observed average CNEP score statistic for a feature, we computed the average CNEP score in all bases in the genome where the feature was defined as being present.”

We have made additional edits to the manuscript to clarify that these are statistics.

CNEP-CSS score: The average CNEP score across a set of bases with the same conservation level is taken, with 100 different conservation states considered as annotated by ConsHMM.

Does it make sense to consider all 100 states?

Thank you for the question. Yes, we think it makes sense to consider all 100 states. Each base is only assigned to one conservation state so in order to consistently use the conservation state assignment of a base all states need to be considered. Since the CSS-CNEP averages associated with different conservation state and constrained element combinations are inferred from every base on the other chromosomes we are in general well powered to consider all states in deriving the score. To clarify on this point we have added a sentence in methods that states:

“For over 99.98% of base predictions, the combination average used was derived from greater than 1,000 bases.”

We note that we used the conservation state annotations from Arneson and Ernst, 2019 and the choice of the number of states in that context was explained there.

How are constrained elements distributed across the 100 ConsHMM categories? How many bases are associated with each of the 100 ConsHMM categories across the genome?

Thank you for the questions. We have added the % of the genome each state covers to main **Fig. 3**. This was previously and still included in what is now **Supplementary Fig. 8**. It is also available in a bar-graph form in Supplementary Fig. 7 of the Arneson and Ernst, 2019 paper.

The % of CNE bases will simply be the product of the genome % of the state and the CNE fold enrichments for a state. The CNE fold enrichments are also reported in main Fig 3 for PhastCons and **Supplementary Fig. 8** for all constrained element sets. If the question is with respect to constrained elements including those that overlap exons we report state enrichments for them in Supplementary Fig. 16 of Arneson and Ernst, 2019.

An overview figure describing what information is used and how would be helpful to understand the study design.

We thank you for the suggestion. We have added it to panel a of figure 1, also shown below

Figure 1: Overview and example predictions of Constrained Non-exonic Predictor (CNEP) and Conservation Signature Score by CNEP (CSS-CNEP). (a) A flow chart giving an overview of the approach to learn the CNEP score and the CSS-CNEP, which are shown in shaded boxes. The input to learn the CNEP score are a set of 63,741 epigenomic and TF binding features such as ChIP-seq and DNase-seq peaks and ChromHMM annotations. These features are integrated using an ensemble of logistic regression classifiers that are trained based on annotations of constrained non-exonic elements

from four methods. The CSS-CNEP is derived based on the CNEP score along with the four constrained element sets and ConsHMM annotations²⁹.

The analysis of CNE/non-CNE with respect to conservation is interesting and could be expanded.

Thank you for the positive comment and suggestions for expansions.

Were there any spatial characteristics distinguishing low and high CNE/non-CNE bases?

Did they tend to be more proximal/distant relative to genes or ncRNAs?

Thank you for the questions. In the revised version of the manuscript we now include seven positional enrichment analyses listed below for each of the four constrained element sets.

(i) High_notCNE bases relative to CNE

(ii) Low_CNE bases relative to exons

(iii) High_notCNE bases relative to exons

(iv) Low_CNE bases relative to TSS

(v) High_notCNE bases relative to TSS

(vi) Low_CNE bases relative to gencode genes (includes both protein coding and non-coding)

(vii) High_notCNE bases relative to gencode genes (includes both protein coding and non-coding)

The first two of these were in our original submission and the remaining five are new to this revised version.

We show below the portion of the new supplementary figures based on PhastCons elements. Plots based on all the constrained element sets can be found in **Supplementary Figs. 10-14**. For each analysis we are showing either the fold enrichment of High_notCNE bases relative to notCNE bases or Low_CNE bases relative to CNE bases. We are also plotting the cumulative fraction of High_notCNE or Low_CNE bases.

Supplementary Figure 10: High_notCNE bases relative to exons. (a) The plot shows the fold enrichment for the cumulative number of PhastCons High_notCNE bases relative to notCNE bases for bases being within each distance to an exon, up to 3,000bp. **(b)** The plot shows the cumulative fraction of PhastCons High_notCNE bases at each distance to the nearest exon, up to 3,000bp.

Supplementary Figure 11: Low_CNE bases relative to transcription start sites. (a) The plot shows the ratio of the fold enrichment for the cumulative number of PhastCons Low_CNE bases relative to CNE bases for bases being within each distance to a transcription start site, up to 100kbp. **(b)** The plot shows the cumulative fraction of PhastCons Low_CNE bases at each distance to the nearest transcription start site, up to 100kbp.

Supplementary Figure 12: High_notCNE bases relative to transcription start sites. (a) The plot shows the ratio of the fold enrichment for the cumulative number of PhastCons High_notCNE bases relative to notCNE bases for bases being within each distance to a transcription start site, up to 100kbp. **(b)** The plot shows the cumulative fraction of PhastCons High_notCNE bases at each distance to the nearest transcription start site, up to 100kbp.

Supplementary Figure 13: Low_CNE bases relative to genes. (a) The plot shows the ratio of the fold enrichment for the cumulative number of PhastCons Low_CNE bases relative to CNE bases for bases being within each distance to a gene, up to 100kbp. **(b)** The plot shows the cumulative fraction of PhastCons Low_CNE bases at each distance to a gene, up to 100kbp.

Supplementary Figure 14: High_notCNE bases relative to genes. (a) The plot shows the fold enrichment for the cumulative number of PhastCons High_notCNE bases relative to notCNE bases for bases being within each distance to a gene, up to 100kbp. **(b)** The plot shows the cumulative fraction of PhastCons High_notCNE bases at each distance to a gene, up to 100kbp.

Was there any relationship to transposable elements?

We thank you for the question. To investigate this we have added an analysis where we show the enrichment for CNE, High_CNE, Low_CNE, notCNE, High_notCNE, and Low_notCNE bases for each constrained element for all bases in RepeatMasker elements, as well as in specific classes or families covering more than a million bases. In general, constrained bases deplete for repeat elements. Interestingly, we did observe substantial differences in the enrichment or depletion for specific repeat element classes among High_notCNE bases.

We have added **Supplementary Fig. 9** also shown below showing these enrichments.

Repeat Type	CNE				High_CNE				Low_CNE				notCNE				High_notCNE				Low_notCNE				High_notCNE/notCNE			
	PhastCons	SIphy-Omega	SIphy-PI	GERP++	PhastCons	SIphy-Omega	SIphy-PI	GERP++	PhastCons	SIphy-Omega	SIphy-PI	GERP++	PhastCons	SIphy-Omega	SIphy-PI	GERP++	PhastCons	SIphy-Omega	SIphy-PI	GERP++	PhastCons	SIphy-Omega	SIphy-PI	GERP++	PhastCons	SIphy-Omega	SIphy-PI	GERP++
repeats - all	0.35	0.17	0.22	0.37	0.19	0.14	0.16	0.14	0.65	0.30	0.39	0.25	1.11	1.11	1.12	1.14	0.62	0.62	0.64	0.66	1.26	1.26	1.26	1.27	0.56	0.56	0.57	0.58
DNA_class	0.64	0.41	0.47	0.41	0.51	0.38	0.41	0.39	0.88	0.56	0.63	0.48	1.10	1.10	1.11	1.12	1.34	1.35	1.39	1.43	1.03	1.03	1.03	1.03	1.22	1.23	1.25	1.28
LINE_class	0.40	0.16	0.19	0.16	0.10	0.11	0.13	0.13	0.80	0.31	0.36	0.27	1.12	1.12	1.13	1.14	0.65	0.65	0.67	0.69	1.26	1.26	1.27	1.27	0.58	0.58	0.59	0.61
Low_complexity_class	1.24	0.84	0.79	0.95	0.92	0.77	0.73	0.87	1.82	1.08	0.96	1.23	1.04	1.05	1.06	1.05	0.90	0.92	0.94	0.92	1.08	1.09	1.09	1.09	0.87	0.88	0.88	0.87
LTR_class	0.20	0.07	0.13	0.06	0.09	0.05	0.07	0.04	0.43	0.17	0.29	0.10	1.12	1.12	1.13	1.15	0.63	0.63	0.65	0.68	1.27	1.26	1.27	1.27	0.56	0.56	0.57	0.59
Other_class	0.14	0.00	0.00	0.00	0.00	0.00	0.00	0.00	0.41	0.00	0.00	0.00	1.12	1.12	1.13	1.15	0.01	0.01	0.01	0.01	1.45	1.45	1.46	1.46	0.01	0.01	0.01	0.01
Satellite_class	0.12	0.01	0.00	0.01	0.02	0.00	0.00	0.00	0.31	0.02	0.01	0.02	1.13	1.12	1.14	1.15	0.09	0.09	0.10	0.10	1.44	1.43	1.44	1.44	0.00	0.00	0.00	0.00
Simple_repeat_class	0.86	0.22	0.27	0.25	0.27	0.18	0.20	0.23	1.97	0.35	0.46	0.30	1.08	1.10	1.11	1.12	0.53	0.54	0.55	0.56	1.24	1.26	1.27	1.27	0.49	0.49	0.49	0.50
SINE_class	0.20	0.14	0.22	0.13	0.12	0.11	0.15	0.11	0.36	0.26	0.42	0.20	1.11	1.11	1.12	1.13	0.40	0.40	0.41	0.43	1.33	1.32	1.32	1.33	0.36	0.36	0.36	0.38
Unknown_class	4.99	6.10	4.96	5.06	6.20	6.57	5.51	5.34	2.71	4.34	3.39	4.15	0.91	0.92	0.89	0.84	1.75	1.74	1.71	1.58	0.66	0.67	0.66	0.64	1.92	1.90	1.91	1.87
Alu_family	0.07	0.00	0.04	0.00	0.00	0.00	0.00	0.00	0.19	0.01	0.14	0.00	1.12	1.11	1.13	1.14	0.04	0.04	0.04	0.04	1.45	1.43	1.44	1.45	0.03	0.03	0.03	0.03
centr_family	0.02	0.00	0.00	0.00	0.00	0.00	0.00	0.00	0.05	0.00	0.00	0.00	1.14	1.13	1.15	1.16	0.06	0.06	0.06	0.07	1.46	1.45	1.46	1.46	0.05	0.05	0.06	0.06
CRI_family	1.97	2.06	1.94	2.16	2.03	1.95	1.87	2.03	1.85	2.48	2.11	2.58	1.04	1.05	1.04	1.01	1.92	1.93	1.94	1.91	0.78	0.78	0.78	0.77	1.84	1.84	1.87	1.88
ERV1_family	0.18	0.02	0.03	0.02	0.06	0.01	0.02	0.02	0.40	0.04	0.07	0.03	1.12	1.12	1.14	1.15	0.54	0.54	0.56	0.58	1.30	1.29	1.30	1.30	0.48	0.48	0.49	0.51
ERV4_family	0.08	0.00	0.00	0.00	0.01	0.00	0.00	0.00	0.21	0.00	0.01	0.00	1.12	1.11	1.13	1.14	0.19	0.19	0.20	0.21	1.40	1.39	1.40	1.40	0.17	0.17	0.17	0.18
ERV1_family	0.23	0.09	0.17	0.07	0.11	0.06	0.09	0.08	0.46	0.21	0.39	0.11	1.12	1.12	1.13	1.15	0.81	0.81	0.84	0.88	1.21	1.21	1.21	1.22	0.72	0.73	0.74	0.77
ERV1-MaLR_family	0.19	0.08	0.15	0.05	0.07	0.04	0.07	0.03	0.43	0.22	0.37	0.11	1.12	1.12	1.13	1.15	0.59	0.59	0.61	0.64	1.28	1.27	1.28	1.28	0.53	0.53	0.54	0.56
Gypsy?_family	0.74	0.72	0.99	0.68	0.64	0.58	0.78	0.59	0.94	1.25	1.60	0.98	1.10	1.10	1.09	1.11	1.58	1.58	1.59	1.66	0.96	0.96	0.95	0.96	1.43	1.44	1.46	1.50
Gypsy_family	0.76	0.65	0.77	0.64	0.62	0.54	0.60	0.56	1.02	1.05	1.25	0.90	1.09	1.09	1.10	1.11	1.35	1.35	1.38	1.42	1.02	1.02	1.01	1.02	1.23	1.24	1.26	1.28
HAT_family	0.55	0.27	0.37	0.19	0.34	0.18	0.26	0.14	0.94	0.62	0.70	0.32	1.09	1.10	1.11	1.12	1.52	1.53	1.57	1.65	0.97	0.97	0.97	0.98	1.39	1.39	1.42	1.47
HAT-Blackjack_family	0.54	0.24	0.37	0.23	0.38	0.18	0.28	0.20	0.84	0.45	0.65	0.34	1.11	1.11	1.12	1.13	1.63	1.64	1.69	1.76	0.95	0.95	0.95	0.96	1.47	1.48	1.51	1.55
HAT-Charlie_family	0.63	0.36	0.47	0.36	0.46	0.29	0.38	0.31	0.95	0.66	0.74	0.51	1.10	1.10	1.11	1.12	1.57	1.58	1.62	1.69	0.95	0.96	0.96	0.96	1.43	1.44	1.47	1.51
HAT-Tip100_family	0.53	0.25	0.35	0.24	0.37	0.19	0.23	0.21	0.85	0.47	0.68	0.34	1.11	1.11	1.12	1.13	1.39	1.40	1.45	1.51	1.02	1.02	1.02	1.03	1.26	1.26	1.30	1.33
L1_family	0.34	0.07	0.08	0.08	0.12	0.04	0.05	0.05	0.77	0.16	0.18	0.15	1.12	1.12	1.14	1.15	0.52	0.52	0.54	0.56	1.30	1.30	1.31	1.31	0.46	0.46	0.47	0.48
L2_family	0.45	0.32	0.49	0.29	0.25	0.20	0.30	0.21	0.83	0.77	1.04	0.54	1.11	1.11	1.11	1.13	1.10	1.09	1.12	1.17	1.11	1.11	1.11	1.12	0.99	0.99	1.01	1.04
MIR_family	0.64	0.57	0.82	0.54	0.48	0.42	0.62	0.45	0.94	1.13	1.39	0.84	1.09	1.09	1.09	1.10	1.72	1.72	1.74	1.82	0.90	0.90	0.90	0.91	1.58	1.58	1.60	1.65
RTX_family	1.75	1.59	1.48	1.87	1.69	1.47	1.42	1.75	1.85	2.06	1.65	2.27	1.05	1.06	1.06	1.03	2.13	2.15	2.19	2.15	0.73	0.74	0.74	0.73	2.02	2.02	2.06	2.08
Satellite_family	0.31	0.03	0.02	0.03	0.06	0.01	0.01	0.01	0.77	0.08	0.04	0.07	1.11	1.11	1.13	1.14	0.16	0.16	0.17	0.18	1.39	1.39	1.40	1.41	0.14	0.15	0.15	0.16
TcMar-Mariner_family	0.70	0.49	0.49	0.62	0.58	0.47	0.48	0.58	0.92	0.56	0.54	0.73	1.07	1.08	1.08	1.08	1.12	1.13	1.16	1.16	1.06	1.06	1.06	1.06	1.05	1.05	1.07	1.07
TcMar-Tc2_family	1.03	0.59	0.55	0.77	0.85	0.50	0.52	0.71	1.37	0.92	0.62	0.96	1.08	1.10	1.11	1.10	1.96	1.99	2.05	2.08	0.82	0.83	0.84	0.83	1.81	1.82	1.86	1.89
TcMar-Tigger_family	0.45	0.27	0.31	0.29	0.33	0.26	0.28	0.28	0.69	0.31	0.39	0.33	1.11	1.11	1.12	1.13	0.96	0.96	0.99	1.02	1.16	1.15	1.16	1.16	0.86	0.87	0.88	0.90

Supplementary Figure 9: Enrichments for Repeat Elements. The figure shows heatmaps for the enrichments for CNE, High_CNE, Low_CNE, notCNE, High_notCNE, and Low_notCNE bases for each of the constrained element sets for repeats called by RepeatMasker followed by a heatmap for the ratio of the High_notCNE to notCNE enrichments. The first row reports enrichments for all repeats and the following set of rows for repeat classes then followed by repeats for families. Only repeat classes or families with at least a million bases are shown.

Was there any bias for promoter or enhancer elements to be associated with low versus high CNE?

We thank you for the question. In response to this question we have added an analysis where we have computed the enrichment of CNE, High_CNE, Low_CNE, notCNE, High_notCNE, and Low_notCNE bases for each of the four constrained element sets across 25 different chromatin states based on the median from 127 reference epigenomes. As expected both promoter and enhancer chromatin states are more likely to be associated with High instead of Low bases for both CNE and notCNE bases.

We have added **Supplementary Fig. 8** also shown below showing these enrichments.

state	Genome %	CNE				High_CNE				Low_CNE				notCNE				High_notCNE				Low_notCNE			
		H3acCons	H3K9me3	H3K9me1	GERP++	H3acCons	H3K9me3	H3K9me1	GERP++	H3acCons	H3K9me3	H3K9me1	GERP++	H3acCons	H3K9me3	H3K9me1	GERP++	H3acCons	H3K9me3	H3K9me1	GERP++	H3acCons	H3K9me3	H3K9me1	GERP++
1_TssA	0.18	3.03	4.17	3.88	3.07	4.36	5.00	4.83	3.79	0.54	1.01	0.96	0.71	0.64	0.52	0.46	0.49	1.70	1.65	1.53	1.61	0.13	0.15	0.18	0.18
2_Promu	0.41	2.10	2.86	2.93	2.24	2.87	3.30	3.49	2.67	0.64	1.21	1.30	0.83	0.79	0.77	0.74	0.75	1.95	1.91	1.82	1.92	0.43	0.42	0.42	0.42
3_PromD1	0.41	1.99	2.83	2.94	2.28	2.75	3.24	3.48	2.72	0.55	1.30	1.38	0.84	0.64	0.62	0.58	0.60	1.81	1.76	1.66	1.74	0.28	0.27	0.26	0.27
4_PromD2	0.19	1.43	1.66	1.71	1.84	1.70	1.74	1.85	1.98	0.91	1.40	1.32	1.35	0.87	0.87	0.85	0.84	1.75	1.75	1.72	1.69	0.61	0.61	0.60	0.60
5_Tss	2.22	0.95	0.74	0.78	0.81	0.74	0.64	0.67	0.74	1.33	1.13	1.09	1.06	1.02	1.03	1.03	1.03	0.97	0.98	0.99	0.99	1.04	1.04	1.04	1.04
6_Tss	0.70	0.71	0.77	0.87	0.70	0.52	0.54	0.61	0.53	1.07	1.66	1.61	1.27	0.94	0.83	0.83	0.84	0.88	0.87	0.88	0.90	0.82	0.82	0.81	0.82
7_Tss	3.48	0.87	0.72	0.90	0.82	0.43	0.46	0.56	0.49	1.13	1.73	1.55	1.25	0.82	0.82	0.82	0.83	0.79	0.78	0.78	0.82	0.84	0.83	0.83	0.83
8_TssA	5.88	0.94	0.80	0.86	0.81	0.75	0.69	0.74	0.73	1.29	1.22	1.24	1.07	0.97	0.97	0.97	0.97	0.91	0.92	0.92	0.93	0.98	0.99	0.98	0.99
9_TssReg	0.30	1.98	2.74	2.73	2.42	2.66	3.07	3.15	2.83	0.67	1.43	1.49	1.16	0.83	0.81	0.78	0.78	2.11	2.06	2.01	2.03	0.44	0.43	0.43	0.43
10_TssEnh5	0.38	1.40	1.80	1.99	1.59	1.69	1.88	2.07	1.74	0.82	1.43	1.67	1.13	0.95	0.94	0.92	0.93	1.74	1.73	1.70	1.74	0.71	0.70	0.69	0.71
11_TssEnh3	0.21	0.85	1.20	1.48	0.98	0.85	1.04	1.23	0.88	0.84	1.84	2.15	1.30	0.75	0.74	0.72	0.74	1.11	1.09	1.06	1.12	0.63	0.62	0.61	0.62
12_TssEnhW	0.51	1.24	1.35	1.46	1.37	1.34	1.34	1.44	1.42	1.00	1.38	1.48	1.23	0.98	0.98	0.97	0.97	1.54	1.54	1.53	1.53	0.81	0.81	0.81	0.81
13_EnhA1	0.22	2.35	3.13	3.20	2.70	3.22	3.64	3.80	3.22	0.68	1.31	1.56	0.97	0.90	0.89	0.85	0.86	2.10	2.05	1.95	2.02	0.52	0.52	0.51	0.52
14_EnhA2	0.54	2.61	3.44	3.54	2.98	3.65	4.03	4.22	3.54	0.68	1.23	1.47	0.95	0.81	0.80	0.86	0.86	2.25	2.21	2.11	2.14	0.51	0.51	0.50	0.51
15_EnhA4	0.48	1.74	2.24	2.41	1.95	2.26	2.87	3.07	2.26	0.77	1.30	1.54	0.98	0.86	0.85	0.92	0.93	1.83	1.80	1.74	1.81	0.69	0.68	0.67	0.69
16_EnhW1	0.28	2.21	2.90	2.93	2.38	3.00	3.34	3.46	2.83	0.73	1.21	1.33	0.90	0.92	0.91	0.88	0.88	2.01	1.98	1.90	1.97	0.59	0.59	0.58	0.59
17_EnhW2	0.95	1.77	2.24	2.38	1.97	2.31	2.48	2.61	2.27	0.79	1.30	1.60	0.99	0.95	0.95	0.92	0.93	1.78	1.75	1.69	1.75	0.71	0.70	0.69	0.71
18_EnhAc	0.27	1.84	2.25	2.24	2.07	2.41	2.57	2.62	2.42	0.80	1.14	1.29	0.96	0.96	0.95	0.94	0.93	1.79	1.78	1.73	1.76	0.70	0.70	0.70	0.70
19_Divise	0.63	2.69	3.44	3.26	2.92	3.71	4.06	3.98	3.51	0.74	1.14	1.19	0.87	0.93	0.92	0.89	0.89	2.24	2.21	2.15	2.16	0.54	0.54	0.54	0.54
20_DNF/Rpts	0.18	0.48	0.19	0.15	0.18	0.07	0.04	0.04	0.04	1.26	0.70	0.48	0.49	0.84	0.85	0.86	0.87	0.18	0.18	0.19	0.19	1.04	1.04	1.05	1.08
21_Het	0.91	0.42	0.16	0.16	0.48	0.06	0.06	0.09	0.06	1.09	0.34	0.29	0.32	1.02	1.02	1.04	1.05	0.19	0.19	0.19	0.19	1.29	1.29	1.30	1.30
22_PromP	0.20	2.23	2.66	2.46	2.47	2.96	3.08	2.92	2.90	0.87	1.07	1.05	0.88	0.86	0.86	0.85	0.83	1.85	1.84	1.81	1.77	0.56	0.57	0.56	0.56
23_PromDv	0.25	2.61	3.52	3.38	2.86	3.63	4.08	4.09	3.43	0.66	1.32	1.32	0.97	0.73	0.72	0.69	0.68	2.09	2.05	1.96	2.02	0.33	0.32	0.32	0.32
24_ReprIC	1.32	1.54	1.85	1.80	1.52	1.89	1.98	1.93	1.68	0.89	1.37	1.40	0.98	0.93	0.93	0.92	0.92	1.47	1.46	1.45	1.48	0.78	0.78	0.78	0.78
25_Oules	78.38	0.94	0.90	0.89	0.94	0.92	0.90	0.88	0.92	0.98	0.92	0.90	0.89	1.02	1.02	1.03	1.02	0.95	0.95	0.96	0.95	1.04	1.04	1.05	1.04
Base	100	3.87	3.01	4.37	5.33	2.53	2.38	3.25	4.08	1.34	0.63	1.12	1.25	92.13	92.99	91.63	90.67	21.14	21.29	20.43	19.99	70.99	71.70	71.21	71.03

state	Genome %	Low_CNE/CNE				High_notCNE/notCNE			
		H3acCons	H3K9me3	H3K9me1	GERP++	H3acCons	H3K9me3	H3K9me1	GERP++
1_TssA	0.18	0.18	0.24	0.25	0.24	3.17	3.15	3.16	3.27
2_Promu	0.41	0.30	0.42	0.45	0.37	2.48	2.47	2.47	2.55
3_PromD1	0.41	0.28	0.46	0.47	0.37	2.84	2.84	2.86	2.92
4_PromD2	0.19	0.64	0.84	0.77	0.73	2.01	2.01	2.02	2.02
5_Tss	2.22	1.40	1.53	1.40	1.31	0.95	0.95	0.96	0.96
6_Tss	0.70	1.51	1.14	1.16	1.03	1.05	1.05	1.06	1.08
7_Tss	3.48	1.67	2.39	2.06	2.00	0.96	0.95	0.96	0.98
8_TssA	5.88	1.37	1.53	1.45	1.33	0.95	0.95	0.95	0.96
9_TssReg	0.30	0.34	0.52	0.54	0.48	2.55	2.54	2.57	2.61
10_TssEnh5	0.38	0.59	0.79	0.84	0.71	1.84	1.84	1.85	1.87
11_TssEnh3	0.21	0.99	1.52	1.45	1.33	1.48	1.48	1.48	1.53
12_TssEnhW	0.51	0.81	1.03	1.02	0.90	1.57	1.57	1.58	1.58
13_EnhA1	0.22	0.29	0.42	0.49	0.36	2.34	2.31	2.29	2.36
14_EnhA2	0.34	0.26	0.36	0.41	0.32	2.47	2.45	2.45	2.49
15_EnhA4	0.48	0.44	0.58	0.68	0.50	1.91	1.90	1.89	1.94
16_EnhW1	0.28	0.33	0.42	0.45	0.38	2.19	2.18	2.17	2.22
17_EnhW2	0.95	0.44	0.58	0.68	0.50	1.86	1.85	1.84	1.88
18_EnhAc	0.27	0.44	0.51	0.57	0.46	1.87	1.86	1.85	1.89
19_Divise	0.63	0.28	0.38	0.36	0.38	2.41	2.40	2.41	2.43
20_DNF/Rpts	0.18	2.51	3.76	3.21	3.39	0.21	0.21	0.22	0.22
21_Het	0.91	2.59	3.49	2.94	3.19	0.12	0.12	0.12	0.12
22_PromP	0.20	0.35	0.40	0.43	0.40	2.14	2.13	2.15	2.14
23_PromDv	0.25	0.25	0.37	0.39	0.34	2.85	2.84	2.86	2.92
24_ReprIC	1.32	0.58	0.74	0.78	0.64	1.58	1.58	1.58	1.60
25_Oules	78.38	1.05	1.02	1.02	1.05	0.93	0.93	0.93	0.93
Base	100								

Supplementary Figure 8: Enrichments for Chromatin States. The heatmaps on the top of the figure shows the median enrichments for CNE, High_CNE, Low_CNE, notCNE, High_notCNE, and Low_notCNE bases for 25 ChromHMM chromatin states based on 12 marks using imputed data across 127 reference epigenomes (Ernst and Kellis, 2015). The values in the table are fold enrichments computed using ChromHMM. The first column reports the % of the genome each state covers and the bottom row the % of the genome the corresponding column covers. The heatmaps on the bottom row show the ratio of the Low_CNE to CNE enrichments and High_notCNE to notCNE enrichments.

We have added a paragraph into the results section summarizing these new analyses, which now states:

“We further analyzed properties of High_notCNE and Low_CNE bases with respect to their enrichments for chromatin states and repeat elements and additional spatial enrichments relative to genes, transcription start sites (TSS), and exons (**Supplementary Fig. 8-14**). This highlighted the enrichment of High_notCNE bases relative to notCNE bases proximal to TSS and exons, in enhancers and promoter chromatin states, and in the DNA class of repeats. Among chromatin states, Low_CNE bases relative to CNE bases showed the strongest enrichment for the heterochromatin and zinc finger gene chromatin states.”

In the discussion, the authors state “The CNEP score is designed to allow a researcher interested in gaining additional insights into a variant prioritized based on being in a constrained base to determine whether there are annotations within a compendium of epigenomic and TF binding data to explain the constraint.” If the authors wish to claim that their tool is useful for variant prioritization, they should consider comparing its performance to other published methods (CADD, DANN, DeepSea, etc) benchmark sets of clinically or biologically relevant non-coding variants (GWAS variants, eQTLs, etc). See “Rentzsch P, Witten D, Cooper GM, Shendure J, Kircher M. CADD: predicting the deleteriousness of variants throughout the human genome. *Nucleic Acids Res.* 2018 Oct 29. doi: 10.1093/nar/gky1016. PubMed PMID: 30371827.” for an example.

We apologize for the confusion surrounding variant prioritization and we thank the reviewer for the feedback that this aspect of the manuscript was not clear. We are not trying to position the CNEP score directly as a variant prioritization score. Our reason for mentioning variant prioritization was they can serve as one motivation for interest in constrained element annotations. Bases prioritized by a number of variant prioritization scores, including scores that consider a diverse set of genomic annotations, heavily overlap with constrained element annotations. To make that connection more explicit we have added a supplementary figure (**Supplementary Fig. 2**, also shown below) showing the enrichment within constrained elements for top 1% non-coding prioritized variants by 14-different variant prioritization scores within non-exonic regions. We used the same set of variant scores and prioritized bases as described in (Arneson and Ernst, 2019), which included CADD and DANN. We did not include DeepSea for this specific analysis as there is no publicly available genomewide DeepSea score, and we also note that DeepSea defines a set of features and depending on how they are integrated and with what other information the score will vary.

We have added the following section in the results section after introducing the constrained elements that we used.

“We note that bases prioritized by a number of different popular variant prioritization scores, including scores that integrate diverse annotations, heavily enrich in constrained elements called by these methods (**Supplementary Fig. 2**).”

The added supplementary figure is shown here:

Constrained element coverage and enrichment of top 1% prioritized non-coding variants in

		Non-exon bases							
Prioritization Score	% Non-exon	% Overlap				Fold Enrichment			
		PhastCons	SiPhy-Omega	SiPhy-PI	GERP++	PhastCons	SiPhy-Omega	SiPhy-PI	GERP++
CADD	0.85	96.2	95.0	94.3	98.7	23.9	30.3	20.7	17.8
PhastCons	1.32	100.0	86.5	85.7	93.8	24.8	27.6	18.8	16.9
Eigen	0.77	91.7	89.2	90.0	93.2	22.7	28.5	19.8	16.8
REMM	0.81	88.6	88.4	89.9	95.4	21.9	28.2	19.7	17.2
FATHMM	0.74	86.2	77.5	80.6	85.1	21.4	24.7	17.7	15.3
GERP++	0.84	82.3	78.2	83.4	88.9	20.4	24.9	18.3	16.0
phyloP	0.76	80.8	67.2	70.1	76.4	20.0	21.4	15.4	13.8
LINSIGHT	0.85	70.3	67.6	74.8	74.8	17.4	21.6	16.4	13.5
funSeq2	0.80	18.0	18.4	24.1	23.6	4.5	5.9	5.3	4.2
fitCons	0.42	13.4	13.6	18.2	18.1	3.3	4.3	4.0	3.3
Eigen_PC	0.75	11.0	12.1	16.8	16.3	2.7	3.9	3.7	2.9
CDTS	0.78	10.4	10.1	15.0	15.0	2.6	3.2	3.3	2.7
DANN	0.90	11.0	6.2	6.5	6.7	2.7	2.0	1.4	1.2
FIRE	0.81	1.4	1.0	2.3	2.2	0.3	0.3	0.5	0.4
% Non-Exons		4.0	3.1	4.6	5.6	4.0	3.1	4.6	5.6

Supplementary Figure 2: Constrained non-exonic element enrichment for bases prioritized by different variant prioritization scores. The figure shows the enrichment within bases in constrained elements for bases that were among the top 1% of prioritized non-coding bases by different variant prioritization scores restricted to non-exonic regions. The set of fourteen different variant prioritization scores and top 1% non-coding bases was taken from (Arneson and Ernst, 2019). The rows correspond to different variant prioritization scores. The first column after the name of the variant prioritization scores is the percent of top 1% of non-coding bases for that score is within non-exonic regions. The percentages are not exactly 1% because of ties in the score and some non-coding bases fall within exons. The next four set of columns report the percent overlap of the prioritized bases with non-exonic bases for four different constrained element set annotations used with CNEP. The final four columns report the fold enrichment of the overlap. The bottom row reports the % of non-exonic genome bases of each of the four constrained element sets cover. This figure highlights how for a number of different variant prioritization scores the top prioritized bases within non-exonic regions heavily enrich for constrained elements including some scores that consider a diverse set of genomic annotations.

Additionally we revised parts of the abstract and introduction to be more general about constrained non-exonic elements and variants within them, and less focused specifically on prioritized variants.

We have revised the beginning of the abstract to now read:

“Annotations of evolutionary constraint and genome-wide maps of epigenomic marks and transcription factor binding provide important complementary information for understanding the genome and genetic variation.”

We have also revised the introduction, the first three paragraphs, which now state:

“A large majority of genetic variation associated with common disease falls into non-exonic regions of the human genome¹ motivating in part the need to better annotate and understand

such portions of the genome. Annotations of evolutionarily constraint²⁻⁵ and maps of epigenomic marks and transcription factor binding represent two complementary types of information to annotate the non-exonic genome⁶⁻⁹. Supporting the importance of evolutionary constraint annotations, heritability analyses have suggested they are heavily enriched for disease associated variants¹⁰ and they have been an important feature to integrative methods for prioritizing potentially deleterious non-exonic mutations^{11,12}.

While useful, evolutionary constraint annotations do not directly provide information on the type of genomic element or the cell or tissue types of activity. Genome-wide maps of histone modifications and variants, open chromatin, chromatin state annotations, and transcription factor (TF) binding can give such insights^{6,7,13-15}. However, such data is specific to the condition and cell or tissue type in which the experiments underlying them were conducted. Previous analyses have shown that while there is an enrichment for evolutionarily constrained bases in epigenomic or transcription factor binding annotations of regulatory activity, some evolutionarily constrained bases lack informative annotations^{4,14,16-21}.

Thus when investigating the role of constrained non-exonic elements, or variants within such elements, with a compendium of epigenomic and TF binding, an initial question is whether the constraint can even be explained by existing data in the compendium. However, with tens of thousands epigenomic and TF binding data sets available, answering this question is not straightforward. Integrative scores such as CADD¹² that combine epigenomic data and TF binding features with conservation features cannot be directly applied to answer such a question, since a base could receive a high score based on the conservation features even if informative epigenomic or TF binding annotations were lacking.

Also we have revised the discussion including adding these two sentences to the discussion to be more explicit on the point that CNEP and CSS-CNEP are not designed to be used directly as variant prioritization scores. We do note a possible future direction would be to evaluate incorporating these scores with other features as part of variant prioritization, but feel this would be outside the scope and focus of this manuscript.

“We note that the CNEP and CSS-CNEP scores are not specifically designed to be used directly to prioritize variants, and would not be expected to be competitive with scores that use more diverse sets of features. A possible future direction of work is to evaluate incorporating these scores with other features used by variant prioritization methods.”

If they simply wish to portray it as annotating variants according to overlap with likely constrained non-coding elements, they should still be sure to compare to the most similar methods (there is some justification for the choice of methods in the introduction, but this needs to be clearer, and it seems to omit some similar methods, for example DeepSea: “Zhou, Jian, and Olga G. Troyanskaya. "Predicting effects of noncoding variants with deep learning–based sequence model." *Nature methods* 12.10 (2015): 931-934.”).

We consider FitCons2 ‘cell-type integrated score’ and the Segway ‘conservation-associated activity score’ the most similar existing scores to the CNEP score that we are aware of, which we have been comparing to along with multiple baseline scores. We have made some edits of the first sentence of the fourth paragraph of the introduction, also shown below, to make clearer what makes these scores more similar to the CNEP score. Scores such as CADD that use sequence

constraint as features we do not consider comparable score to CNEP as explained in the third paragraph of the introduction shown above.

We consider sequence based predictions, whether from DNA sequence motifs or machine learning based models such as DeepSea, to have a different complementary and purpose. Sequence based predictions do not directly provide information for a given base in a constrained element as to whether the constraint can be explained by experimental data in a compendium. In our original submission we had conducted analyses using sequence based motifs of transcription factors thus demonstrating their complementary nature. We have added a sentence at the end of the fourth paragraph shown below discussing sequence based predictions.

Additionally we have added a comparison showing that the DeepSea sequence based predictions of 919 chromatin features are less predictive of bases in constrained non-exonic elements than CNEP predictions. The average AUC was in the range [0.52,0.54] and the maximum AUC was in the range [0.64,0.66]. With the full set of results in **Supplementary Table 3**, also shown above.

The revised fourth paragraph of the introduction now reads:

“A few scores have been proposed that quantify information or activity in epigenomic or TF binding data based on evolutionary information, and thus do not use conservation as features. The FitCons and FitCons2 methods^{22,23} quantified information in epigenomic data using probabilistic evolutionary models to provide cell type specific estimates of fitness. A ‘conservation-associated activity score’ was recently introduced, which provided cell type specific scores using Segway chromatin state annotations and evolutionary constraint information²⁴. Additionally, for both of these approaches a single summary score of the cell type specific scores was also computed. While informative, if one is interested in a single score summarizing information in a compendium of epigenomic and TF data, an approach that defines the single score without going through cell type specific scores would have greater flexibility in terms of what data sets are used to generate the score and how information within them are combined. There also exist methods to computationally predict epigenomic marks, TF binding data, and constrained elements from DNA sequence^{25–27}. However, those methods do not directly provide information for a given base in a constrained element as to whether the constraint can be explained by available experimental data in a compendium.”

We have added this sentence into this section of the results “CNEP outperforms baselines and related scores at predicting constrained non-exonic elements”

“We also evaluated the predictive performance of individual sequence based predictions for a set of 919 chromatin features⁸, and also saw substantially lower predictive performance (maximum AUC in the range [0.64,0.66]).”

Also, it was unclear in the manuscript whether CNEP and CSS-CNEP annotate a variant site according to the specific features associated with a variant site being designated a CNE (i.e. does the variant affect a specific transcription factor binding site, or a ChIP peak for a specific target?).

We thank the reviewer for pointing out this was unclear. The CNEP or CSS-CNEP scores do not directly do that, but we have added to the CNEP software the ability to rank data sets with a peak overlapping a position in the genome by the expected CNEP score statistics. This is now discussed in the discussion and we include in the revised Fig. 1b examples of the top ranking data sets. The added text in the discussions is as follows:

“For a researcher interested in studying a CNE base, the CNEP score provides information as to whether data in a compendium is sufficient to explain the constraint. To facilitate further follow-up on bases with a high CNEP score, the CNEP software provides the ability to receive a ranked list of input features that overlap a variant with the features ranked by their expected CNEP score statistic (**Fig. 1b**). We do not recommend trying to directly interpret the weights of CNEP classifiers due to multicollinearity expected when using a large number of features. If one is interested in determining whether a subset of features are sufficient to explain the constraint, such as from the same cell type, CNEP could be run with that specific subset of features.”

Can the authors show an application of the tool to annotate a set of variants?

We have changed the example in **Fig. 1b** to show variants in a constrained non-exonic element shown to have a role in isolated pancreatic agenesis from Weedon et al, Nature Genetics 2014, as explained in this paragraph added in the results section:

“As a specific example of the CNEP score, we consider the PTF1A loci, where in a previous study variants in distal non-exonic constrained bases were shown to cause isolated pancreatic agenesis (**Fig. 1b**). The study identified genetic variants associated with pancreatic agenesis mapping to conserved sequences, but noted that it lacked informative annotations from the ENCODE or Roadmap Epigenomics projects²¹. After conducting epigenomic and transcription factor mapping in human embryonic-derived pancreatic progenitor cells, the study obtained informative enhancer annotations. The CNEP score at those bases is relatively high, conveying to a researcher that there now exist epigenomic and transcription factor data sets that explain the constraint, but which would still not be apparent from several baseline or related existing scores (**Fig. 1b**).”

Figure 1: Overview and example predictions of Constrained Non-exonic Predictor (CNEP) and Conservation Signature Score by CNEP (CSS-CNEP). (b) An example genomic locus containing the PTF1A gene illustrating CNEP scores and the CSS-CNEP. The top line is the GENCODE gene annotation track. The next line shows the location of five variants in close proximity that were previously identified to be associated with isolated pancreatic agenesis and fell into constrained non-exonic elements, but had limited prior epigenomic and TF binding annotations²¹. A vertical box is also shown going through these variants. The next line is the CNEP score track, which is followed by tracks for CSS-CNEP and the PhastCons score. Below the PhastCons score track are tracks for the four constrained element sets used to train CNEP: PhastCons, GERP++, SiPhy-omega, and SiPhy-pi^{3,49,50}. The next track is a dense view of the ConSHMM annotation track²⁹, which combined with the four constrained element sets were used to define the conservation signatures for CSS-CNEP. A color legend for the ConSHMM conservation state annotations is available in **Fig. 5a**. Segments in red show high aligning and matching the human reference genome through fish, while those in orange have it through some non-mammalian vertebrates. The next three tracks show baseline scores of counts of total features, DNase-seq features, and DNase-seq features from Roadmap Epigenomics overlapping a base. The two tracks after that show to related existing scores, the FitCons2 cell type integrated score²² and the Segway Encyclopedia conservation-associated activity score²⁴. CNEP score gave a higher relative score to the boxed variants overlapping constrained elements than the three baseline and two related scores. For example, for the initial associated variant discovered associated with the phenotype, (chr10:23508437), was ranked among CNEP non-exonic bases at the top 5.4% of CNEP, 22.8% for the Segway Encyclopedia conservation-associated activity score, and in the top 46.8-49.4% of the remaining baselines and FitCons2. The final set of three tracks show peak calls of data sets that overlapped the SNP and were the top three data sets in terms of their overlap with constrained non-exonic elements as defined by the expected CNEP score statistic. These data sets were PDX1 and FoxA1 in hESC derived pancreatic progenitors^{21,51}.

Minor Comments:

Figure 1 is crowded, the labels are sometimes masked or cutoff and many of the labels are cryptic, particularly for the ChromHMM track data.

We thank the reviewer for the feedback. As discussed above we have changed the example loci for Fig. 1 and the set of tracks displayed. We are no longer displaying the ChromHMM tracks in the figure.

Figure 2e-f: consider including AUC statistics.

We thank the reviewer for the suggestion. We have added them and also in the corresponding **Supplementary Fig. 5**. We show here the updated Fig. 2e-f with the AUC values included

There are typos and grammatical issues. Some examples:

Thank you for pointing these out. We have fixed these below and made some additional edits.

Abstract

- “evolutionarily constraint” -> “evolutionary constraint”

Thank you for point this out. We have made this fix.

- “Using human genetic variation, regulatory sequence motifs, mouse epigenomic data, and retrospectively considered additional human data” - awkward phrase in bold – maybe ‘and retrospectively considering existing human data’?

Thank you for point this out. We have replaced “and retrospectively considered additional human data” with the reviewer’s suggestion of “and retrospectively considered existing human data”.

Introduction

- “evolutionary constrained bases” -> “evolutionarily constrained bases”

Thank you for point this out. We have made this fix.

- “quantified information in epigenomic data with polymorphism and divergence information” – awkward phrasing and difficult to follow

Thank you for point this out. We have revised to now read: “quantified information in epigenomic data using probabilistic evolutionary models”.

- “based on the genome-wide distribution of an evolutionary constraint score in the annotation in the cell type” – also unclear. What is meant by ‘score in the annotation in the cell type’?

Thank you for point this out. We have simplified the sentence to now read “A ‘conservation-associated activity score’ was recently introduced, which provided cell type specific scores using Segway chromatin state annotations and evolutionary constraint information.”

“greater flexibility in terms of what data sets are used to and how” – awkward phrasing. Better to just say “what data sets are used to generate the score and how their information is combined”

Thank you for point this out and the suggestion. We have followed the suggestion, and the second part of the suggestion now says “greater flexibility in terms of what data sets are used to generate the score and how information within them are combined.”

Results:

- “ReMap provides a uniform processing TF binding data”

Thank you for point this out. We have updated it to read “ReMap provides a uniform processing of TF binding data”

- “at which CNE are bases are defined”

Thank you for point this out. We have updated it to read “at which CNE bases are defined”

- “substantially stronger than compared to”

Thank you for point this out. We have updated it to just read “substantially stronger than”

- “are regulatory active”

Thank you for point this out. We have revised the sentence to now read “These results provide evidence to suggest that Low_CNE bases have modest enrichments in corresponding positions in available mouse samples, particularly related to the brain.”

Methods:

- “Computing actual observed and expected average CNEP scores for features and genome-wide”

Thank you for point this out. We have modified it to now be: “Computing observed and expected average CNEP score statistics”

Reviewer #2 (Remarks to the Author):

This paper presents a novel machine learning algorithm that predicts which genomic regions are likely to be conserved in multi species alignments based on various chromatin state annotation data. The resulting CNEP scores predict conserved regions with better precision and recall than previously published similar methods, but still appear to have difficulty predicting the conservation status of some genomic regions using available functional genomic data.

Overall, the method seems interesting and useful,

Thank you for summary and the positive comment. We also thank you for the feedback that led us to clarify the manuscript and strengthen it.

but I found the paper extremely hard to follow.

We apologize for this. We have revised the manuscript in attempt to make it easier to follow and in particular clarified on all the specific points of confusion that were raised in the reviews.

There is a lot of jargon but not many precise definitions of the key terms being used.

We expanded the description of CNEP underestimate value in the methods as discussed below. We note the CNE, Low_CNE, High_CNE, notCNE, Low_notCNE, and High_notCNE terms were defined in the main text, methods, the current **Supplementary Fig. 4**, and **Supplementary Table 4**. We found it necessary to define these terms in order to keep the manuscript concise. If there are specific other terms the reviewer feels are not precisely defined it would be helpful to know the specific terms.

There are vague statements about the utility of the method, but no demonstrations of concrete toy applications in the context of tasks such as variant prioritization.

We apologize for the confusion here. We have made revisions to the introduction to better clarify what the CNEP score is designed to do including now having this text:

“Here we developed a method, the Constrained Non-Exonic Predictor (CNEP), designed to produce a single probabilistic score, without respect to cell type, that reflects the evidence within a large-scale compendium of epigenomic and TF binding data that a base will be in an evolutionarily constrained non-exonic element. The CNEP score thus quantifies for a researcher interested in specific constrained non-exonic elements, or variants within them, the extent to which the constraint can be explained by existing data in the compendium. Furthermore, the CNEP score enables investigating more general scientific questions about the extent to which information in current epigenomic and TF binding annotations can explain non-exonic sequence constraint, and the nature of constrained bases that cannot be explained by available data.”

We did not intend to motivate CNEP to be used directly as a variant prioritization score. Our reason for mentioning variant prioritization is that it can serve as one motivation for interest in constrained element annotations. Bases prioritized by a number of variant prioritization scores, including scores that consider a diverse set of genomic annotations, heavily overlap with constrained element annotations. To make that connection more explicit we have added a supplementary figure (**Supplementary Fig. 2**, also shown below) showing the enrichment within

constrained elements for top 1% non-coding prioritized variants by 14-different variant prioritization scores within non-exonic regions. We used the same set of variant scores and prioritized bases as described in (Arneson and Ernst, 2019), which included CADD and DANN.

We have added the following section in the results section after introducing the constrained elements that we used.

“We note that bases prioritized by a number of different popular variant prioritization scores, including scores that integrate diverse annotations, heavily enrich in constrained elements called by these methods (**Supplementary Fig. 2**).”

The added supplementary figure is shown here:

Constrained element coverage and enrichment of top 1% prioritized non-coding variants in
Non-exon bases

Prioritization Score	% Non-exon	% Overlap				Fold Enrichment			
		PhastCons	SiPhy-Omega	SiPhy-PI	GERP++	PhastCons	SiPhy-Omega	SiPhy-PI	GERP++
CADD	0.85	96.2	95.0	94.3	98.7	23.9	30.3	20.7	17.8
PhastCons	1.32	100.0	86.5	85.7	93.8	24.8	27.6	18.8	16.9
Eigen	0.77	91.7	89.2	90.0	93.2	22.7	28.5	19.8	16.8
REMM	0.81	88.6	88.4	89.9	95.4	21.9	28.2	19.7	17.2
FATHMM	0.74	86.2	77.5	80.6	85.1	21.4	24.7	17.7	15.3
GERP++	0.84	82.3	78.2	83.4	88.9	20.4	24.9	18.3	16.0
phyloP	0.76	80.8	67.2	70.1	76.4	20.0	21.4	15.4	13.8
LINSIGHT	0.85	70.3	67.6	74.8	74.8	17.4	21.6	16.4	13.5
funSeq2	0.80	18.0	18.4	24.1	23.6	4.5	5.9	5.3	4.2
fitCons	0.42	13.4	13.6	18.2	18.1	3.3	4.3	4.0	3.3
Eigen_PC	0.75	11.0	12.1	16.8	16.3	2.7	3.9	3.7	2.9
CDTS	0.78	10.4	10.1	15.0	15.0	2.6	3.2	3.3	2.7
DANN	0.90	11.0	6.2	6.5	6.7	2.7	2.0	1.4	1.2
FIRE	0.81	1.4	1.0	2.3	2.2	0.3	0.3	0.5	0.4
% Non-Exons		4.0	3.1	4.6	5.6	4.0	3.1	4.6	5.6

Supplementary Figure 2: Constrained non-exonic element enrichment for bases prioritized by different variant prioritization scores. The figure shows the enrichment within bases in constrained elements for bases that were among the top 1% of prioritized non-coding bases by different variant prioritization scores restricted to non-exonic regions. The set of fourteen different variant prioritization scores and top 1% non-coding bases was taken from (Arneson and Ernst, 2019). The rows correspond to different variant prioritization scores. The first column after the name of the variant prioritization scores is the percent of top 1% of non-coding bases for that score is within non-exonic regions. The percentages are not exactly 1% because of ties in the score and some non-coding bases fall within exons. The next four set of columns report the percent overlap of the prioritized bases with non-exonic bases for four different constrained element set annotations used with CNEP. The final four columns report the fold enrichment of the overlap. The bottom row reports the % of non-exonic genome bases of each of the four constrained element sets cover. This figure highlights how for a number of different variant prioritization scores the top prioritized bases within non-exonic regions heavily enrich for constrained elements including some scores that consider a diverse set of genomic annotations.

Additionally we revised parts of the abstract and introduction to be more general about constrained non-exonic variants, and less focused specifically on prioritized variants.

We have revised the beginning of the abstract to now read:

“Annotations of evolutionary constraint and genome-wide maps of epigenomic marks and transcription factor binding provide important complementary information for understanding the genome and genetic variation.”

We have also revised the introduction, the first three paragraphs, which now state:

“A large majority of genetic variation associated with common disease falls into non-exonic regions of the human genome¹ motivating in part the need to better annotate and understand such portions of the genome. Annotations of evolutionarily constraint^{2–5} and maps of epigenomic marks and transcription factor binding represent two complementary types of information to annotate the non-exonic genome^{6–9}. Supporting the importance of evolutionary constraint annotations, heritability analyses have suggested they are heavily enriched for disease associated variants¹⁰ and they have been an important feature to integrative methods for prioritizing potentially deleterious non-exonic mutations^{11,12}.

While useful, evolutionary constraint annotations do not directly provide information on the type of genomic element or the cell or tissue types of activity. Genome-wide maps of histone modifications and variants, open chromatin, chromatin state annotations, and transcription factor (TF) binding can give such insights^{6,7,13–15}. However, such data is specific to the condition and cell or tissue type in which the experiments underlying them were conducted. Previous analyses have shown that while there is an enrichment for evolutionarily constrained bases in epigenomic or transcription factor binding annotations of regulatory activity, some evolutionarily constrained bases lack informative annotations^{4,14,16–21}.

Thus when investigating the role of constrained non-exonic elements, or variants within such elements, with a compendium of epigenomic and TF binding, an initial question is whether the constraint can even be explained by existing data in the compendium. However, with tens of thousands epigenomic and TF binding data sets available, answering this question is not straightforward. Integrative scores such as CADD¹² that combine epigenomic data and TF binding features with conservation features cannot be directly applied to answer such a question, since a base could receive a high score based on the conservation features even if informative epigenomic or TF binding annotations were lacking.

Also we have revised the discussion including adding these two sentences to the discussion to be more explicit on the point that CNEP and CSS-CNEP are not designed to be used directly as variant prioritization scores. We do note a possible future direction would be to evaluate incorporating these scores with other features as part of variant prioritization, but feel this would be outside the scope and focus of this manuscript.

“We note that the CNEP and CSS-CNEP scores are not specifically designed to be used directly to prioritize variants, and would not be expected to be competitive with scores that use more diverse sets of features. A possible future direction of work is to evaluate incorporating these scores with other features used by variant prioritization methods.”

To give a more concrete example of the CNEP in the context of analyzing prioritized of variants, we now show the PTF1A loci highlighted in Weedon et al, Nature Genetics 2014 in **Fig. 1b**, and explain the information the CNEP score conveys about the loci:

“As a specific example of the CNEP score, we consider the PTF1A loci, where in a previous study variants in distal non-exonic constrained bases were shown to cause isolated pancreatic agenesis (**Fig. 1b**). The study identified genetic variants associated with pancreatic agenesis mapping to conserved sequences, but noted that it lacked informative annotations from the ENCODE or Roadmap Epigenomics projects²¹. After conducting epigenomic and transcription factor mapping in human embryonic-derived pancreatic progenitor cells, the study obtained informative enhancer annotations. The CNEP score at those bases is relatively high, conveying to a researcher that there now exist epigenomic and transcription factor data sets that explain the constraint, but which would still not be apparent from several baseline or related existing scores (**Fig. 1b**).”

Figure 1: Overview and example predictions of Constrained Non-exonic Predictor (CNEP) and Conservation Signature Score by CNEP (CSS-CNEP). (b) An example genomic locus containing the PTF1A gene illustrating CNEP scores and the CSS-CNEP. The top line is the GENCODE gene annotation track. The next line shows the location of five variants in close proximity that were previously identified to be associated with isolated pancreatic agenesis and fell into constrained non-exonic elements, but had limited prior epigenomic and TF binding annotations²¹. A vertical box is also shown going through these variants. The next line is the CNEP score track, which is followed by tracks for CSS-CNEP and the PhastCons score. Below the PhastCons score track are tracks for the four constrained element sets used to train CNEP: PhastCons, GERP++, SiPhy-omega, and SiPhy-pi^{3,49,50}. The next track is a dense view of the ConsHMM annotation track²⁹, which combined with the four constrained element sets were used to define the conservation signatures for CSS-CNEP. A color legend for the ConsHMM conservation state annotations is available in **Fig. 5a**. Segments in red show high aligning and matching the human reference genome through fish, while those in orange have it through some non-mammalian vertebrates. The next three tracks show baseline scores of counts of total features, DNase-seq features, and DNase-seq features from Roadmap Epigenomics overlapping a base. The two tracks after that show to related existing scores, the FitCons2 cell type integrated score²² and the Segway Encyclopedia conservation-associated activity score²⁴. CNEP score gave a higher relative score to the boxed variants overlapping constrained elements

than the three baseline and two related scores. For example, for the initial associated variant discovered associated with the phenotype, (chr10:23508437), was ranked among CNEP non-exonic bases at the top 5.4% of CNEP, 22.8% for the Segway Encyclopedia conservation-associated activity score, and in the top 46.8-49.4% of the remaining baselines and FitCons2. The final set of three tracks show peak calls of data sets that overlapped the SNP and were the top three data sets in terms of their overlap with constrained non-exonic elements as defined by the expected CNEP score statistic. These data sets were PDX1 and FoxA1 in hESC derived pancreatic progenitors^{21,51}.

We also believe there is more general scientific value in the CNEP score for understanding the agreement between constraint annotations and epigenomic and transcription factor binding data, and where there might be gaps in existing data. Based on feedback from reviewer 1 we in the revised manuscript have expanded these analyses and also more explicitly highlighted this aspect of the work in the abstract and introduction.

Fundamentally, it isn't clear whether CNEP is intended to reproduce input conservation tracks as closely as possible, to error-correct these conservation tracks in some way, or to augment them with annotations of regions whose function might have led to their accelerated evolution.

We apologize that this was unclear and thank you for the feedback. The CNEP score is designed to predict the constrained element annotations based on only available experimental epigenomic and transcription factor binding annotations. With available epigenomic and transcription factor binding data it is only possible to partially predict constrained elements tracks and not reproduce them. The agreement and disagreements of the CNEP predictions and constrained element annotations are thus informative.

Places where there is agreement between the CNEP score and the constrained element annotations means that available epigenomic and TF binding data can explain the constraint. There can be multiple reasons for disagreement between CNEP score and constrained element annotations including errors in the conservation tracks, accelerated evolution, or data from the condition in which the constrained element is active not being available in a compendium. The CSS-CNEP score can be used to better identify which disagreements are more likely due to errors in the constraint element annotations.

As mentioned above we have revised the introduction to expand the explanation of what the CNEP score is designed to do and why.

We have expanded a sentence in our discussion on the point about accelerated evolution

“Another subset of High_notCNE bases may correspond to adaptive and recently evolved bases with a potentially important regulatory role that share epigenomic marks and TF binding patterns associated with CNE bases, though we note that some adaptive and recently evolved bases might have distinct epigenomic mark and TF binding patterns that CNEP is not optimized to detect.”

We have expanded our treatment of the CSS-CNEP score including having a new main figure and subsection of the results as discussed below.

The CNEP method seems clearly useful for investigating whether certain patterns of noncoding DNA conservation are driven by the need to conserve patterns of gene regulatory activity in particular tissues; in other words, for learning about the distribution of fitness effects of noncoding mutations and the relationship between fitness effect magnitude and disruption of gene regulation in a particular tissue.

However, the introduction instead motivates CNEP as a useful tool for “variant prioritization,” and I can’t see how it’s actually useful for that task. The paper doesn’t provide a precise definition for “variant prioritization,” but the idea seems to be that it is useful to know which genomic regions are conserved because variants occurring within conserved regions are more likely to explain disease heritability than variants occurring outside conserved regions. However, since we already know where the conserved regions are, how is it helpful to have an additional tool that can predict where these conserved regions are, but does so imperfectly?

We apologize for the confusion surrounding CNEP and variant prioritization and clarified as described above what CNEP is designed for. We did not intend CNEP to be used directly as a variant prioritization score, and as explained above mention variant prioritization to motivate our focus on constrained element annotations.

For the task of variant prioritization, isn’t it better to simply look at which variants are present in conserved regions and in which tissues those variants occur in regulatory peaks? How exactly does CNEP help with this task?

We apologize for the confusion surrounding CNEP and variant prioritization and clarified as described above that we did not intend CNEP to be used directly as a variant prioritization score.

As mentioned above we do note a possible future direction of incorporating the CNEP score as a feature to existing prioritization scores though consider this outside the focus of the current manuscript. Many variant prioritization scores are based on machine learning methods that are given a relatively limited number of epigenomic and TF features compared to CNEP and have simple ways to summarize epigenomic annotations across many cell/tissues. For example, CADD summarizes ChromHMM annotations with one feature based on the number of cell/tissue types that state is found in, but ignores the identity of the specific cell or tissue type.

Maybe it would be helpful to ask how much of the landscape of conservation can be explained by e.g. brain tissue functional annotation alone, but that analysis doesn’t appear to be a focus of the paper. Instead, functional data from all tissues is pooled together to explain the entire conservation landscape as much as possible, and the paper does not explain how this reveals variants likely cause disease and which tissues explain the disease’s mechanism of action.

The output of FitCons and FitCons2 seems better tailored to solve the motivating problem of variant prioritization than CNEP is, and the paper doesn’t explain how having a single, non-cell-type-dependent score is more useful than the outputs of these programs.

We believe having a single score based on all the data has complementary value to cell type specific scores. Before one tries to explain the mechanism of action of a constrained non-exonic element based on epigenomic and TF binding data, a researcher may want to rigorously assess whether the compendium even supports a base being in a constrained non-exonic element. As the number of different epigenomic and TF binding data sets has become large, it is challenging

to accurately assess this as the majority of bases overlap peaks in multiple different data sets. To make this point more explicit, we have added this analysis of the set features used to the results:

“The median number of features present for a given bases in the genome is 532, of which at most 263 are chromatin state features. In total 75% of the bases in the genome were in a DNase-seq peak in at least one experiment, including 51% of bases in the genome that were in a peak in at least one of the 350 Roadmap Epigenomics DNase-seq experiments (**Supplementary Fig. 1**). This highlights the limited specificity in the annotation of simply whether a base in the genome overlaps either any input feature or even more specifically a DNase-seq peak.”

We have added this corresponding **Supplementary Fig. 1**:

Supplementary Figure 1: Cumulative Distribution of Overlap with Features. (a) The figure shows the cumulative distribution of the number of features of the 63,741 that overlap a base in the genome. (b) The same as (a) except restricted to the subset of 4,522 features corresponding to a DNase-seq experiment. (c) The same as (b) but further restricted to the subset of 350 DNase-seq experiments uniformly processed

by the Roadmap Epigenomics Consortium. This shows that the majority of bases in the genome are overlapped by one or more of these data sets.

We recognize that if a researcher sees a base with a high CNEP score, they may want to follow-up and have more information about datasets overlapping the base that are associated with increased constraint. While not a direct output of the CNEP method, we have added to the CNEP software the ability to rank data sets with a peak overlapping a position in the genome by the expected CNEP score statistics. This is now discussed in the discussion and we include in the revised **Fig. 1b** examples of the top ranking data sets. In the discussion we now state:

“For a researcher interested in studying a CNE base, the CNEP score provides information as to whether data in a compendium is sufficient to explain the constraint. To facilitate further follow-up on bases with a high CNEP score, the CNEP software provides the ability to receive a ranked list of input features that overlap a variant with the features ranked by their expected CNEP score statistic (**Fig. 1b**).”

Also as discussed above we have placed greater emphasis on gaining a deeper scientific understanding of the agreements and disagreements between constrained elements and epigenomic and TF binding, which should be done in a cell and tissue type agnostic manner since conservation is defined without respect to a specific cell or tissue type.

We also note that the FitCons2 paper also had a cell type agnostic score that summarized the cell type specific scores we explicitly compare to in this manuscript. As discussed in the Introduction if one is going to use a cell type agnostic score there are advantages to directly learning such a score opposed to going through cell type specific scores as we discuss in the Introduction:

“Additionally, for both of these approaches a single summary score of the cell type specific scores was also computed. While informative, if one is interested in a single score summarizing information in a compendium of epigenomic and TF data, an approach that defines the single score without going through cell type specific scores would have greater flexibility in terms of what data sets are used to generate the score and how information within them are combined.”

While not the focus of this manuscript, in principle one could use the CNEP framework to learn cell type specific scores, or scores based on any subset of features of interest. We now mention that in the discussion with this statement

“If one is interested in determining whether a subset of features are sufficient to explain the constraint, such as from the same cell type, CNEP could be run with that specific subset of features.”

There is some interesting discussion of the regions that CNEP fails to classify correctly, but I wasn't clear on the takeaway of this section. At first, the paper presents CNEP's superior classification ROC curve as an unequivocal advantage over similar methods, but this can't be taken for granted if some of the discrepancy between CNEP and conservation annotations might be due to false positive and false negative calls of constraint, or a failure of conserved element detectors to identify regions like human accelerated regions that are functionally significant but not completely conserved.

We apologize for the confusion here. We are only evaluating and claiming that CNEP outperforms other methods at the specific task of predicting constrained non-exonic elements from epigenomic and TF binding data, and not claiming CNEP is necessarily better for other tasks. If one wants to make a statement that a constrained element annotation lacks support from available epigenomic and transcription factor binding data on the basis of a score, then that score should be optimized for predicting constrained non-exonic elements from epigenomic and transcription factor binding data. Otherwise, it is difficult to know if disagreements is due to limitations of the available data or because the score is trying to optimize a different objective.

As with any annotation of the genome, there are some limitations of constrained element annotations, however there is still strong support for the importance of these annotations. For a number of popular variant prioritization scores, including some that use a diverse set of annotations, prioritized non-exonic variants heavily enrich for being in constrained elements. As discussed above, in the revised manuscript we now show this directly in **Supplementary Fig. 2**. Also as referenced in the introduction, heritability enrichment analyses have found constrained element annotations to be more enriched for phenotypic heritability (Finucane et al, Nature Genetics 2015) than other annotations.

We note that the difference in AUC values between CNEP and the Segway Encyclopedia - 'conservation-associated activity score' and the FitCons2 - 'cell-type integrated scores' for predicting constrained non-exonic elements were large and consistent across four different constrained element annotation sets [0.79,0.86] vs. [0.57,0.66]. Seeing such a large and consistent improvement makes it highly unlikely annotation errors can explain the difference in performance. The increased predictive power of constrained non-exonic bases by CNEP shows a better agreement between epigenomic and TF binding annotations and constraint than could have been previously appreciated, and thus providing additional evidence supporting the biological relevance of constrained element annotations.

We have added a sentence to the end of the comparison results section explicitly stating what the comparison was of:

“We note these evaluations were specifically evaluating predicting annotated CNE bases and are not comparing the relative performance of score for predicting adaptive and recently evolved bases.”

If the paper is acknowledging that conserved element calls might have errors and that it is desirable for CNEP to detect other modes of functionality besides purifying selection, then how can we tell whether CNEP is outperforming other underperforming the competing methods that are less good at reproducing the input conservation tracks?

CNEP is not specifically designed to identify other modes of functionality besides purifying selection though bases that score highly by CNEP may still enrich for them. We are not claiming CNEP is outperforming other methods at tasks other than predicting non-exonic constrained element annotations from a compendium of epigenomic and transcription factor binding data. As explained above we believe this task is of interest on its own and the difference between CNEP and other scores is large, in the range [0.79,0.86] for CNEP vs. [0.57,0.66] for other scores with the specific value depending on the specific constrained element set. As mentioned above, we

have also made clarifications to the text related to this point and also as discussed below in response to the next question.

Is the ultimate goal for CNEP to be able to identify all nonneutrally evolving regions?

Thank you for the question and we apologize that this was not already clear from the manuscript. We expect that bases that score highly by CNEP score will enrich for non-neutrally evolving regions, but no, this is not the ultimate goal of the method. A score that also takes into account sequence constraint features would be expected to be better suited for this task. CNEP might identify bases under recent selection that might be missed by sequence constraint, though there is an assumption that bases under recent selection would have similar epigenomic marks and transcription factor binding as those in constrained elements that CNEP is trained on. Alternative methods that identify selection patterns based on variation within humans could potentially identify patterns in epigenomic marks and transcription factor binding that is unique to humans.

As mentioned above we have revised our introduction to be clear on what the CNEP score is designed to do and have text that now states:

“Here we developed a method, the Constrained Non-Exonic Predictor (CNEP), designed to produce a single probabilistic score, without respect to cell type, that reflects the evidence within a large-scale compendium of epigenomic and TF binding data that a base will be in an evolutionarily constrained non-exonic element. The CNEP score thus quantifies for a researcher interested in specific constrained non-exonic elements, or variants within them, the extent to which the constraint can be explained by existing data in the compendium. Furthermore, the CNEP score enables investigating more general scientific questions about the extent to which information in current epigenomic and TF binding annotations can explain non-exonic sequence constraint, and the nature of constrained bases that cannot be explained by available data.”

We have also expanded a sentence we had in the Discussion to now state:

“Another subset of High_notCNE bases may correspond to adaptive and recently evolved bases with a potentially important regulatory role that share epigenomic marks and TF binding patterns associated with CNE bases, though we note that some adaptive and recently evolved bases might have distinct epigenomic mark and TF binding patterns that CNEP is not optimized to detect.”

What is the paper’s final verdict on how much of the discrepancy between CNEP and conservation tracks is a deficiency on CNEP’s part versus a limitation of the input conservation tracks?

We thank the reviewer for the question. To address this question we now present a cumulative distribution plot of the CSS-CNEP for CNE bases in addition to notCNE bases and genomewide. We also present motif and mouse DHS enrichments for CNE bases with values less than multiple different thresholds on the CSS-CNEP to provide additional context on the different score values. The specific amount of the discrepancy depends on the constrained element and is a matter of degree at different thresholds.

In the revised text we now state:

“We also evaluated enrichments of subsets of CNE bases with CSS-CNEP less than specific values for regulatory sequence motifs and mouse DHS (**Fig 6c,d, Supplementary Fig. 30-31**).

This showed substantially reduced enrichments for CNE bases that had lower CSS-CNEP values.”

We have also added the following main Figure 6, and have additional figures in the supplement for other constrained element sets:

Figure 6: Conservation signature score by CNEP (CSS-CNEP) (a) Cumulative distribution of the CSS-CNEP score genomewide and specifically in CNE and notCNE defined by PhastCons. (b) Precision-recall analysis for predicting PhastCons Low_CNE bases among CNE bases using additional comparative genomics information. In this analysis, Low_CNE bases are positive bases and High_CNE bases are negative bases. The predictions based on the CSS-CNEP score as well as the PhastCons, PhyloP, and GERP++ constraint scores are shown based on ranking from lowest to highest value. Also shown for the PhastCons, PhyloP, and GERP++ scores are precision-recall curves, based on dividing a score into four hundred bins and ordering the bins based on increasing enrichment on a training set containing separate positions than used for the evaluation (**Methods**). The plot also shows the cumulative precision recall of the conservation states when ordered based on enrichment for Low_CNE bases in the training data. Additionally, a single point is shown for each of the other three constrained element sets corresponding to predictions based on bases not covered by them. (c) Similar plot to **Fig. 3c**, but showing the difference of the distribution of motif enrichments relative to the distribution for a randomized set of the motifs for PhastCons CNE bases and the subsets that had CSS-CNEP scores ≤ 0.05 , 0.10 , and 0.20 . (d) Similar plot to **Fig. 3e**, but showing enrichments for DNase I Hypersensitive Sites (DHS) from 156 experiments in

mouse, for PhastCons CNE bases and the subsets that had CSS-CNEP scores ≤ 0.05 , 0.10, and 0.20. Similar plots for additional constrained elements can be found in **Supplementary Fig. 28-31**.

In the section of the paper that describes CSS-CNEP, it's hard to tell whether the usage of conservation scores to predict conservation cores becomes circular. It sounds like the method is looking at the conservation state at site s , training the model across sites on other chromosomes that have the same conservation state, and then predicting how conserved site s is relative to those other sites with the same conservation state. What is this score meant to be predictive of if not the fact that site s has a certain conservation state? This seems not very far from the tautological exercise of using CNEP to predict which sites in the genome have low CNEP scores. What is the orthogonal information being predicted here?

We apologize for the confusion here. No the method is not circular. The CSS-CNEP score is designed to provide the expected CNEP score based on the combination of the conservation state and which of the four constrained elements overlap a base. The CSS-CNEP is mapping these 1,600 combinations of annotation possibilities to a univariate score. This score could then be used for example to rank among constrained non-exonic bases with a low CNEP score, which are more likely to be false calls of constraint opposed to lack informative data sets in the compendium. The CNEP score overlapping a base is not used in defining the CSS-CNEP score for a base, only the comparative genomic annotations at the base. The additional information brought in and the reason not all bases in a constrained element called by one method have the same score, is this score also depends on the conservation state assignments in addition to the constrained element annotations called by the other methods.

As mentioned above we have also added additional analyses with orthogonal information showing constrained non-exonic bases that have CSS-CNEP values less than specific thresholds have lower enrichments for motifs and mouse DHS.

To make things clearer we have made a separate sub-section of the results specifically about CSS-CNEP, which now appears at the end and we have added the figure described above. We have revised the paragraph that is now the first paragraph of this results section, which states:

“To rank among CNE bases that have a low CNEP score those that more likely represent false calls of constraint opposed to the compendium lacking informative experimental data of the bases, we defined the Conservation Signature Score by CNEP (CSS-CNEP) based on combining multiple comparative genomic annotations (**Methods, Fig. 1,6a, Supplementary Fig. 28**). The CSS-CNEP integrates the conservation state assignment and combination of which of the four constrained element sets were present at a base to define a score that represents a prediction of the CNEP score using these annotations.”

The definition of the CNEP underestimation value also seems counterintuitive. When you compare the average CNEP score under a set of peaks to the expected CNEP score based on the overlap between those peaks and constrained elements, I believe that this evaluates the performance of CNEP to predict how functionally important these peaks are, but this doesn't necessarily imply that adding these peaks to the CNEP model would bring the predicted and actual scores closer together.

We thank the reviewer for the feedback that the description of the CNEP underestimation value was unclear and we apologize for the confusion. The CNEP underestimate values leverages properties of a logistic regression classifier being a well-calibrated classifier. It is a property of a logistic regression classifier that the average prediction probabilities equals the positive class frequency on training data when trained without any regularization. Additionally the sum of a feature's values for positive training instances equals the weighted sum of the feature's values when weighted by the positive class prediction probabilities on the training data. This means that at least on the training data without regularization, if a feature is included the observed CNEP score statistic for a feature would have to match exactly with what was expected based on the features overlap for constrained non-exonic bases.

To better show that this still also holds approximately for data unseen in training when the model was learned with regularization we have extended the current **Supplementary Fig. 23** to include two additional panels (g) and (h), also shown below. Panel (g) shows the observed and expected CNEP score statistics for the features included in the retrospective model, analogous to panels (a)-(c) for the additional human data sets. Panel (h) shows the distribution of the differences between the observed and expected CNEP score statistics for these same set of features. This shows that for features given to CNEP as input we do not see the distribution of underestimate values exceeding what is expected based on shuffled data, as we did in panels (d-f) for additional data sets. We note that the current **Supplementary Fig. 24b** shows the CNEP underestimate values for the input features given to CNEP plotted against the number of bases a peak set overlaps.

We have updated the main text in response to this point and other points related to CNEP underestimate value as discussed below. Shown here is the current **Supplementary Fig. 23**:

Supplementary Figure 23: Expected vs. observed average CNEP scores for additional human datasets. Similar scatter plots as shown in Fig. 2b except for (a) ChIP-atlas, (b) ENCODE portal, (c) and ReMap 2018 datasets using the retrospective CNEP score based on a subset of features available in 2015 (Methods). Each point corresponding to one dataset of peak calls. The x-axis shows the average CNEP score in bases covered by a peak, while the y-axis shows the expected CNEP score based on the peaks overlap with constrained non-exonic bases. Only datasets that cover at least 200kb are shown. The full table corresponding to these values can be found in **Supplementary Table 8**. The diagonal line is the $y=x$ line. The vertical line corresponds to the genome-wide observed average CNEP score. The horizontal line corresponds to the genome-wide expected average CNEP score. (d-f) Plots for (d) ChIP-atlas (e) ENCODE portal, and (f) ReMap 2018 showing the distribution of prediction underestimate values for datasets with peaks covering at least 200kb. The prediction underestimate value for a dataset is the average difference between the expected CNEP score based on a subset of features available in 2015 (Methods) and the prediction value for each base covered by a peak. Results are shown for prediction values based on the genome-wide average expected CNEP score (blue) and the CNEP score (red). Also shown is the distribution of using the CNEP score for the prediction values, but applied to a shuffled version of each dataset (green). There was a relatively large gap in distribution based on using the observed CNEP score instead of genome-wide expected average CNEP score when computing the difference, highlighting that the CNEP score captures a relatively large amount of information about CNE bases. There is a difference in distributions between using the CNEP score on the actual peaks and a shuffled version of the peaks,

though it was much smaller. These results suggest that CNEP captures most of the marginal information contained in any peak call dataset about the expectation on the frequency of CNE bases. However, there are some datasets that capture some additional marginal information on CNE bases than given by the CNEP score. **(g,h)** Similar plots to (a-c) and (d-f) respectively, but based on the subset of features available in 2015 used for predictions in the retrospective model. The increase of datasets with a positive CNEP underestimate score compared to shuffles was not seen here as it was in (d-f).

What if the predicted CNEP score is higher than the actual score?

Then the predicted functionality would be an overestimate, not an underestimate, and presumably addition this additional set of peaks would cause the difference between the two scores to increase, not decrease.

We thank the reviewer for the question. Yes, if the predicted CNEP score is higher than the actual score then the CNEP underestimate score would be negative corresponding to an overestimate. We primarily focused on positive CNEP underestimate scores since this can highlight datasets that annotate constrained non-exonic elements that previously had few if any informative annotations. Additionally, we observed more data sets that had relatively large underestimate values compared to overestimate values, which can be seen in **Supplementary Fig. 23**. We have revised **Supplementary Fig. 24** to include an additional panel corresponding to **Fig. 4**, but now showing negative CNEP underestimate values. We also now show negative CNEP underestimate values for the panels previously in **Supplementary Fig. 24**. The updated **Supplementary Fig. 24** is also shown below.

We have also updated the **Fig. 4** legend to state:

“A version of this plot showing negative CNEP underestimate values, and versions of the plot based on the input features used by CNEP to generate the predictions and based on shuffled versions of the additional datasets can be found in **Supplementary Fig. 24**.”

Supplementary Figure 24: CNEP underestimates values for input features and shuffled versions in retrospective analysis of additional data. (a) Similar plots to Fig. 5 showing a scatter plot of CNEP underestimate values, but also including negative CNEP underestimate values, meaning the average observed CNEP score for bases covered by the feature is greater than expected based on the features overlap with CNE bases. **(b,c)** Similar to (a) but for the **(b)** input features to CNEP for the retrospective analysis based on features available in 2015 and **(c)** shuffled versions of the additional datasets considered in (a) and Fig. 5 (Methods). Greater underestimation values for the same genomic coverage is seen in (a) than in both these controls, demonstrating additional marginal additive information about CNE bases in the additional datasets.

Alternatively, the CNEP score at a peak set might be close to the value predicted based on conservation, but the marginal value of that set of peak information might be high if a CNEP score computed without that track of peak information would have been much farther from the expected score.

In this retrospective analysis, we are focused on evaluating the marginal additional information of each data set conditioned on the set of data that was available and accessible at a fixed point of time. The motivation for this is to gain insights into data sets that can still provide additional information about constrained non-exonic bases even when a substantial set of data already exist. Our goal with this specific analysis is not to identify datasets that are marginally informative on their own, but specifically still informative conditioned on the available and accessible data at a fixed point of time.

There could be two functional tracks that have a peak on the same genomic position, and in that case it doesn't appear that this criterion would be able to tell whether the peak is still predicted to be conserved after leaving out only one of the sets.

As explained above since we are interested in the marginal additional information for a set of data if the data set being evaluated is the same as one already available, then the data set should get a CNEP underestimate value close to 0 regardless of whether the dataset is predictive on its own.

I assumed the goal of computing a CNEP underestimation value would be to estimate how much better the model gets at predicting conserved elements when adding a specific set of peaks to the training set, and if this is the correct goal, wouldn't it be better to report the correlation between the CNEP scores computed without this additional peak set and the CNEP scores computed with this peak set added? If I'm misunderstanding the goal of this score, it would be helpful to spell out that goal more explicitly.

We thank the reviewer for the question. We have clarified the goal of this analysis is to identify and understand data sets that can still provide additional information about constrained non-exonic bases not already captured by a set of data already available at a fixed point of time. We see two main advantages for using the CNEP underestimation value compared to the correlation of the predictions with and without the additional peak set. The first advantage is that the CNEP underestimate value is not confounded by the size of the peak set. With the CNEP underestimate value one can more directly identify peak sets that provide highly informative information for the peaks that they cover, but only apply to a more limited set of bases of the genome. The second main advantage is that the CNEP underestimate is much faster to compute as it does not require recomputing the CNEP score. It would not be practical computationally to recompute the CNEP separately for each additional feature.

In response to the above set of points on the CNEP underestimation value we have revised this paragraph in the results section of the main text to state:

"This suggests that some individual peak sets may provide additional marginal information predictive of CNE bases even after considering more than ten thousand features. To identify peak sets that could provide particularly informative additional annotations of CNE bases in the genome after conditioning on the information within the 10,836 features, we defined the CNEP underestimation value for a set of peaks. The CNEP underestimation value for a set of peaks was defined as the difference between the statistics of the expected average CNEP score based on

the peaks overlap with CNE bases and the observed average value of the CNEP score for bases in peaks (**Methods**). A high CNEP underestimation value corresponds to peak sets that provide particularly informative additional annotations for the CNE bases that the peak set overlaps beyond the information in the set of 10,836 features. We note that peak sets that receive a CNEP underestimation value close to zero can still be highly informative of CNE bases when considered in isolation, and that a negative CNEP underestimate value corresponds to the CNEP predictions being higher than expected based on the CNEP overlap with CNE bases. We analyzed the CNEP underestimate values as a function of the number of bases the set of peaks covered.”

Additionally we have added this text to “Retrospective analysis on information in additional human datasets” of the **Methods**

“The CNEP underestimate value for a set of peaks was defined as the difference of its expected average CNEP score and its observed average CNEP score statistic, where the observed CNEP score is computed based on a subset of features available in 2015 as described above. This score evaluates marginal additional information of a set of peaks for annotating CNE bases conditioned on the 2015 subset and was analyzed relative to the number of bases the peak set covers. A property of a logistic regression classifier is that the sum of a feature’s values for the positive instances equals the weighted sum of the feature’s values weighted by the prediction probabilities for the positive class, on the training data without regularization. This leads to the expectation that including a feature would cause its CNEP underestimate to be close to 0. We verified this holds approximately for those features used to generate the CNEP score on its actual predictions (**Supplementary Fig. 23g,h**).

We note that a set of peaks with a CNEP underestimate of 0 can still be informative on constraint in isolation, but here we are evaluating the marginal additional information conditioned on the 2015 subset. Higher positive values correspond to peak sets that have greater overlap with CNE bases than expected based on the CNEP score derived from information in the 2015 subset, while more negative values correspond to fewer overlaps and an overestimate of the CNEP score for the peak set. We primarily focused on positive CNEP underestimate scores since this can highlight datasets that annotate constrained non-exonic elements that previously had few if any informative annotations. Additionally, we observed more data sets that had relatively large CNEP underestimate values compared to overestimate values (**Supplementary Fig. 23**).

The CNEP underestimate values has two advantages compared to directly analyzing the correlations of CNEP predictions with and without each additional feature. One advantage is it allows identifying data sets that might provide a highly informative annotation, but for only a relatively small number of CNE bases. The other advantage is that it is practical to compute for a large number of features since it does not require retraining the CNEP score for each additional feature.”

REVIEWERS' COMMENTS:

Reviewer #1 (Remarks to the Author):

The reviewers have done considerable work to address my previous comments. The manuscript is much clearer and more cohesive. I have the following additional minor comments:

The example in Figure 1b is helpful. Perhaps this is a place where you could mention that not only does CNEP indicate that there exists relevant TF/epigenetic data, but that you could potentially figure out what data are most relevant, since CNEP now allows ranking of datasets based on peaks overlapping a position of interest? Also the Figure 1b legend is a bit long.

Figure S2. Why constrain to non-coding bases for the variant prioritization analysis when CNEs were detected in exonic bases as well?

Figures S10-S14 – consider combining plots when they measure the identical distance (e.g. distance to TSS, distance to exon, proximity to genes so that the curves can be directly compared. Also consider including all relevant categories of base (e.g. low CNE, high CNE, low notCNE, high notCNE) as these might provide additional context for interpreting high notCNE and lowCNE.

Supplementary Table S3 provides a nice overview of the performance of CNEP. The figure legend is a bit confusing and includes details that may be better described in the methods. For example: "The AUC values were computed based on using the same 10 classifiers as the ensemble with classifiers, but split into either two groups of five or ten groups of one and the AUC values based on each group was averaged."

A lot of figures are called out in figure legends. This is helpful in that it links related figures, but may be problematic in some cases with respect to the figure numbering. For example, the figure 1 legend calls out Figure 5a, but Figure 5a is referenced considerably later in the main manuscript.

Also, while considerably improved, there are still a couple of spots with typos or where the language could be a little clearer. A couple of examples:

Figure 4 legend: "Scatter plot where each point corresponds to a dataset, the x-axis is the number of bases it covers, and the y-axis is the prediction underestimate value when using the CNEP score for prediction values."

Figure 2 legend: "bases..Also shown"

'present features'? Maybe 'available features'?

'for a given bases in the genome'

'We applied CNEP using for labels constrained element sets previously produced by four different methods: PhastCons5,32, GERP++2, SiPhy-pi and SiPhy-omega3,4(Methods).' Consider rephrasing. Maybe "Bases were labeled as CNE or notCNE based on ..."

"next to a CNE bases"

Reviewer #2 (Remarks to the Author):

The revised version of this manuscript is much improved. It is now quite a bit clearer what the method is designed to accomplish and what can be learned from the CNEP scores being presented.

That being said, I did still find the narrative somewhat hard to follow because of terms that were not defined or whose meaning only became apparent toward the end of the manuscript. Below I've attempted to delineate these terms more comprehensively than I did has time:

1. A fundamental point of confusion that I still struggled with beginning in the introduction is the question of what defines an "evolutionarily constrained base." Does it have to show evidence of purifying selection within humans? Conservation across apes? Conservation across a wider set of mammals or other vertebrates? Does it have to be completely devoid of variation or is it enough to be statistically significantly depleted of variation compared to some genomic neighborhood or the whole genome? Is it defined as a function of the output of one or more methods for inferring conservation, or is conservation here defined to be a hidden state that depends on parameters of the evolutionary process that other methods are also trying (imperfectly) to infer? I think by the end of the manuscript I understand that a constrained base has a sort of floating definition that's just the arbitrary output of another method for inferring constrained elements, but it would be better to clarify this at the beginning when the term first starts being used. If future users of the method want to try inputting some other constrained element set besides the four that are utilized here, are there fundamental limits on what sorts of conserved element tracks are appropriate to use, or would it be appropriate to use quite different conserved element tracks such as loci that show evidence of very recent stabilizing selection?

2. Although "variant prioritization" is no longer such prominent feature of the motivating narrative, it would still be helpful to define this term since those outside the medical genetics community are not likely to know it.

3. In the last paragraph of page 2, what is the difference between using "evolutionary information" and using "conservation as features"? I would think that all evolutionary information is tantamount to conservation or the lack thereof. Probabilistic evolutionary models of the type used in FitCons definitely use conservation information, since the states of the HMM describe presence/absence of a specific definition of evolutionary constraint.

4. In the first full paragraph of page 4, what is "ConsHMM?" I know a reference is cited, but it'd be good to have a complete enough description of this key method in the main text such that we don't have to go and read this other paper to be able to get something out of this paragraph.

5. Page 5 specifies that CNEP is trained on four different call sets of conserved elements, but it isn't clear at the time that these represent 4 different call sets rather than joint inference from 4 sets of conserved elements. This is alluded to in the next paragraph, but it would be clearer to state up front that CNEP uses 1 arbitrary conserved element set and that these are averaged post hoc.

6. The last paragraph on page 5 states that "We confirmed that the CNEP score, based on averaging the four predictions, was even more highly correlated with scores based on a single constrained element set." It isn't clear what the average score is being correlated with. Is this referring to correlations between the original CNEP scores trained on 1 conserved element set and the final averaged scores? It's tautologically always the case that the average of 4 measurements is more correlated with the 4 original measurements overall than these 4 measurements are with each other.

7. It'd be good to make it clearer in the text which parts of the genome are used for training versus which are used for testing. Is something like leave-one-out cross-validation employed at the level of bases or whole chromosomes or something in between?

8. On page 13, it should be clarified whether the multi species alignments being utilized by CNEP were also utilized by phastcons, fitcons, etc. to call evolutionarily constrained elements.

Reviewer #1 (Remarks to the Author):

The reviewers have done considerable work to address my previous comments. The manuscript is much clearer and more cohesive.

We thank the reviewer for the positive comments. We also thank the reviewer again for their previous review, which led to many of these improvements.

I have the following additional minor comments:

We thank the reviewer for these additional comments, which we have addressed as described below.

The example in Figure 1b is helpful.

We thank the reviewer for the positive comment.

Perhaps this is a place where you could mention that not only does CNEP indicate that there exists relevant TF/epigenetic data, but that you could potentially figure out what data are most relevant, since CNEP now allows ranking of datasets based on peaks overlapping a position of interest?

We thank the reviewer for this suggestion. We have added a mention of this in the figure legend in 1b, which now states:

“The final set of tracks show peak calls for PDX1 and FoxA1 in hESC-derived pancreatic progenitors^{21,50} that overlapped the SNP. These datasets were ranked using the CNEP software to be the top three datasets in terms of their overlap with constrained non-exonic elements as defined by the expected CNEP score statistic.”

Also the Figure 1b legend is a bit long.

We thank the reviewer for the feedback. We have shortened the fig 1b legend. The length of the entire figure 1 legend now fits within the journal’s guidelines of 350 words. The revised Fig 1b now states:

“(b) An example genomic locus containing the *PTF1A* gene illustrating the CNEP score and the CSS-CNEP. The top line is the GENCODE gene annotation track. The next line shows the location of five variants in close proximity that were previously identified to be associated with isolated pancreatic agenesis and fell into constrained non-exonic elements, but had limited prior information epigenomic and TF binding annotations²¹. A vertical box goes through these variants. The following tracks show the CNEP score, CSS-CNEP, and the PhastCons score. This is followed by tracks for the PhastCons, GERP++, SiPhy-omega, and SiPhy-pi^{2,3,5} constrained element sets and the ConsHMM annotations³² in dense view. Along the bottom is a color legend for the groups of ConsHMM conservation states as defined in Ref. 32. For segments in red, species through fish have high aligning and matching to the human reference genome, while those in orange have that through some non-mammalian vertebrates. The next sets of tracks show baseline scores of counts of total features, DNase-seq features, and DNase-seq features from Roadmap Epigenomics overlapping a base followed by the FitCons2 cell type integrated score²² and the Segway Encyclopedia conservation-associated activity

score²⁴. The final set of tracks show peak calls for PDX1 and FoxA1 in hESC-derived pancreatic progenitors^{21,50} that overlapped the SNP. These datasets were ranked using the CNEP software to be the top three datasets in terms of their overlap with constrained non-exonic elements as defined by the expected CNEP score statistic.”

We also moved part of the Fig 1b legend into the results section of the main text, which now states:

“For example the initial discovered associated variant (chr10:23508437) was in the top 5.4% by CNEP, while at the top 22.8% for the Segway Encyclopedia ‘conservation-associated activity score’²⁴ and at the top 46.8-49.4% for the FitCons2 ‘cell-type integrated score’²² and three baseline scores (**Methods**).”

Figure S2. Why constrain to non-coding bases for the variant prioritization analysis when CNEs were detected in exonic bases as well?

We thank the reviewer for the question. To clarify CNEs (constrained non-exonic bases) by definition are not in exons. The CNEP predictions were made in exons, but were not trained to predict exon associated constraint. Many of the top prioritized variants by variant prioritization scores do fall in exons and in particular coding sequences. The focus of CNEP is on non-exon regions as exons are already well annotated and known to be heavily constrained. We feel it would provide a more meaningful motivation if we show that top prioritized variants outside of exons also overlap constrained elements.

To clarify on this point we have added this sentence to the Fig. S2 legend:

“We restricted to non-exonic regions as exons are already well annotated and CNEP is trained to predict constraint in non-exonic regions.”

Figures S10-S14 – consider combining plots when they measure the identical distance (e.g. distance to TSS, distance to exon, proximity to genes so that the curves can be directly compared. Also consider including all relevant categories of base (e.g. low CNE, high CNE, low notCNE, high notCNE) as these might provide additional context for interpreting high notCNE and lowCNE.

We thank the reviewer for the suggestions. We have reorganized and updated the spatial enrichment supplementary figures (10-14 and also 6-7), which are now Supplementary Figures 6-9. We now have one figure each for spatial enrichments relative to nearest CNE, exon, TSS, and gene. In each of these figures we display enrichments of Low_notCNE and High_notCNE bases relative to the enrichment of notCNE bases. We also display in these figures enrichments of Low_CNE and High_CNE bases relative to the enrichment of CNE bases with the exception

of nearest CNE where we only show it for Low_notCNE and High_notCNE bases (as the distance is always 0 for CNE bases). The updated figures for PhastCons are now shown below.

Supplementary Figure 6: Spatial enrichments relative to CNE bases. (a) The plot shows the fold enrichment for the cumulative number of PhastCons High_notCNE and Low_notCNE bases relative to the enrichment of notCNE bases for bases being within each distance to the nearest CNE base, up to 3,000bp. (b) The plot shows the cumulative fraction of PhastCons High_notCNE, Low_notCNE, High_CNE, and Low_CNE bases at each distance to the nearest CNE base up to 3,000 bp.

Supplementary Figure 7: Spatial enrichments relative to exons. (a) The plot shows the fold enrichment for the cumulative number of PhastCons Low_CNE and High_CNE bases relative to the enrichment of CNE bases (PhastCons) for bases being within each distance to the nearest exon, up to 3,000bp, and similarly for Low_notCNE and High_notCNE relative to notCNE bases. (b) The plot shows the cumulative fraction of PhastCons Low_CNE, High_CNE, Low_notCNE, and High_notCNE bases at each distance to the nearest exon, up to 3,000bp.

Supplementary Figure 8: Spatial enrichments relative to transcription start sites. (a) The plot shows the fold enrichment for the cumulative number of PhastCons Low_notCNE and High_notCNE bases relative to the enrichment of notCNE bases for bases being within each

distance to a transcription start site, up to 100kb, and similarly for Low_CNE and High_CNE bases relative to the enrichment of CNE bases. **(b)** The plot shows the cumulative fraction of PhastCons Low_CNE, High_CNE, Low_notCNE, and High_notCNE bases at each distance to the nearest transcription start site, up to 100kb.

Supplementary Figure 9: Spatial enrichments relative to genes. (a) The plot shows the fold enrichment for the cumulative number of PhastCons Low_notCNE and High_notCNE bases relative to the enrichment of notCNE bases for bases being within each distance to a gene, up to 100kb, and similarly for Low_CNE and High_CNE bases relative to the enrichment of CNE bases. **(b)** The plot shows the cumulative fraction of PhastCons Low_CNE bases at each distance to a gene, up to 100kb.

Supplementary Table S3 provides a nice overview of the performance of CNEP.

We thank the reviewer for the positive comment.

The figure legend is a bit confusing and includes details that may be better described in the methods. For example:

“The AUC values were computed based on using the same 10 classifiers as the ensemble with classifiers, but split into either two groups of five or ten groups of one and the AUC values based on each group was averaged.”

We thank the reviewer for the feedback. We have revised the figure legend to make it clearer. We have move parts of it to the methods including the indicated sentence or removed parts that were already included in the methods. The legend now reads:

“Supplementary Table 1: AUC performance at predicting CNE bases – The table reports the area under the ROC curves (AUC) of CNEP and other scores for predicting CNE bases. AUC values are shown for four constrained non-exonic element sets when bases overlapping exons are included in the negative set (left) and when they are excluded from both the positive and negative sets for the evaluations (right). The first row of the table provides the AUC values for the CNEP score. The second row of the table shows the AUC values for CNEP predictions based on training on only the constrained element set used in the evaluations instead of averaging predictions based on training on four different constrained elements sets. The next

two rows show the predictive performance when CNEP uses five and one classifier per chromosome instead of ten. The next row shows the performance of CNEP with the feature subset used in the retrospective analysis, all of which were available by 2015. The following two rows report the performance of CNEP using all features except ChromHMM features and the performance of CNEP when using only the ChromHMM features. The next three rows report the performance of three baselines based on counting either all overlapping features, all overlapping DNase I features, or just overlapping Roadmap Epigenomics DNase I features. The following two rows report the performance of two existing scores, the Segway Encyclopedia 'conservation-associated activity score'³ and FitCons2 'cell-type integrated scores'⁴. After the section break, three rows report the performance on a sample of 50,000 randomly sampled genomic positions. The first of those is the CNEP score followed by the maximum and average performance of DeepSea predictions for 919 chromatin features⁵. In the last section, two rows that report performance on a million randomly sampled positions on chr10. These rows report CNEP's performance using a logistic regression classifier and using a random forest classifier in place of a logistic regression classifier. Source data are provided as a Source Data file."

A lot of figures are called out in figure legends. This is helpful in that it links related figures, but may be problematic in some cases with respect to the figure numbering. For example, the figure 1 legend calls out Figure 5a, but Figure 5a is referenced considerably later in the main manuscript.

We thank the reviewer for pointing this out. We have removed the reference to Figure 5a in Figure 1 and placed a color legend directly in Figure 1. We are providing a reference to Arneson and Ernst, 2019. The portion of the Fig 1b legend referencing ConsHMM now states:

"The following tracks show the CNEP score, CSS-CNEP, and the PhastCons score. This is followed by tracks for the PhastCons, GERP++, SiPhy-omega, and SiPhy-pi^{2,3,5} constrained element sets and the ConsHMM annotations³² in dense view. Along the bottom is a color legend for the groups of ConsHMM conservation states as defined in Ref. 32. For segments in red, species through fish have high aligning and matching to the human reference genome, while those in orange have that through some non-mammalian vertebrates."

The other references to figures from the main figure legends are to supplementary figures and do not appear to pose a problem in terms of figure numbering.

Also, while considerably improved, there are still a couple of spots with typos or where the language could be a little clearer. A couple of examples:

We thank the reviewer for recognizing the improvement, and pointing out that there are still remaining issues. We have addressed each instance the author noted below, and made additional edits throughout the text to improve clarity.

Figure 4 legend: "Scatter plot where each point corresponds to a dataset, the x-axis is the number of bases it covers, and the y-axis is the prediction underestimate value when using the CNEP score for prediction values."

We thank the reviewer for pointing this out. We have rephrased this so it now reads:

“Scatter plot of additional human datasets in a retrospective analysis. The x-axis is the number of bases covered by the dataset. The y-axis is the CNEP underestimation value for the dataset.”

Figure 2 legend: “bases..Also shown”

We thank the reviewer for pointing this out. We have edited this so it now reads

“bases. Also shown”

‘present features’? Maybe ‘available features’?

We thank the reviewer for pointing this out. We have revised the entire sentence that contained that phrase. The sentence now reads

“The next three rows report the performance of three baselines based on counting either all overlapping features, all overlapping DNase I features, or just overlapping Roadmap Epigenomics DNase I features.”

‘for a given bases in the genome’

We thank the reviewer for pointing this out. We have made addition edits to the phrase which now reads

“The median number of features that overlap a base in the genome is 532”

‘We applied CNEP using for labels constrained element sets previously produced by four different methods: PhastCons^{5,32}, GERP++², SiPhy-pi and SiPhy-omega^{3,4}(Methods).’
Consider rephrasing. Maybe “Bases were labeled as CNE or notCNE based on ...”

We thank the reviewer for pointing out this sentence was unclear. We have revised this sentence and moved it earlier in the paragraph where we also address comment 5 of Reviewer 2. The sentences at the beginning of the paragraph now read:

“CNEP trains an ensemble of logistic regression classifiers to discriminate between bases overlapping evolutionarily constrained elements outside of annotated exons and those bases in the rest of the genome (**Methods**). We trained CNEP with multiple different constrained element sets by applying CNEP separately with each, and then averaging the resulting predictions producing what we termed the CNEP score (**Fig. 2a, Supplementary Fig. 2**). We applied CNEP with constrained element sets previously produced by four methods: PhastCons^{5,31}, GERP++², SiPhy-pi and SiPhy-omega^{3,4}.”

“next to a CNE bases”

We thank the reviewer for pointing this out. We have changed this to read:

“next to a CNE base”

Reviewer #2 (Remarks to the Author):

The revised version of this manuscript is much improved. It is now quite a bit clearer what the method is designed to accomplish and what can be learned from the CNEP scores being presented.

We thank the reviewer for the positive comments. We apologize again about the previous clarity issues. We thank the reviewer again for their previous comments, which helped lead to these improvements.

That being said, I did still find the narrative somewhat hard to follow because of terms that were not defined or whose meaning only became apparent toward the end of the manuscript. Below I've attempted to delineate these terms more comprehensively than I did has time:

We thank the reviewer for feedback and apologize that parts of the manuscript were still unclear. We have further revised the manuscript in response to each specific comment as described below and made other edits to improve clarity.

1. A fundamental point of confusion that I still struggled with beginning in the introduction is the question of what defines an “evolutionarily constrained base.” Does it have to show evidence of purifying selection within humans? Conservation across apes? Conservation across a wider set of mammals or other vertebrates? Does it have to be completely devoid of variation or is it enough to be statistically significantly depleted of variation compared to some genomic neighborhood or the whole genome? Is it defined as a function of the output of one or more methods for inferring conservation, or is conservation here defined to be a hidden state that depends on parameters of the evolutionary process that other methods are also trying (imperfectly) to infer? I think by the end of the manuscript I understand that a constrained base has a sort of floating definition that’s just the arbitrary output of another method for inferring constrained elements, but it would be better to clarify this at the beginning when the term first starts being used. If future users of the method want to try inputting some other constrained element set besides the four that are utilized here, are there fundamental limits on what sorts of conserved element tracks are appropriate to use, or would it be appropriate to use quite different conserved element tracks such as loci that show evidence of very recent stabilizing selection?

We thank the reviewer for raising these points. We have revised a sentence in the introduction to be explicit that we are using the constrained non-exonic elements as defined by prior methods by stating:

“Here we develop the Constrained Non-Exonic Predictor (CNEP) to produce a single probabilistic score per base, without respect to cell type, that reflects the evidence within a large-scale compendium of epigenomic and TF binding data that the base is in a non-exonic evolutionarily constrained element, as defined by prior methods for calling them²⁻⁵.”

We also added a citation to these methods again in our first sentence of the results

“We developed the Constrained Non-Exonic Predictor (CNEP) to make a probabilistic prediction using features defined from large-scale epigenomics and TF binding data as to whether a base

in the human genome is in a constrained non-exonic element previously called from comparative genomics sequence analysis²⁻⁵ (**Fig. 1**)."

The CNEP software is general and can support using other types of conservation annotations in place of the constrained elements we used here. How effective the predictions will depend on the specifics of what these other annotations are and would be an interesting direction for future work. We have added a sentence to the discussion on this point:

"A direction for future work could be to investigate training CNEP so it is optimized to detect other types of conservation annotations."

2. Although "variant prioritization" is no longer such prominent feature of the motivating narrative, it would still be helpful to define this term since those outside the medical genetics community are not likely to know it.

We thank the reviewer for pointing this out. We have removed the use of the term "variant prioritization" from the main text and instead replaced it with more informative wording. The following are the parts of the results and discussion that we revised to no longer use "variant prioritization"

"These constrained element sets highly enrich for non-exonic bases prioritized by a number of scores used to predict the relative phenotypic impact of genetic variants, including scores that integrate diverse annotations (**Supplementary Fig. 3**)."

"We note that CNEP and CSS-CNEP are not specifically designed to be used directly for the task of prioritizing phenotypic associated genetic variants, and would not be expected to be competitive with scores designed for this task that use more diverse sets of features. A possible direction of future work is to evaluate incorporating the CNEP and CSS-CNEP scores with other features for the task of prioritizing variants."

3. In the last paragraph of page 2, what is the difference between using "evolutionary information" and using "conservation as features"? I would think that all evolutionary information is tantamount to conservation or the lack thereof. Probabilistic evolutionary models of the type used in FitCons definitely use conservation information, since the states of the HMM describe presence/absence of a specific definition of evolutionary constraint.

We thank the reviewer for pointing out this is unclear. We are not trying to form a distinction between conservation features and evolutionary information. Rather we are forming a distinction between scores that use evolutionary information to learn a mapping from epigenomic and TF features from scores where evolutionary information features are integrated with epigenomic features, which were discussed in the previous paragraph in the introduction. Also we think the reviewer is describing PhastCons and not FitCons. FitCons does not use a HMM modeling presence/absence of a specific definition of constraint. FitCons learns a mapping from epigenomic annotations to a score based on evolutionarily information.

We have revised the text to now state:

“A few scores have been proposed that quantify information or activity in epigenomic or TF binding annotations where evolutionary information is used to learn a mapping from epigenomic or TF binding features to a score, but not as features for predictions.”

4. In the first full paragraph of page 4, what is “ConsHMM?” I know a reference is cited, but it’d be good to have a complete enough description of this key method in the main text such that we don’t have to go and read this other paper to be able to get something out of this paragraph.

We thank the reviewer for pointing this out. We have removed the entire sentence containing “ConsHMM” from the introduction as we realized that this part was getting too detailed for the introduction. We provide more information about ConsHMM when we first mention it in the results section, which we have expanded in the revised version to make explicit the connection to ChromHMM. This part of the results text now states:

“Specifically for this analysis we used ConsHMM conservation states annotations based on the combinatorial and spatial patterns of which species match and align the human genome in a 100-way vertebrate sequence alignment, defined in an analogous way to ChromHMM chromatin states^{32,38}.”

5. Page 5 specifies that CNEP is trained on four different call sets of conserved elements, but it isn’t clear at the time that these represent 4 different call sets rather than joint inference from 4 sets of conserved elements. This is alluded to in the next paragraph, but it would be clearer to state up front that CNEP uses 1 arbitrary conserved element set and that these are averaged post hoc.

We thank the reviewer for the suggestion. We have revised the main text to now state:

“CNEP trains an ensemble of logistic regression classifiers to discriminate between bases overlapping evolutionarily constrained elements outside of annotated exons and those bases in the rest of the genome (**Methods**). We trained CNEP with multiple different constrained element sets by applying CNEP separately with each, and then averaging the resulting predictions producing what we termed the CNEP score (**Fig. 2a, Supplementary Fig. 2**). We applied CNEP with constrained element sets previously produced by four methods: PhastCons^{5,31}, GERP++², SiPhy-pi and SiPhy-omega^{3,4}.”

6. The last paragraph on page 5 states that “We confirmed that the CNEP score, based on averaging the four predictions, was even more highly correlated with scores based on a single constrained element set.” It isn’t clear what the average score is being correlated with. Is this referring to correlations between the original CNEP scores trained on 1 conserved element set and the final averaged scores? It’s tautologically always the case that the average of 4 measurements is more correlated with the 4 original measurements overall than these 4 measurements are with each other.

We thank the reviewer for pointing out that this paragraph was unclear. Yes, this is referring to the original CNEP scores trained on one constrained element set and the final average score.

While it may not be surprising the correlation based on the average is greater than any individual element with each other, it does not tautologically have to be the case. For example if two of the individual scores were perfectly correlated with each other, but not two other scores, then the average would be less correlated with some individual scores than at least one pair of individual scores.

We have revised the text to make the paragraph clearer and also to reflect the changes made to address point 5 earlier. Specifically the paragraph now states:

“CNEP’s predictions from training using any two different individual constrained element sets were all highly correlated ([0.88,0.93]), and greater than the correlations directly between different constrained element sets (**Supplementary Fig. 4**). As the CNEP score is based on the average of predictions based on training separately with four different constrained element sets, this is expected to reduce biases associated with the choice of one specific constrained element set. Consistent with this, the CNEP score correlation with CNEP’s predictions based on training on a single constrained element set were even higher ([0.96,0.98]).”

7. It’d be good to make it clearer in the text which parts of the genome are used for training versus which are used for testing. Is something like leave-one-out cross-validation employed at the level of bases or whole chromosomes or something in between?

We thank the reviewer for the question. We leave the entire chromosome out from training when we generate predictions for it. To clarify on this point we have added these sentences to the method

“We conducted these comparisons genome-wide. Since CNEP predictions were made for each chromosome by leaving that chromosome out when training, there was no need to create additional training and test sets for these comparisons.”

We note that we also have had this sentence main text

“For each chromosome, CNEP trains a separate set of classifiers based on subsamples of positions from all chromosomes except the target chromosome.”

and have had this sentence in the methods:

“For generating CNEP scores on one chromosome based on one constrained element set, CNEP trained ten logistic regression classifiers using ten different sets of 1,000,000 randomly sampled positions from the other 22 chromosomes.”

8. On page 13, it should be clarified whether the multi species alignments being utilized by CNEP were also utilized by phastcons, fitcons, etc. to call evolutionarily constrained elements.

We thank the reviewer for the suggestion. We first note we assume the reviewer means ‘ConsHMM’ instead of CNEP as CNEP does not directly use multi-species alignments, it only indirectly uses them through the constrained element sets that provide labels. PhastCons

elements were defined from the same alignment as ConsHMM. We now note in the results section

“For example, for PhastCons, which was defined on the same alignment as ConsHMM,..”

We note FitCons was not used to call evolutionarily constrained elements. The other constrained element sets CNEP used besides PhastCons were SiPhy-omega, SiPhy-pi, and GERP++. SiPhy-omega, SiPhy-pi, and GERP++ were defined from different alignments than PhastCons and ConsHMM. The specific alignments each of these constrained element sets were defined based on is specified in the ‘Constrained elements set’ subsection of the methods.